# The role of dew and radiation fog inputs in the local water cycling of a temperate grassland during dry spells in Central Europe

Yafei Li[1], Franziska Aemisegger[2], Andreas Riedl[1], Nina Buchmann[1], Werner Eugster[1]

[1]Institute of Agricultural Sciences, ETH Zurich, Zurich, Switzerland

[2]Institute for Atmospheric and Climate Science, ETH Zurich, Zurich, Switzerland

*Correspondence to*: Yafei Li (yafei.li@usys.ethz.ch); Werner Eugster (werner.eugster@usys.ethz.ch)

**Abstract**. During dry spells, non-rainfall water (hereafter NRW) mostly formed from dew and fog potentially plays an increasingly important role in temperate grassland ecosystems with ongoing global warming. Dew and radiation fog occur in combination during clear and calm nights, and both use ambient water vapor as a source. Research on the combined mechanisms involved in NRW inputs to ecosystems is rare, and distillation of water vapor from the soil as a NRW input pathway for dew formation has hardly been studied. Furthermore, eddy covariance (EC) measurements are associated with large uncertainties in clear calm nights when dew and radiation fog occur. The aim of this paper is thus to use stable isotopes as tracers to investigate the different NRW input pathways into a temperate Swiss grassland at Chamau during dry spells in summer 2018. Stable isotopes provide additional information on the pathways from water vapor to liquid water (dew and fog) that cannot be measured otherwise. We measured the isotopic composition ($\delta^{18}O$, $\delta^{2}H$, and $d = \delta^{2}H - 8 \cdot \delta^{18}O$) of ambient water vapor, NRW droplets on leaf surfaces, and soil moisture, and combined them with EC and meteorological observations during one dew-only and two combined dew and radiation fog events. The ambient water vapor $d$ was found to be strongly linked with local surface relative humidity ($r = -0.94$), highlighting the dominant role of local moisture as a source for ambient water vapor in the synoptic context of the studied dry spells. Detailed observations of the temporal evolution of the ambient water vapor and foliage NRW isotopic signals suggest two different NRW input pathways: (1) the downward pathway through the condensation of ambient water vapor; (2) the upward pathway through the distillation of water vapor from soil onto foliage. We employed a simple two end-member mixing model using $\delta^{18}O$ and $\delta^{2}H$ to quantify the NRW inputs from these two different sources. With this approach, we found that distillation contributed 9–42 % to the total foliage NRW, which compares well with estimates derived from a near-surface vertical temperature gradient method proposed by Monteith in 1957. The dew and radiation fog potentially produced 0.17–0.54 mm d$^{-1}$ NRW gain on foliage, thereby constituting a non-negligible water flux to the canopy, as compared to the evapotranspiration of 2.7 mm d$^{-1}$. Our results thus underline the importance of NRW inputs to temperate grasslands during dry spells and reveal the complexity of the local water cycle in such conditions, including different pathways of dew and radiation fog water inputs.

## 1 Introduction

The role of dew and fog inputs in the hydrological cycle is well understood in desert areas, where rainfall totals are small (Malek et al., 1999; Jacobs et al., 2002; Kidron et al., 2002; Agam and Berliner, 2006; del Prado and Sancho, 2007; Pan et al., 2010; Ucles et al., 2013; McHugh et al., 2015). Such water inputs are, however, mostly neglected in regions where average rainfall is abundant, and thus the expected water gains from dew of up to 0.7–0.8 L m$^{-2}$ d$^{-1}$ during nights with perfect clear-sky conditions (Beysens, 2018), or fog providing on the order of 8.5 L m$^{-2}$ d$^{-1}$ in tropical montane cloud forests (Bruijnzeel et al., 2006) appear to be small and negligible in comparison to average precipitation rates. However, during dry spells, especially during the warm season when daily evapotranspiration rates are high, it can be expected that, although small, non-rainfall water (hereafter NRW) inputs from various sources (see below) may become essential for the vegetation to alleviate stress

(Tuller and Chilton, 1973). This may even be the case in temperate climates, where average annual precipitation typically balances or exceeds actual annual evapotranspiration. Grasslands tend to be the first to suffer from prolonged dry spells and droughts (Wolf et al., 2013). Here we investigate the small-scale processes of how fog and dew water influence the water cycling over a grassland at a Central European temperate climate site during representative warm-season nights.

Rainfall measurements with conventional rain gauges collect liquid and solid precipitation (Glickman and Zenk, 2000), and thus the vast amount of above-ground water entering the vegetation canopy in wet climates, but in temperate and even more pronounced in dry climates some important components of the hydrological cycle are missed, e.g., NRW inputs. NRW inputs include a number of components: (1) dew formation (Monteith, 1957); (2) fog deposition (Dawson, 1998); (3) water vapor adsorption (Agam and Berliner, 2006); (4) rime ice deposition (Hindman et al., 1983); (5) hoar frost (Monteith and Unsworth, 2013); and (6) guttation (Long, 1955). During extended periods without rainfall, it is well known that mainly dew and fog (out of the long list of NRW components) are essential water sources for plants in (1) arid and semi-arid regions (Malek et al., 1999; Jacobs et al., 2002; Kidron et al., 2002; Agam and Berliner, 2006; del Prado and Sancho, 2007; Kidron and Temina, 2013; Ucles et al., 2013; He and Richards, 2015; McHugh et al., 2015; Tomaszkiewicz et al., 2017); (2) Mediterranean coastal regions (Beysens et al., 2007); (3) temperate ecosystems (Jacobs et al., 2006); and (4) tropical climates (Clus et al., 2008). In clear calm nights when dew and radiation fog occur, the atmospheric boundary layer becomes stably stratified, leading to a shallow stable nocturnal boundary layer (hereafter NBL) with a depth on the order of no more than 50–100 m (Garratt, 1992). Dew and radiation fog occur at the bottom of the NBL (Stull, 1988; Garratt, 1992; Oke, 2002; Monteith and Unsworth, 2013). Both dew and radiation fog are formed due to the cooling of the Earth's surface after sunset by long-wave radiation losses in clear nights (Oke, 2002). This radiative cooling is a process due to which a body loses heat by long-wave thermal radiation, whereby its surface cools below the dew point of the adjacent air. Under such conditions, dew can form on plant surfaces while fog forms on activated aerosol particles in the near-surface atmosphere.

NRW inputs contribute to the water budget across many ecosystems including croplands (Atzema et al., 1990; Wen et al., 2012; He and Richards, 2015; Meng and Wen, 2016; Tomaszkiewicz et al., 2017), grasslands (Jacobs et al., 2006; Wen et al., 2012; He and Richards, 2015), and forests (Fritschen and Doraiswamy, 1973; Dawson, 1998; Lai and Ehleringer, 2011; Hiatt et al., 2012; Berkelhammer et al., 2013). As compared to forests, grasslands present favorable conditions for dew and radiation fog formation: 1) cooler surface due to a higher albedo and thus lower net solar radiation input (Moore, 1976), as well as higher evapotranspiration (Kelliher et al., 1993; Williams et al., 2012); 2) weaker aerosol particle deposition due to shorter roughness length of grasslands (Gallagher et al., 2002), and thus more aerosol particles remain in the near-surface atmosphere, which consequently results in better conditions for radiation fog formation over grasslands. From the perspective of ecological functioning, small amounts of NRW inputs have a more important influence on grasslands than forests because of a reduced capability to increase crop water use efficiency (WUE), defined as gross carbon uptake per unit water lost, when water availability is low (Wolf et al., 2013), but also due to lower soil water availability and shallower rooting depth in grasslands. At the beginning of drought stress in ecosystems, forests increase their WUE by closing their stomata, which increases stomatal resistance and thus reduces evapotranspiration, while grasslands maintain their evapotranspiration as long as the soil water is available to supply the evaporative demand (e.g., Wolf et al. (2013)). Therefore, grasslands are more prone to suffer from soil water scarcity. In addition, as opposed to the deep-rooted systems for forest plants, grassland plants take up water from the topsoil, where scarcity of soil water occurs more frequently in the absence of precipitation, therefore grasslands tend to anticipate lower soil water availability compared to forests.

Ambient water vapor is the main vapor source for both dew and radiation fog; therefore, dew and radiation fog usually occur in combination. Because of the variability of temperature and humidity conditions, a single NRW night may transit from dew only to intermittent dew and radiation fog in combination. Before the atmospheric humidity reaches saturation at the standard measurement height at 2 m a.g.l., dew can only form if the surface temperature drops below air temperature. When the ambient water vapor reaches saturation or even super-saturation, dew and radiation fog can form intermittently. Kaseke et

al. (2017) used hydrogen and oxygen stable isotope regression to separate the different types of dew and fog, but they focused on dew and fog events separately. Research that focusses on relevant phase change processes during dew and radiation fog in combination is thus rare.

The moisture movement in the soil–plant–atmosphere continuum has been well understood by eddy covariance (hereafter EC) technique, but the reliability of the method suffers during nighttime with weak turbulence (Berkelhammer et al., 2013) when dew and radiation fog occur. In principle, downward water vapor flux measured by EC should provide a quantitative estimate of dew formation on the vegetation surface (termed "phantom dew" by Gay et al. (1996)). The results by Jacobs et al. (2006), however, showed that dew formation quantified by EC was less than one third of the dew amount measured by lysimeter (estimated from Figure 1 in their paper). Moreover, katabatic cold-air drainage flows in non-flat topography lead to advective fluxes that are not directly captured by EC measurements (e.g., Eugster and Siegrist (2000), and Sun et al. (2006)), which typically leads to a gap in the local energy budget $Rn = H + LE + G + \Delta Q$, with Rn net all-wave radiation, H sensible heat flux, LE latent heat flux, and $\Delta Q$ the energy flux to close the budget (see also Wilson et al. (2002), and Franssen et al. (2010)), which makes estimates of dew formation during calm nights highly uncertain and unreliable if $\Delta Q \neq 0$, and are thus not further addressed in this paper.

Monteith (1957) identified two input pathways for dew formation: 1) the downward pathway through the condensation of ambient water vapor on the plants and/or on soil surface, and 2) the upward pathway through distillation of water vapor from soil onto plant surfaces. Soil vapor diffusion from the soil to the atmosphere is driven by the temperature gradient between the soil and the atmosphere, and between different depths of the soil (Monteith, 1957; Oke, 1970). The temperature gradient generally reaches a maximum at the soil–atmosphere interface, where soil surface is roughly 2–5 °C warmer than the adjacent air at 1 cm a.g.l. for short grass cover according to Monteith (1957). The diffusing soil vapor can therefore condense onto cooler foliage. Since Monteith (1957) had quantified the downward and upward components of dew formation by absorbing NRW on foliage with filter paper, research has rarely focused on distinguishing these two pathways of dew formation. Furthermore, Monteith (1957) distinguished the two pathways by collecting NRW in separate nights when only one or the other of the two pathways was assumed to occur. In Monteith (1957), distillation of water vapor from soil as one component of NRW was quantified in very calm nights with a 2 m wind speed (hereafter $u_{2m}$) of less than 0.5 m s$^{-1}$, whereas the maximum NRW condensing from ambient water was assumed to occur in slightly windy nights with $u_{2m}$ in the range of 2–3 m s$^{-1}$. However, for clear calm nights with $u_{2m}$ between 0.5 and 2 m s$^{-1}$, condensation of ambient water vapor and soil-diffusing vapor can occur in combination, with NRW on the foliage being a mix from these two pathways. Research focusing on distinguishing and quantifying the ratio of these two NRW components, i.e., NRW from ambient water vapor, and distillation is scarce.

When the condensation of ambient water vapor and distillation occur simultaneously, the partitioning of NRW into these two components becomes difficult because there is no direct measurement possible to quantify distillation amounts. Hydrometric approaches, e.g., using lysimeters, can easily quantify the condensation amount of ambient water vapor, but cannot quantify the distillation amount, if the water vapor condensing on the above-ground parts (e.g. leaf surfaces) of the lysimeter stems from the below-ground part (soil) of the same lysimeter without a net change in lysimeter weight. Monteith (1957) provided the equations to calculate the distillation rate through measuring the soil surface temperature, and air temperature at 1 cm a.g.l. (see Sect. 3.2.5), which he compared with filter paper measurements and interpreted that the "agreement was reasonable", with a mean ratio of observed vs. calculated distillation of 0.76 (i.e., calculated distillation was ≈ 32% higher than the observed distillation; see Monteith (1957)). The disagreement according to Monteith (1957) is not only related to the unknown collection efficiency of the filter paper he used, but may have arisen from errors or uncertainties in the following three assumptions: 1) the assumption of purely molecular and thus nonturbulent transfer, 2) the assumption of linear (not curvilinear or exponential) temperature gradient, and 3) the assumption of saturation at the soil surface that air in direct contact with the soil may be undersaturated if the 1-cm temperature is lower than the soil surface temperature. To overcome the above-mentioned challenges of quantifying distillation with traditional methods such as EC, filter paper or the vertical

temperature gradient method by Monteith (1957), a useful approach to quantify the ratio between condensation and distillation is the use of stable isotopes: NRW inputs from ambient water vapor and from distillation carry different isotopic signatures due to their different sources, i.e. the atmosphere and the soil moisture respectively. Therefore, a two end-member mixing model using stable isotopes in water (Keeling, 1958; Dawson, 1998; Phillips et al., 2005) can be employed to quantify the individual contributions of these two sources (see details in Sect. 3.2.4).

Our aim was thus to (1) investigate the isotopic fractionations during dew-only and dew–fog combined events, and (2) estimate the contribution of NRW from atmospheric vapor and from soil-diffusing vapor. We carried out three 24 h observation campaigns during summer 2018 using stable isotopes combined with EC and meteorological measurements to characterize the meteorological conditions, to analyze the isotope fractionation of dew and radiation fog formation, to quantify the NRW contribution from ambient water vapor and soil-diffusing vapor, and to explore the potential role of dew and radiation fog during dry spells in temperate grasslands.

## 2 Background

### 2.1 Hydrogen and oxygen isotopes

Hydrogen and oxygen stable isotopes are a useful research tool to investigate the dynamics of the continental water cycle (Aemisegger et al., 2014; Huang and Wen, 2014; Delattre et al., 2015; Parkes et al., 2017), and can therefore be used to trace dew formation and radiation fog deposition into ecosystems (Spiegel et al., 2012; Wen et al., 2012; Delattre et al., 2015; He and Richards, 2015; Parkes et al., 2017). The isotopic composition of a water sample is expressed in terms of the abundance of hydrogen ($^2$H and $^1$H) or oxygen ($^{18}$O and $^{16}$O) isotopes by using the delta notation (hereafter $\delta$) as $\delta = (R_{sample}/R_{standard} - 1) \cdot 1000$ ‰, where $R_{sample}$ and $R_{standard}$ are the molar ratios of either $^2$H/$^1$H or $^{18}$O/$^{16}$O for the sample and standard, respectively. Water molecules with different isotopes are termed isotopologues. Three isotopologues, i.e., $^1$H$_2$$^{16}$O, $^1$H$_2$$^{18}$O, and $^1$H$^2$H$^{16}$O are the most abundant in the water cycle. During phase changes, such as evaporation and condensation, heavier isotopologues (i.e., $^1$H$_2$$^{18}$O, and $^1$H$^2$H$^{16}$O) become enriched in the liquid phase, and depleted in the gaseous phase, which thus causes an increase of $\delta^2$H and $\delta^{18}$O in the liquid phase, and a decrease of $\delta^2$H and $\delta^{18}$O in the gaseous phase. During the evaporation and condensation processes, equilibrium fractionation always occurs at the interface between two phases, and results in a ratio of 1:8 between the variability of $\delta^{18}$O and $\delta^2$H. When the ambient air is unsaturated, a deviation from the 1:8 ratio becomes measurable due to non-equilibrium fractionation (Dansgaard, 1964) driven by faster molecular diffusivity of the lighter isotopologue (i.e., $^1$H$_2$$^{16}$O) than its heavier counterparts (i.e., $^1$H$_2$$^{18}$O and $^1$H$^2$H$^{16}$O). The second order parameter deuterium excess (hereafter $d$), defined as $d = \delta^2$H $- 8 \cdot \delta^{18}$O after Dansgaard (1964), is a useful measure of non-equilibrium fractionation and provides information complementary to $\delta^2$H and $\delta^{18}$O. The $d$ is often used as a tracer for the water vapor source of a given water pool in the water cycle (Gat, 1996; Yakir and Sternberg, 2000; Yepez et al., 2003; Welp et al., 2012; Aemisegger et al., 2014; Galewsky et al., 2016). For example, at the local scale, as compared to the higher $d$ vapor of entrained free tropospheric air, local evapotranspiration is a vapor source with lower $d$, because soil water vapor at the evaporation front had a lower $d$ value (Delattre et al., 2015; Parkes et al., 2017). The diurnal cycle of deuterium excess in a well-mixed convective boundary layer has been studied previously (e.g., Lai and Ehleringer (2011)), whereas relevant processes affecting $d$ in the NBL are much less well known, in particular over grasslands.

### 2.2 Excluding the confusion of guttation

Long (1955) pointed out that guttation droplets distributed on the edges of plant leaves and can easily be mistaken by observers for dew droplets. Dew is however distinct from guttation, which is the exudation of drops of liquid from the hydathodes of the leaves of grasses driven by root pressure (Long, 1955; Stocking, 1956; Hughes and Brimblecombe, 1994). Both dew and guttation occur under high relative humidity. A soil water content near field capacity is favorable for guttation, whilst dew can

also occur at very low soil water contents. In our study, we exclusively focused on the role of NRW during warm-season dry spells when soil water content in the main rooting zone was rather low, closer to the wilting point than to the field capacity, and hence guttation can be neglected here. Furthermore, guttation could easily be distinguished from dew by analyzing the stable isotopes of the respective water component: guttation stems from plant-internal water, whilst dew is plant-external water condensed from ambient water vapor or distilled from vapor related to the soil water isotopic signals. Consequently, the isotopic composition of guttation droplets should vary by species in parallel with the plant-internal water, because no isotopic fractionation is expected during the guttation process. In all our samples, however, the isotopic composition of dew water was not related to the plant species from the surfaces of which the water was collected, which allowed us to exclude guttation as a relevant process during dry-spell periods.

## 3 Materials and Methods

### 3.1 Eddy covariance and meteorological measurements

The Chamau site (47°12′36.8″ N, 8°24′37.6″ E) is an intensively managed temperate grassland (4–6 cuts per year) at 393 m a.s.l., located in a valley bottom in Switzerland. The EC and meteorological measurement station (Fig. A1 in Appendix A) have been operational since 2005. The EC measurement setup consisted of a 3-D sonic anemometer (Gill R3, Gill Instruments Ltd., Lymington, UK), and an open-path Infrared Gas Analyzer (IRGA, Li-7500, Li-Cor, Lincoln, NE, USA). The center of the sonic anemometer axis was at 2.4 m a.g.l. (see Zeeman (2008) for more details). The EC measurements at 20 Hz were processed to 30 min averages using EddyPro Version 7.0.6 (LI-COR, 2017) and following established community guidelines (Aubinet et al. (2012); see also Appendix B) for horizontal wind speed (hereafter $u_{2m}$, in m s$^{-1}$), atmospheric specific humidity (hereafter $q_{a2m}$, in g kg$^{-1}$), dew point temperature (hereafter $T_d$, in °C) , turbulent latent heat flux (hereafter LE in W m$^{-2}$), turbulent sensible heat flux (hereafter H in W m$^{-2}$), and net radiation flux (hereafter Rn in W m$^{-2}$); negative fluxes denote a downward flux, whilst positive values stand for upward fluxes. Evapotranspiration (ET in mm h$^{-1}$) was derived from LE (see Appendix B). Ground heat flux (hereafter G in W m$^{-2}$) was measured at 0.02 m depth with two heat flux plates (HFP01 heat flux sensor, Hukseflux, Delft, The Netherlands).

The meteorological instruments were installed at 2.0 m a.g.l. (see Zeeman et al. (2010) for more details). Measurements were taken every 10 s and then aggregated to 30 min averages for air temperature (hereafter $T_{a2m}$, in °C), relative humidity (hereafter RH, in %) (a shaded, sheltered and ventilated HydroClip S3, Rotronic AG, Bassersdorf, Switzerland), as well as long-wave outgoing and ingoing radiation (hereafter LW$_{out}$ and LW$_{in}$, in W m$^{-2}$; obtained from a ventilated 4-way CNR1 radiometer, Kipp & Zonen B.V., Delft, Netherlands that also provided all-wave net radiation Rn), . The horizontal visibility (in km) was measured every 10 s with a fog sensor (MiniOFS, Optical Sensors Inc., Göteborg, Sweden) and a present weather detector (PWD10, Vaisala Oyj, Helsinki, Finland). The meteorological measurements were processed to 30 min averages for $T_{a2m}$, RH, LW$_{out}$, and to 1 min averages for visibility. The vegetation surface temperature ($T_0$, in °C) was determined following Stefan–Boltzmann's law as (Moene and van Dam, 2014):

$$T_0 = \sqrt[4]{\frac{\mathrm{LW_{surface}}}{\varepsilon \cdot \sigma}} - 273.15 \;, \tag{1}$$

where an emissivity (hereafter $\varepsilon$) of 0.98 was used to calculate temperatures for wet leaf surfaces (hereafter index w; $T_0 = T_{0w}$), and a value of 0.96 was used for dry leaf surfaces (hereafter index d; $T_0 = T_{0d}$) after López et al. (2012); $\sigma$ is Stefan-Boltzmann constant at $5.67 \cdot 10^{-8}$ W m$^{-2}$ K$^{-1}$. The LW$_{surface}$ was derived as suggested by Moene and van Dam (2014) as the difference between measured upwelling long-wave radiation LW$_{out}$ corrected for the first-order reflection of downwelling long-wave radiation LW$_{in}$, i.e., LW$_{surface}$ = LW$_{out}$ − (1 − $\varepsilon$) · LW$_{in}$.

The saturation specific humidity ($q_0$, in g kg$^{-1}$) and the relative humidity ($h_0$) with respect to surface temperature $T_0$ for wet and dry vegetation surfaces was calculated following Tetens formula (Buck, 1981; Campbell and Norman, 1998) (see the equations in Appendix B).

Flux measurements were also used to assess the local surface energy budget as:

$$\text{Rn} = \text{H} + \text{LE} + \text{G} + \Delta Q \,, \tag{2}$$

where $\Delta Q$ is the energy budget closure term remaining when all other components (Rn, H, LE, G; in W m$^{-2}$) are measured. A deviation of $\Delta Q$ from 0 W m$^{-2}$ is typically a result of inaccuracies in determining the components of the energy budget, differences in footprint areas covered by the three different types of measurements (Rn: radiation flux; H and LE: turbulent

fluxes; G: molecular flux), or advection of sensible and latent heat. Here we make the assumption that inaccuracies of the individual measurements do not change substantially over each field campaign, and variations of footprint areas mostly relate to H and LE with smaller footprints during daytime and larger ones at night, whereas advective influences should be best detectable on the hourly timescale during the day/night transition around sunrise and sunset.

### 3.2 Experiment setup during the three 24 h observation campaigns

Three 24 h observation campaigns were carried out during expected dew/fog events on 25–26 July (event 1), 20–21 August (event 2), and 9–10 September (event 3) 2018. The time series were all recorded in CET (UTC+1). The precipitation at the Chamau site was 870 mm in 2018, which was 297 mm (about 25%) less than the multiyear average over 2006–2017. The year-to-date precipitation before the three events was 393 mm, 474 mm, and 536 mm, respectively, which was 311 mm (–44%), 359 mm (–43%), and 367 mm (–41%) less than the corresponding 2006–2017 averages (Fig. 1a). From April to September

2018, the average temperature was 17.3 °C, which was 1.8 °C higher than the corresponding 2006–2017 average (Fig. 1b). The corresponding consecutive rainless periods were 23–27 July, 18–21 August, and 8–12 September 2018 respectively. The daily average ET during the rainless periods was 2.7 mm (Fig. 1c).

Because of the extreme summer drought in 2018, no harvesting of the grassland was carried out during the three campaigns, but harvests were possible 46 d before event 1 on 9 June 2018, and one day after event 3 on 10 September 2018. The leaf area

index was 1.5–2.5 m$^2$ m$^{-2}$ as measured 7 d before events 1 and 2 with LAI-2000 plant canopy analyser (LI-COR Biosciences, Lincoln, NE, USA). The mean vegetation height ($z_c$) was roughly 0.2–0.3 m during the three campaigns. The wilting point, field capacity, and saturation water content (all in volumetric soil water content) were 12–14 %, 27–30 %, and 47–49 %, respectively, according to the soil texture reported by Roth (2006), and the equations by Saxton et al. (1986) (see details in Appendix C). The volumetric soil water content (SWC) was measured at 10 cm, 20 cm, 30 cm, and 50 cm respectively (ML2x

sensors, Delta-T Devices Ltd., Cambridge, UK). The plant roots were mainly distributed in the top 0–15 cm of the soil (Prechsl et al., 2015), and SWC in this layer was 17–20 % during the three events (Fig. 1d). The rainfall after event 1 was not sufficiently to refill the deficient soil water storage, which explains why the observed SWC remained low until event 3.

### 3.2.1 Isotopic composition of non-rainfall water on foliage, and in leaf and soil water

To analyze the isotopic composition of NRW on foliage (hereafter fNRW), leaf water, and soil water, the sampling was carried

out on a grassland area of $100 \times 130$ m$^2$ around the EC & meteorological installations (Fig. A1 in Appendix A). NRW droplets on foliage (fNRW) were absorbed in triplicates with cotton balls from the leaf surfaces of randomly selected plants for *Lolium sp.* with long and narrow leaves; taller vegetation *Taraxacum sp.* with long and wide leaves; shorter vegetation *Trifolium spp.* with short and wide leaves, as well as both shorter and taller vegetation. The fNRW samples were taken at the end of the nights of events 1 and 3 (once sampling per event), but every two hours during the night of event 2 (i.e., four times of sampling in

event 2). Simultaneously, leaf samples were taken in triplicates from the randomly selected plants for the three species after softly drying the leaf surfaces with tissue paper. To prevent the disturbance of destructive sampling on the effect of dew and fog formation, the NRW droplets and leaf samples were taken from different plants of the same species in the sampling area.

The soil cores were taken with a soil auger, and were then cut into slabs to separate the soil depths of 0–5 cm, 5–10 cm, 10–15 cm, 15–20 cm, and 20–40 cm. Soil samples in event 1 were taken without replicate within 2 h before sunset, and at the end of the night; soil samples in event 2 was taken without replicate within 2 h before sunset, as well as every two hours (i.e., four times of sampling in event 2) during the night; soil samples in event 3 were taken in triplicates within 2 h before sunset, and at the end of the night.

After collection, NRW droplets on foliage (fNRW), leaf and soil samples were immediately transferred into gas-tight 12 ml exetainers (Labco Exetainer® vials, High Wycombe, UK), and stored in a portable cooling box filled with ice blocks. Before extracting the water in a cryogenic vacuum extraction system (Prechsl et al., 2015), the samples were stored at –19°C. The isotopic composition of extracted water samples for fNRW (hereafter $\delta^{18}O_{fNRW}$, and $\delta^2H_{fNRW}$), leaf water ($\delta^{18}O_{leaf}$, and $\delta^2H_{leaf}$), and soil water (hereafter $\delta^{18}O_s$, and $\delta^2H_s$) were measured using an isotope ratio mass spectrometer (IRMS, DELTAplusXP, Finnigan MAT, Bremen, Germany). The measured uncertainties of $\delta^{18}O$ and $\delta^2H$ using IRMS are ±0.1‰ and better than ±1.0‰, respectively (Werner and Brand, 2001; Gehre et al., 2004).

### 3.2.2 Isotopic composition of ambient water vapor and non-rainfall water condensed from this vapor

The isotopic composition and the volumetric mixing ratio of ambient water vapor were measured at 0.5–1 Hz using a cavity ring-down laser absorption spectrometer (L2130-i, Picarro Inc., Santa Clara, CA, USA). The L2130-i was placed in a house 200 m away from the EC & meteorological measurements (Fig. A1 in Appendix A). Ambient air was pulled into the instrument through a PTFE intake hose, with an outer diameter of 1/4 inch, and a PTFE-filter inlet (FS-15-100 and TF50, Solberg International Ltd., Itasca, IL, USA) fixed at 6 m a.g.l.. The intake hose was thermally isolated and heated using a resistive heating wire (Raychem 5BTV2-CT, Von Rotz, Kerns, Switzerland) that was wrapped around the entire length of the intake tube to prevent condensation and minimize the response time of the inlet system. An external membrane pump (N022, KNF Neuberger GmbH, Munzingen, Freiburg, Germany) with a flow rate of 9 L min$^{-1}$ was used to maintain turbulent flow (Reynolds number $Re > 2900$) in the tube to minimize memory effects within the inlet system. The isotopic composition of ambient water vapor (hereafter $\delta_a$) and the volumetric ambient water vapor mixing ratio (hereafter $w_a$) were measured using a flow split with a flow rate of 300 mL min$^{-1}$ through the L2130-i cavity. The instrument's response time in this setup was found to be on the order of 10 s in Aemisegger et al. (2012).

To correct for instrument drifts and to normalize the data to the international VSMOW-SLAP scale, the raw data were calibrated using a Standard Delivery Module (SDM; A0101, Picarro Inc., Santa Clara, CA, USA) by performing two-point calibrations every 12 h (Aemisegger et al., 2012) using two liquid standards (standard 1: $\delta^{18}O = -11.43‰$, $\delta^2H = -81.84‰$, $d = 9.64‰$; standard 2: $\delta^{18}O = -40.66‰$, $\delta^2H = -325.67‰$, $d = -0.37‰$ measured by an IRMS). The $\delta^{18}O$ and $\delta^2H$ of the standards thus bracket the range of the measured $\delta^{18}O_a$ and $\delta^2H_a$. Laser spectrometric measurements are known to be affected by a water vapor mixing ratio dependent bias due to spectroscopic effects (absorption peak fitting, and baseline effects). In our study, all measurements were performed at $w_a > 12$ mmol mol$^{-1}$, therefore no mixing ratio dependent isotope bias correction was necessary (see more details in Aemisegger et al. (2012)). The L2130-i was calibrated using a dew point generator (LI-610, Li-Cor Inc., Lincoln, NE, USA) following the procedure by Thurnherr et al. (2020). Calibrated $\delta^{18}O_a$ and $\delta^2H_a$ were then averaged over 30 min intervals. The second-order parameter $d$ of ambient water vapor (hereafter $d_a$) was calculated with the calibrated $\delta^{18}O_a$ and $\delta^2H_a$. The overall random uncertainties of $\delta^{18}O$ and $\delta^2H$ measurements were 0.2‰ and 0.8‰ respectively (for more details about the uncertainty quantification, see Aemisegger et al. (2012)).

To analyze the correlation between $d_a$ and surface humidity, the surface relative humidity (RH$_0$ in %) computed from water vapor mixing ratio $w_a$ and surface saturation specific humidity ($q_{0w}$; see Eq. B1 in Appendix B) was calculated as:

$$RH_0 = \frac{q_{a\_L2130i}}{q_{0w}} = \frac{\frac{w_a \cdot M_v \cdot a}{w_a \cdot M_v \cdot a + (1 - w_a \cdot a) \cdot M_d}}{q_{0w}} \quad , \tag{3}$$

where $M_v = 0.018015$ kg mol$^{-1}$ is mole weight for water vapor, $M_d = 0.028965$ kg mol$^{-1}$ is mole weight for dry air, a is a unit conversion factor ($10^{-3}$ mol mmol$^{-1}$ · $10^3$g kg$^{-1}$), $q_{a\_L2130i}$ is specific humidity (in g kg$^{-1}$) computed from $w_a$.

Ambient water vapor is one source of NRW on foliage (fNRW) which experiences fractionation during the condensation process. With the assumption of equilibrium fractionation, the isotopic composition of equilibrium liquid (hereafter aNRW, and its isotopic composition $\delta^{18}O_{aNRW}$ and $\delta^2H_{aNRW}$) formed from ambient water vapor $\delta^{18}O_a$ and $\delta^2H_a$ was calculated using the temperature-dependent equilibrium fractionation factors following Horita and Wesolowski (1994) as:

$$\delta^{18}O_{aNRW} = \alpha_{18O} \cdot \left(10^3 + \delta^{18}O_a\right) - 10^3 , \tag{4}$$

$$\delta^2H_{aNRW} = \alpha_{2H} \cdot \left(10^3 + \delta H_a\right) - 10^3 . \tag{5}$$

where $\alpha_{18O}$ and $\alpha_{2H}$ were equilibrium fractionation factors calculated as (Horita and Wesolowski, 1994):

$$\alpha_{18O} = \exp\left(0.35041 \cdot \frac{10^6}{(T_{0w}+273.15)^3} - 1.6664 \cdot \frac{10^3}{(T_{0w}+273.15)^2} + \frac{6.7123}{T_{0w}+273.15} - \frac{7.685}{10^3}\right) , \tag{6}$$

$$\alpha_{2H} = \exp\left(1.1588 \frac{(T_{0w}+273.15)^3}{10^9} - 1.6201 \frac{(T_{0w}+273.15)^2}{10^6} + 0.79484 \cdot \frac{(T_{0w}+273.15)}{10^3} - 0.16104 + 2.9992 \cdot \frac{10^6}{(T_{0w}+273.15)^3}\right) . \tag{7}$$

An approach to calculate the NRW isotope composition from ambient vapor, which considers both equilibrium and non-equilibrium fractionation in the laminar sublayer of the leaf boundary layer has been proposed by Wen et al. (2012). The isotope composition of the NRW formed from ambient vapor under such conditions (hereafter naNRW, and its isotopic composition $\delta^{18}O_{naNRW}$ and $\delta^2H_{naNRW}$), was calculated as follows:

$$\delta_{naNRW} = \frac{\delta_a + \epsilon_{eq}/h_0 + (1-h_0)\epsilon_k/h_0}{1 + \epsilon_k/1000 - (\epsilon_{eq}+\epsilon_k)(1/h_0)/1000} , \tag{8}$$

where $\delta_{naNRW}$ is either $\delta^{18}O_{naNRW}$ or $\delta^2H_{naNRW}$, $\epsilon_k$ is the non-equilibrium fractionation factor in permil, calculated from $\epsilon_k = m \cdot (1 - D_i/D_l) \times 1000$ ‰, given $D_i/D_l$ ($^{18}O$) = 0.9723, $D_i/D_l$ ($^2H$) = 0.9755 following Merlivat (1978), and $m = 0.67$ for laminar flow following Dongmann et al. (1974); $\epsilon_{eq}$ is equilibrium fractionation factor in permil calculated from $(\alpha - 1) \times 1000$ ‰ in Eqs. 6 and 7.

### 3.2.3 Determination of the atmospheric layer heights and assessment of eddy covariance setup height

The isotopic fractionation during phase change at the Earth surface is linked to the micrometeorological layers near the surface (Fig. 2). The inclusion of a zero-plane displacement (or fluid dynamic height origin, $z_d$) (Fig. 2) in wind profiles allows us to separate the downward flux from ambient water vapor and the upward flux from soil-diffusing vapor. The average wind speed is zero at $z_d + z_0$, where $z_0$ is aerodynamic roughness length ($z_0$). The roughness length $z_0$ at the Chamau site was 0.03 m on average. It was computed by solving the logarithmic wind profile equation for $z_0$ using measured horizontal wind speed $u_{2m}$ and friction velocity $u_*$,

$$z_0 = \frac{z_{2m} - z_d}{\exp\left(\frac{u_{2m} \cdot \kappa}{u_*}\right)} \tag{9}$$

during neutral atmospheric stratification (e.g., Panofsky (1984); see data in Appendix D), with $z_{2m}$ the measurement height (2 m) and $\kappa$ the von Kármán constant (0.40). The zero-plane displacement $z_d$ can be approximated as two thirds of vegetation height (Stull, 1988; Oke, 2002) = 0.13–0.20 m. With respect to $z_d + z_0$ = 0.16–0.23 m, we consider three pathways of NRW inputs onto the foliage of grasslands for dew and radiation fog: 1) the downward component of dew formation condensing from ambient water vapor, 2) the upward component of dew formation via distillation of water vapor from soil, and 3) radiation fog deposition.

The top of NBL is difficult to quantify, because in many cases the NBL does not have a strong demarcation at its top. Therefore, many definitions of the NBL are based on relative comparisons of the stable boundary layer state aloft to near-surface state (Stull, 1988). We determined the top of the NBL as the lowest height where the vertical stratification of the

atmosphere becomes isothermal, i.e., $\partial T/\partial z = 0$ (Stull, 1988; Garratt, 1992), where $T$ is air temperature extracted from the hourly COSMO-1 model (Consortium for Small-scale Modeling) with a resolution of 1.1 km (meridional) $\times$ 1.1 km (zonal) over Switzerland (Doms et al., 2018; Westerhuis et al., 2020) and 80 vertical levels. During the three events in this study, the NBL top was at 114 m, 55 m, and 193 m a.g.l., respectively (Fig. 3). Therefore, the EC measurement setup at 2.4 m a.g.l. are expected to have captured roughly 98% of the expected flux (Eugster and Merbold, 2015). The roughness sublayer (1–3 times

of the vegetation height according to Oke (2002)) was at 0.2–0.9 m at the Chamau site, therefore the EC instruments were installed well above the roughness sublayer. Here we simply use NBL as a background information on atmospheric stability, but did not use it for NBL budgets (Denmead et al., 1996) as was done by Stieger et al. (2015) at this exact same site, and thus the uncertainty in the exact value extracted for the NBL top from the COSMO-1 model output has no influence on our dew estimates.

**3.2.4 Partitioning of non-rainfall water inputs using a two end-member mixing model**

We partitioned the contribution of NRW input pathways into the two main processes: (1) the downward component of dew formation and fog droplet deposition (aNRW), and (2) the distillation of soil-diffusing vapor on plant leaves. With unsaturated conditions, NRW on foliage (fNRW) was a mix of aNRW and distillation, while with saturated conditions, fNRW was originating from dew or from fog (aNRW), which could lead to a mixture of water from both sources over the course of a night

when dew and fog occur intermittently. "Unsaturated conditions" in this context refers to the standard meteorological measurements at 2 m a.g.l. level. Dew forming in unsaturated conditions is a mixture of aNRW and distillation but lacks contribution from fog deposition. Thus, the isotopic signature of NRW resulting from the isotopic composition of distillation (hereafter $\delta^{18}O_{distillation}$ and $\delta^{2}H_{distillation}$) and the proportion of distillation (hereafter $f_{distillation}$) in fNRW can be expressed as:

$$\delta^{18}O_{fNRW} = f_{distillation} \cdot \delta^{18}O_{distillation} + f_{aNRW} \cdot \delta^{18}O_{aNRW} \ , \tag{10}$$

$$\delta^{2}H_{fNRW} = f_{distillation} \cdot \delta^{2}H_{distillation} + f_{aNRW} \cdot \delta^{2}H_{aNRW} \ , \tag{11}$$

$$1 = f_{distillation} + f_{aNRW} \ , \tag{12}$$

where $f_{aNRW}$ is the proportion of aNRW in fNRW. The four parameters $\delta^{18}O_{distillation}$, $\delta^{2}H_{distillation}$, $f_{distillation}$, and $f_{aNRW}$ are unknown. Therefore, to solve for four unknowns with only three equations (Eqs. 10–12) requires two time points of measurements (here we used 23:00 CET and 1:00 CET in event 2), to obtain empirical estimates for the four unknowns. By

doing so, we implicitly assumed that $\delta^{18}O_{distillation}$ and $\delta^{2}H_{distillation}$ were constant over time (i.e., did not change within this 2 h interval during event 2), and only $f_{distillation}$ and $f_{aNRW}$ were allowed to change between these two sampling times. For $\delta_{fNRW}$, the median value for each sampling was taken, and for $\delta_{aNRW}$ the period between two measurements was computed from 30 min data. Consequently, the three equations (Eqs. 10–12) can be expanded to six equations via the inclusion of two sampling times ($\tau$, and $\tau+1$) as:

$$\delta^{18}O_{fNRW\_\tau} = f_{distillation\_\tau} \cdot \delta^{18}O_{distillation} + f_{aNRW\_\tau} \cdot \delta^{18}O_{aNRW\_\tau} \ , \tag{13}$$

$$\delta^{2}H_{fNRW\_\tau} = f_{distillation\_\tau} \cdot \delta^{2}H_{distillation} + f_{aNRW\_\tau} \cdot \delta^{2}H_{aNRW\_\tau} \ , \tag{14}$$

$$1 = f_{distillation\_\tau} + f_{aNRW\_\tau} \ , \tag{15}$$

$$\delta^{18}O_{fNRW\_\tau+1} = f_{distillation\_\tau+1} \cdot \delta^{18}O_{distillation} + f_{aNRW\_\tau+1} \cdot \delta^{18}O_{aNRW\_\tau+1} \ , \tag{16}$$

$$\delta^{2}H_{fNRW\_\tau+1} = f_{distillation\_\tau+1} \cdot \delta^{2}H_{distillation} + f_{aNRW\_\tau+1} \cdot \delta^{2}H_{aNRW\_\tau+1} \ , \tag{17}$$

$$1 = f_{distillation\_\tau+1} + f_{aNRW\_\tau+1} \ , \tag{18}$$

which can be solved for the six unknowns $\delta^{18}O_{distillation}$, $\delta^{2}H_{distillation}$, $f_{distillation\_\tau}$, $f_{distillation\_\tau+1}$, $f_{aNRW\_\tau}$, and $f_{aNRW\_\tau+1}$ using "limSolve::Solve" function in R (Venables and Ripley, 2002).

### 3.2.5 Partitioning of non-rainfall water inputs using Monteith (1957) approach (M57)

To assess the results from our mixing model by Eqs. 13–18, the partitioning of NRW components was also performed by Monteith (1957) approach (hereafter M57), i.e., partitioning the NRW components from the amount of aNRW and distillation. The amount of NRW from soil diffusing vapor was calculated as follows based on the near-ground vertical temperature gradient:

$$D = K_v \cdot (T_{s1cm} - T_{a1cm}) \cdot \left( \frac{\chi_{s1cm} - \chi_{a1cm}}{T_{s1cm} - T_{a1cm}} \right), \tag{19}$$

where $K_v$ is diffusion coefficient given $2.4 \cdot 10^{-5}$ m$^2$ s$^{-1}$ (Monteith, 1957); $T_{s1cm}$ in °C is the soil temperature measured at 1 cm in depth; $T_{a1cm}$ in °C is the air temperature at 1 cm a.g.l. which was computed from the simulated wet vegetation surface temperature $T_{0w}$, and measured soil temperature $T_{s1cm}$:

$$T_{a1cm} = T_{s1cm} - \frac{1cm}{z_0 + z_d} \cdot (T_{s1cm} - T_{0w}), \tag{20}$$

and the saturated absolute humidity $\chi_{s1cm}$ and $\chi_{a1cm}$ at soil temperature at 1 cm depth ($T_{s1cm}$) and air temperature at 1 cm ($T_{a1cm}$) were calculated following Parish and Putnam (1977) as:

$$\chi = 0.21668 \cdot \frac{6.11 \cdot \exp(\frac{17.502 \cdot T}{T + 240.97})}{T + 273.15} \cdot h, \tag{21}$$

where $T$ is substituted by either $T_{s1cm}$ or $T_{a1cm}$ to calculate $\chi_{s1cm}$ and $\chi_{a1cm}$ with relative humidity $h = 100\%$, respectively.

The condensation rate of ambient water vapor was calculated as (Pasquill, 1949; Monteith, 1957):

$$F = \frac{\kappa^2 \cdot z_{2m} \cdot u \cdot (\frac{\partial \chi}{\partial z})}{\ln(\frac{z}{z_0})} \cdot \Phi, \tag{22}$$

where $\partial \chi / \partial z$ is the gradient of absolute humidity from $T_a$ at $z = 2$ m, and from $T_{0w}$ at $z = z_0 + z_d$, thus $\partial \chi / \partial z = [\chi(T_{a2m}) - \chi(T_{0w})]/[z_{2m} - (z_0 + z_d)]$; $\Phi$ is stability parameter proportional to Richardson number $Ri$ with numerous semi-empirical forms (Garratt, 1992); here we followed Monteith (1957) given $\Phi = 1/(1+10 \cdot Ri)$ in which σ is a proportionality factor associated with thermal stratification assumed to be on the order of 10 (Pasquill, 1949; Monteith, 1957); $Ri$ is calculated as (Wyngaard, 2010):

$$Ri = \frac{z_{2m}/L}{1 + 5 \cdot z_{2m}/L}, \tag{23}$$

where $g = 9.81$ m s$^{-2}$ is gravity acceleration; $L$ in m is Monin-Obukhov length calculated following Monin and Obukhov (1954); other semi-empirical forms of $\Phi$ and its effect on NRW amount estimates were given in Appendix E.

### 3.3 Statistics and imaging

We report means ± SD (standard deviation), unless specified differently. For the isotopic composition of NRW on foliage ($\delta^{18}O_{fNRW}$, $\delta^2H_{fNRW}$, and $d_{fNRW}$) and leaf water ($\delta^{18}O_{leaf}$, $\delta^2H_{leaf}$, and $d_{leaf}$), we report the inter-quartile range (25% and 75% quantile) together with the median to account for the unknown empirical distribution of destructive sampling of individual plants. The statistical significance of among-species differences was assessed with Tukey's honest significant difference test using the "agricolae:: HSD.test" function in R. All analyses were performed with R version 3.6.3 (R Core Team, 2020). Orthogonal regression was performed using the "mcr::mcreg" function in R (total least square, Gat (1981)) for all linear regression analyses.

### 4 Results

### 4.1 Environmental conditions during dew and radiation fog events

Dew and radiation fog generally form during clear-sky nights with low wind speed and weak turbulence. During the three field campaigns presented in this study, wind speed ($u_{2m}$) and latent heat flux (LE) showed an abrupt weakening from around 17:00

CET onwards (Fig. 4a, b). With nightfall, $u_{2m}$ remained below 0.7 m s$^{-1}$ (Fig. 4a), and LE was very low (–26 to 14 W m$^{-2}$; Fig. 4b), indicating a vanishing of turbulent fluxes. These are favorable conditions for dew and radiation fog formation.

The three events with dew or radiation fog were characterized by high relative humidity (RH) with respect to air temperature measured at 2 m a.g.l.. From around 17:00 CET, RH increased rapidly, and reached 100% around 03:00 CET during event 2, and around 20:30 CET during event 3 (Fig. 4c). These saturated conditions led to the formation of fog characterized by a horizontal visibility < 1 km (Fig. 4d). Fog appeared around 05:00 CET during event 2, lasting for less than an hour until sunrise, whilst the onset of fog was much earlier during event 3 (around 23:00 CET), lasting for a longer period until dissipation around sunrise. The visibility was always > 1 km in event 1, indicating that fog was absent during event 1. Therefore, event 1 can be considered as a dew-only event, whilst events 2 and 3 were characterized by a combination of dew and partial influence of radiation fog.

Dew or radiation fog occurred when the surface cooled below dew point. Both grassland surfaces and ambient air started to cool from around 17:00 CET onwards, due to substantial net long-wave radiation loss (–36 W m$^{-2}$ at sunset; Fig. 5a). The vegetation surfaces of the grassland cooled more rapidly than the near-surface atmosphere, thus with nightfall, the vegetation surface temperature $T_0$ derived from radiation measurement remained cooler than air temperature $T_{a2m}$ at 2 m a.g.l., although both gradually decreased (Fig. 5b). The first sign of condensation occurred when the leaf surfaces cooled below dew point temperature (i.e., $T_0 < T_d$; Fig. 5b). The level of computed dry surface temperature $T_{0d}$ became lower than dew point $T_d$ at around 0:30 CET in event 1, 21:30 CET in event 2, and 19:00 CET in event 3 (Fig. 5b), determining the time when the first signs of condensation can be expected. During event 3, the surface already cooled below the dew point rapidly after sunset (i.e., $T_0 < T_d$ in Fig. 5b), indicating that condensation already started with nightfall.

Dew and radiation fog were characterized by a decrease in specific humidity (Fig. 5c). But before the formation of dew and fog set in, the specific humidity of the air ($q_{a2m}$) steeply increased by 2.0–3.5 g kg$^{-1}$ from around 17:00 CET until sunset (Fig. 5c), suggesting the mixing of moisture from local evapotranspiration into a shallow inversion layer. With nightfall, $q_{a2m}$ reached a nighttime maximum of 9.6–12.5 g kg$^{-1}$ (Fig. 5c). Especially, in events 1 and 2, before starting to decrease, $q_{a2m}$ fluctuated for a short period from sunset until the first sign of condensation (Fig. 5c). When condensation started ($T_0 < T_d$, Fig. 5b), $q_{a2m}$ gradually decreased (Fig. 5c). With the saturation specific humidity at surface temperature ($q_0$) falling to values below $q_{a2m}$ (Fig. 5c), computed theoretical surface relative humidity $h_0$ exceeded 100% (Fig. 4c). The decrease of $q_{a2m}$ was much faster in event 3 (0.4 g kg$^{-1}$ h$^{-1}$; Fig. 5c) than that in events 1 and 2 (0.2 and 0.3 g kg$^{-1}$ h$^{-1}$; Fig. 5c), indicating stronger condensation of ambient water vapor.

According to the variability of environmental conditions, water vapor and thermal dynamics of dew and radiation fog events can be separated into four periods from 17:00 CET until sunrise: 1) pre-condensation period (hereafter P1 period) with the gradual weakening of turbulence, and warmer surface above the dew point ($T_0 > T_d$; Fig. 5b); and 2) condensation period (hereafter P2 period) with cooler surface below dew point ($T_0 < T_d$; Fig. 5b). The P1 period was further separated into: P1a period starting around 17:00 CET until sunset with the weakening of turbulence and the increase of specific humidity $q_{a2m}$; and P1b period from sunset until the first sign of condensation with short-term fluctuations of specific humidity ($q_{a2m}$). The P2b period was further split into: P2a period with dew-only in the conditions of RH < 100%; and P2b period with combined dew and radiation fog in the conditions of RH = 100%.

## 4.2 Isotopic dynamics of ambient water vapor during dew and fog events

The four periods of water vapor and thermal dynamics defined in Sect. 4.1 are reflected in the temporal evolution of volumetric water vapor mixing ratio ($w_a$) and isotopic composition of ambient water vapor ($\delta^{18}O_a$, $\delta^2H_a$, and $d_a$; Fig. 6). From 17:00 CET until sunset (P1a period), when the turbulence was weakening and the surface was cooling, $w_a$, $\delta^{18}O_a$, and $\delta^2H_a$ showed a steep increase by 0.3–0.4 mmol mol$^{-1}$, 2.0–3.2‰, and 7.4–12.5‰, respectively (Fig. 6a, b, c), whilst $d_a$ showed a steep decrease by 11.6–16.9‰ (Fig. 6d). The decrease in $d_a$, and increase in $\delta^{18}O_a$ and $\delta^2H_a$ was due to the effect of local evapotranspiration

under the conditions of reduced entrainment from the free troposphere. The vapor sourced from local evapotranspiration features a lower $d_a$, and higher $\delta^{18}O_a$ and $\delta^2H_a$ than the free troposphere. With nightfall, $w_a$, $\delta^{18}O_a$, and $\delta^2H_a$ reached a plateau with 15.5 to 17.8 mmol mol$^{-1}$ in $w_a$ (Fig. 6a), $-15.5$ to $-14.3$‰ in $\delta^{18}O_a$ (Fig. 6b), and $-128.0$‰ to $-113.2$‰ in $\delta^2H_a$ (Fig. 6c).

The start of condensation then caused a decrease of $\delta^{18}O_a$ and $\delta^2H_a$, because heavier water isotopologues have a lower partial vapor pressure at saturation than their lighter counterpart ($p_{sat}[^1H^2H^{16}O] < p_{sat}[^1H_2^{18}O] < p_{sat}[^1H_2^{16}O]$), and thus preferentially prevail in the phase with stronger bonds (liquid > vapor; Bigeleisen (1961)) . During the condensation period with RH < 100% (P2a period), $w_a$ steeply decreased by 0.8–5.5 mmol mol$^{-1}$ (Fig. 6a), $\delta^2H_a$ decreased by 3.3–16.7‰ (Fig. 6c), and $d_a$ reached its minimum at $-11.8$‰ to $-4.7$‰ (Fig. 6d). During the condensation period with RH = 100% (P2b period), the decreasing rate of $\delta^2H_a$ in event 3 (1.6‰ $\delta^2H_a$ h$^{-1}$) was almost double compared to that in events 1 and 2 (0.8 and 1.0 $\delta^2H_a$ h$^{-1}$ respectively, Fig. 6c), suggesting stronger condensation in event 3. Note that the changes of $\delta^{18}O_a$ and $d_a$ (Fig. 6b, d) depended on the humidity dynamics and the occurrence of dew and fog (Fig. 4c, d). During the dew-only periods (P2a period) in events 1 and 2 (Fig. 4d), $\delta^2H_a$ decreased by 3.3–5.7‰ (Fig. 6c), and $d_a$ slightly decreased by 3.4–3.7‰ (Fig. 6d), while $\delta^{18}O_a$ showed fluctuations around the maximum reached 4 h and 2 h after nightfall of events 1 and 2 respectively ($-15.5$‰ to $-14.3$‰, Fig. 6b). The slight fluctuation of $\delta^{18}O_a$, and decrease of $d_a$ during P2a period was a result of concurrent evaporation, which leads to an additional non-equilibrium fractionation with variations of $\delta^{18}O_a$: $\delta^2H_a$ deviating from 1:8. Furthermore, as condensation was stronger than evaporation (i.e., net condensation), this caused a decrease of $w_a$ and $\delta^2H_a$ (Fig. 6a, c). Because $\delta^{18}O_a$ is more sensitive to evaporation than $\delta^2H_a$ due to the higher partial vapor pressure of $^1H_2^{18}O$ than $^1H_2H^{16}O$, evaporation accompanying condensation is the likely reason for the fluctuations of $\delta^{18}O_a$ (Fig. 6b), but had only a minor effect on the variability of $\delta^2H_a$ (Fig. 6c). During P2b periods in events 2 and 3 with dew and fog in combination, both $\delta^{18}O_a$ and $\delta^2H_a$ gradually decreased (by 0.3–1.5‰, and 2.1–12.8‰ respectively) with a ratio of around 1:8 (Fig. 6b, c), hence $d_a$ was relatively constant during the nighttime minimum ($-6.0$‰ to $-4.7$‰, Fig. 6d) with only small fluctuations. In this saturated condition, evaporation was negligible, and condensation was the dominant process. This is confirmed by the constant values of $d_a$ during P2b (Fig. 6d) showing that this period was dominated by equilibrium fractionation.

### 4.3 Isotopic composition of different non-rainfall water components

As one of the components of NRW on foliage (fNRW), the isotopic composition of NRW equilibrium liquid from ambient water vapor (aNRW) was comparable with the isotopic composition of fNRW. The isotopic composition of aNRW was $-5.0$‰ to $-4.3$‰ for $\delta^{18}O_{aNRW}$, $-47.4$‰ to $-38.6$‰ for $\delta^2H_{aNRW}$, and $-12.1$‰ to $-2.4$‰ for $d_{aNRW}$ (Fig. 7a, b, c). For comparison, NRW on foliage (fNRW) was $-6.1$‰ to $-1.5$‰ for $\delta^{18}O_{fNRW}$, $-64.3$‰ to $-35.6$‰ for $\delta^2H_{fNRW}$, and $-33.8$‰ to 8.0‰ for $d_{fNRW}$ (Fig. 7a, b, c). The isotopic composition of fNRW varied over time with gradually decreasing $\delta^{18}O_{fNRW}$ (Fig. 7a), but gradually increasing $\delta^2H_{fNRW}$ (Fig. 7b) and $d_{fNRW}$ (Fig. 7c). The relationship between the isotopic composition of fNRW and aNRW was related to humidity conditions. With unsaturated conditions when dew formation occurred, $\delta^{18}O_{aNRW}$ ($-4.4\pm0.1$‰; Fig. 7a) was lower than $\delta^{18}O_{fNRW}$ ($-3.8$‰; Fig. 7a), while $\delta^2H_{aNRW}$ ($-42.3\pm3.8$‰; Fig. 7b) was higher than $\delta^2H_{fNRW}$ ($-47.7$‰; Fig. 7b), and $d_{aNRW}$ ($-7.1\pm3.6$‰; Fig. 7c) was higher than $d_{fNRW}$ ($-20.5$‰; Fig. 7c). With saturated conditions at 3:00 and 5:00 CET of event 2, the isotopic composition of aNRW ($-4.6\pm0.8$‰ in $\delta^{18}O_{aNRW}$, $-41.8\pm3.4$‰ for $\delta^2H_{aNRW}$, and $-5.4\pm5.9$‰ for $d_{aNRW}$; Fig. 7) was identical to the isotopic composition of fNRW ($-4.7$‰ for $\delta^{18}O_{fNRW}$, $-43.0$‰ for $\delta^2H_{fNRW}$, and $-5.4$‰ for $d_{fNRW}$; Fig. 7). Especially, with saturated condition at 5:00 CET in event 3 when radiation fog occurred, $\delta^{18}O_{fNRW}$ and $\delta^2H_{fNRW}$ were lowered by 0.7‰ and 1.4‰ with respect to $\delta^{18}O_{aNRW}$ and $\delta^2H_{aNRW}$ respectively (Fig. 7a, b), and $d_{aNRW}$ was 5.5‰ higher than $d_{fNRW}$ (Fig. 7c).

The isotopic composition of the distillation component, i.e., NRW from soil-diffusing vapor, was computed with a two end-member mixing model using the values from 23:00 to 1:00 CET in event 2. In unsaturated conditions, with respect to aNRW, $\delta^{18}O_{fNRW}$ and $\delta^2H_{fNRW}$ deviated to the higher and lower sides of $\delta^{18}O_{aNRW}$ and $\delta^2H_{aNRW}$, respectively (Fig. 7a, b). This is in contrast to the effect that evaporation would have had, then both $\delta^{18}O_{fNRW}$ and $\delta^2H_{fNRW}$ would be higher than $\delta^{18}O_{aNRW}$

and $\delta^2H_{aNRW}$. Therefore, we assumed that the observed deviations of $\delta^{18}O_{fNRW}$ and $\delta^2H_{fNRW}$ with respect to $\delta^{18}O_{aNRW}$ and

$\delta^2H_{aNRW}$ were caused by the mixed source of NRW on foliage, i.e., the mixing of NRW from ambient water vapor and soil-

diffusing vapor (i.e., distillation). Based on the measurements from 23:00 to 1:00 CET in event 2, the averages of $\delta^{18}O_{distillation}$,

$\delta^2H_{distillation}$, and $d_{distillation}$ during this 2 h period were computed as –1.0‰, –71.8‰, and –63.4‰ respectively (Fig. 7) via the

mixing model. As a comparison, from 1 h before sunset till sunrise in event 2, the isotopic composition of soil water in 0–40

cm varied in the range of –10.4‰ to 5.5‰ for $\delta^{18}O_s$, –78.8‰ to –8.5‰ for $\delta^2H_s$, and –52.4‰ to 4.1‰ for $d_s$ (Fig. 8). The

computed distillation $\delta^{18}O_{distillation}$ and $\delta^2H_{distillation}$ fell in the range of the soil water $\delta^{18}O_s$ and $\delta^2H_s$ (Fig. 8a, b), whilst $d_{distillation}$

was lower than the soil water $d_s$ (Fig. 8c) probably derived from the uncertainty of $\delta^{18}O_{distillation}$ and $\delta^2H_{distillation}$ estimates (see

in Sect. 5.3).

    The relationships of $\delta^2H_{fNRW}$–$\delta^{18}O_{fNRW}$ and $\delta^2H_{aNRW}$–$\delta^{18}O_{aNRW}$ with respect to the local meteoric water line (LMWL: $\delta^2H$

$= 7.68 \times \delta^{18}O + 6.97$, Prechsl et al. (2014)) suggested that the local vapor is the primary source for dew and radiation fog

during all three events (Fig. 9). Both $\delta^2H_{fNRW}$–$\delta^{18}O_{fNRW}$ and $\delta^2H_{aNRW}$–$\delta^{18}O_{aNRW}$ fell to the right-hand sides of the LMWL,

suggesting lower $d$ from NRW inputs as compared to local precipitation. When we only considered the condensation of ambient

water vapor under equilibrium fractionation, $\delta^2H_{fNRW}$ and $\delta^{18}O_{fNRW}$ pairs fell on the $\delta^2H_{aNRW}$–$\delta^{18}O_{aNRW}$ regression line (for the

sampling at 3:00 and 5:00 CET in event 2; Fig. 9). However, with the mix of the component condensing from soil-diffusing

vapor (distillation) in the conditions of RH < 100%, the $\delta^2H_{fNRW}$–$\delta^{18}O_{fNRW}$ pairs fell to the right-hand sides of the $\delta^2H_{aNRW}$–

$\delta^{18}O_{aNRW}$ regression line (for the sampling at 3:00 CET in event 1, and the samplings at 23:00 and 1:00 in event 2; Fig. 9), and

$d_{fNRW}$ was lower than $d_{aNRW}$ (–13.5±9.7‰ for $d_{fNRW}$, and –6.4±2.9 ‰ for $d_{aNRW}$; Fig. 7c). This suggested that the soil-diffusing

vapor was a lower $d$ vapor source as compared to the ambient water vapor $d_{aNRW}$ (Fig. 9), which corresponded to the fact that

the soil water $d_s$ (–11.0±14.0 ‰, Fig. 8c) was lower than $d_{aNRW}$. Whereas, with the mix of the component from radiation fog

deposition in the conditions of RH = 100 %, $\delta^2H_{fNRW}$–$\delta^{18}O_{fNRW}$ pairs fell to the left-hand sides of the $\delta^2H_{aNRW}$–$\delta^{18}O_{aNRW}$

regression line (for the sampling at 5:00 CET in event 3), hence the corresponding $d_{fNRW}$ was higher than $d_{aNRW}$ (Fig. 9).

The condensation of ambient water vapor for dew formation can be approximated as an equilibrium fractionation process

(e.g., Wen et al. (2012), and Delattre et al. (2015)); the condensation of ambient water vapor to form radiation fog can cause

lower $\delta^{18}O$ and $\delta^2H$ of NRW on foliage compared to NRW equilibrium liquid obtained from ambient water vapor. When

considering non-equilibrium fractionation, the isotopic composition of NRW from ambient water vapor ($\delta^{18}O_{naNRW}$ and

$\delta^2H_{naNRW}$; Fig. 7a, b) was much lower than the isotopic composition of NRW on foliage ($\delta^{18}O_{fNRW}$ and $\delta^2H_{fNRW}$; Fig. 7a, b),

and the lowering of $\delta^{18}O_{naNRW}$ and $\delta^2H_{naNRW}$ was more severe with the increase of the computed relative humidity ($h_0$; Fig. 4c)

at surface temperature. The lowering of $\delta^{18}O_{naNRW}$ and $\delta^2H_{naNRW}$ with respect to $\delta^{18}O_{fNRW}$ and $\delta^2H_{fNRW}$ was most likely due to

the overestimate of the non-equilibrium fractionation factor when computed $h_0$ exceeded 100% (going up to 132%; Fig. 4c).

Non-equilibrium fractionation is usually considered to be negligible above –10 °C in the process of vapor condensing to liquid

in clouds (Jouzel et al., 1987). However, non-equilibrium fractionation driven by molecular diffusion might have played an

important role in a laminar fog boundary layer (FBL) (Castillo and Rosner, 1989; Epstein et al., 1992), which led to lower

$\delta^{18}O_{fNRW}$ and $\delta^2H_{fNRW}$ than $\delta^{18}O_{aNRW}$ and $\delta^2H_{aNRW}$ at 5:00 CET in event 3 (Fig. 7a, b) when radiation fog occurred (Fig. 4d).

Heavier isotopologues move more slowly than their lighter counterpart in air (molecular diffusivity: $D[^1H_2^{18}O] < D[^1H^2H^{16}O]$

$< D[^1H_2^{16}O]$, Merlivat (1978)), hence the rate at which heavy isotopologues ($^1H_2^{18}O$ and $^1H^2H^{16}O$) in ambient air pass through

the laminar FBL to be condensed at the liquid–vapor interface is smaller than the rate of condensation of their lighter

counterpart ($^1H_2^{16}O$). Therefore, $\delta^{18}O_{fNRW}$ and $\delta^2H_{fNRW}$ can become lower than $\delta^{18}O_{aNRW}$ and $\delta^2H_{aNRW}$. Fog lasted from 23:00

CET until sunrise of event 3, and appeared around 5:00 CET within half an hour before sunrise in event 2 (Fig. 4d). However,

we only observed a lower $\delta^{18}O_{fNRW}$ and $\delta^2H_{fNRW}$ than $\delta^{18}O_{aNRW}$ and $\delta^2H_{aNRW}$ in event 3 (Fig. 7a, b), suggesting that the lowering

of $\delta^{18}O_{fNRW}$ and $\delta^2H_{fNRW}$ might also be related to the duration of radiation fog.

**4.4 Contribution of distillation in the total non-rainfall water on foliage**

The contribution of distillation in the total NRW on foliage (fNRW) was computed via the mixing model using the values from 23:00 to 1:00 CET in event 2, and M57 approach, respectively (Fig. 10). Based on this model we estimated a contribution of 28% and 9% of foliage NRW (fNRW) from distillation at 23:00 CET and 1:00 CET of event 2, respectively (Fig. 10b; Table 1), hence 72% and 91%, respectively, was dew condensed from ambient water vapor. A linear extrapolation of $f_{distillation}$ to the beginning of dew formation at 21:30 CET of event 2 increased the contribution of distillation to 42% (Fig. 10b; Table 1), and thus the contribution of aNRW was 58%. Similarly, when using the values of $\delta^{18}O_{distillation}$ and $\delta^2H_{distillation}$ computed from event 2 for estimating $f_{distillation}$ during event 1, the contribution of distillation was around 18–31%, and thus the contribution of aNRW was around 69–82 % for our sampling at 3:00 CET of event 1 (vertical whiskers in Fig. 10b; Table 1). For comparison, the contribution of distillation was also calculated using M57 approach (Eqs. 19 and 22; Fig. 10a). The dew and radiation fog potentially produced 0.17–0.54 mm d$^{-1}$ NRW gain on foliage, which, compared to evapotranspiration water loss of on average 2.7 mm d$^{-1}$, constitutes a non-negligible water flux into the canopy. The computed dew water gain from aNRW (0.12–0.50 mm) was generally larger than the internal redistribution via distillation (0.04–0.05 mm) (Fig. 10a). As the nights progressed, the contribution of distillation ($f_{distillation}$) to NRW on foliage (fNRW) decreased from 76% at 0:30 CET to 27% before dawn in event 1, and from 45% at 21:30 CET to 19% before dawn in event 2. Overall lower $f_{distillation}$ were observed in events 2 and 3 as compared to that of event 1. No clear trend was observed for $f_{distillation}$ in event 3 with slight variations around 6–8 % (Fig. 10b). The $f_{distillation}$ estimate from the mixing model during events 1 and 2 agree well with the M57-approach (compare black and grey data in Fig. 10b).

**5 Discussion**

**5.1 Diurnal patterns of isotopic composition in ambient water vapor**

The diurnal patterns of $d$ for ambient water vapor were mainly affected by the entrainment from the free troposphere and local evapotranspiration (Lee et al., 2006; Lai and Ehleringer, 2011; Welp et al., 2012; Huang and Wen, 2014; Delattre et al., 2015; Parkes et al., 2017). Moreover, the effect of local evapotranspiration might be enhanced by density-driven katabatic drainage flow down the slopes of the local topography (Drobinski et al., 2003; Whiteman et al., 2010; Nadeau et al., 2013; Duine et al., 2016), and by the regional thermodynamic conditions with weak large-scale influence during clear and calm nights (Eugster and Siegrist, 2000; Goulden et al., 2006; Eugster and Merbold, 2015). Entrainment from the free troposphere played a dominant role in midday atmospheric water vapor dynamics, whilst local evapotranspiration was the main driver of atmospheric water vapor dynamics in the late afternoon when entrainment from free troposphere was already reduced. Entrainment from the free troposphere is a vapor source with lower $\delta^{18}O$ and $\delta^2H$, and higher $d$, whilst local evapotranspiration is a vapor source with higher $\delta^{18}O$ and $\delta^2H$, and lower $d$ (Parkes et al., 2017). Consequently, as compared to the nighttime periods, we observed a higher $d_a$ (Fig. 6d), and a decrease in $\delta^{18}O_a$ and $\delta^2H_a$ (Fig. 6b, c) during 13:00–17:00 CET. Although evapotranspiration is stronger at midday as compared to late afternoon, evapotranspiration is not the main factor controlling $\delta^{18}O_a$ and $\delta^2H_a$ variabilities at midday. On the contrary, during the periods of turbulence weakening and surface cooling from around 17:00 CET to sunset with the reduced entrainment from the free troposphere (weakened $u_{2m}$, and reduced LE in Fig. 4a, b), local evapotranspiration became the main driver of isotopic dynamics of ambient water vapor for the three events in our study. This combination of weakening entrainment and evapotranspiration into a shallower mixed layer hence caused a steep decrease in $d_a$ (Fig. 6d), and increases in $\delta^{18}O_a$ and $\delta^2H_a$ (Fig. 6b, c) during P1a period, which is in accordance with previous studies by Huang and Wen (2014) and Parkes et al. (2017). The soil moisture at 0–5 cm in a short period before sunset (e.g., within 1 h before sunset) showed extremely varied isotopic composition from –8.5‰ to 5.9‰ for $\delta^{18}O_{s\_0–5cm}$, from –72.8‰ to –8.5‰ for $\delta^2H_{s\_0–5cm}$, and from –52.4‰ to –4.1‰ for $d_{s\_0–5cm}$ (Fig. 8), which is in accordance with the report by Welp et al. (2012) that soil evaporation showed very large variability of isotopic signals. The chamber experiment by Parkes et al. (2017) showed that

the soil water vapor at the evaporation front had much higher $\delta^{18}O$ and $\delta^2H$, and much lower $d$ as compared to the soil water at 0–5 cm. This much higher $\delta^{18}O$ and $\delta^2H$, and lower $d$ vapor source at the soil evaporation front might have caused an enhanced variability observed in $\delta^{18}O_a$, $\delta^2H_a$, and $d_a$ in ambient water vapor in P1a period. As the Chamau site studied here is located in a valley bottom, the relative energy budget closure $\Delta Q/Rn$ differed slightly from zero (Fig. 11e) in period P1a, suggesting that the effect of local evapotranspiration on the isotopic dynamics of ambient water vapor might have been sightly accompanied by cold-air drainage towards the valley bottom. Non-equilibrium fractionation is intrinsically dominant in the processes of evaporation with unsaturated ambient air (RH < 100 % at 2 m a.g.l.), which induced a slight decrease of $d_a$ during the condensation period P2a with RH < 100 % (Fig. 6d). During the dew and radiation fog period P2b with RH = 100% at 2 m a.g.l., the condensation of ambient water vapor could essentially be described by an equilibrium fractionation process, with $d_a$ remaining constant at a low nighttime minimum level (Fig. 6d), which is in accordance with the results by Huang and Wen (2014) and Delattre et al. (2015).

Isotopic signals in ambient water vapor provide information on the strength of continental moisture recycling (Aemisegger et al., 2014). In particular, the $d_a$ has been shown to be a useful tracer for moisture source conditions and to be strongly anticorrelated with the surface relative humidity $RH_0$ (computed from $w_a$ using Eq. 3) at the moisture source location (Craig and Gordon, 1965; Pfahl and Wernli, 2008; Welp et al., 2012; Aemisegger et al., 2014). The physical foundation for this strong link is the sensitivity of $d_a$ to the non-equilibrium fractionation effect. The lower surface relative humidity ($RH_0$), the stronger non-equilibrium fractionation, and the higher $d_a$ becomes. Spiegel et al. (2012) found an exceptionally high $d$ in fog droplets after the passage of a cold front in Central Europe with important moisture advected from the subpolar North Atlantic with anomalously low $RH_0$. In Aemisegger et al. (2014), synoptic events were classified into events with remote or local moisture source based on backward trajectories and a detailed correlation analysis between $d_a$ and surface relative humidity. They found that events dominated by local sources show a strong anticorrelation between $d_a$ and local surface relative humidity. In our study, $d_a$ shows a strong anticorrelation with $RH_0$ ($r = -0.94$; Fig. 12), suggesting that dew and radiation fog was formed from local moisture as a vapor source. The slope of the $d_a$–$RH_0$ relation found here (–0.26‰ %$^{-1}$) is similar to the relations found at another Swiss grassland site in dry summer periods (–0.17‰ %$^{-1}$ by Aemisegger et al. (2014)). From this analysis, we conclude that during the studied events, the isotopic signals were dominated by local moisture and that large-scale advection with the weak synoptic-scale flow in the context of central European anticyclones likely had a negligible influence.

**5.2 Processes affecting non-rainfall water on foliage**

Besides the main contribution of NRW from ambient water vapor to dew formation and radiation fog deposition, NRW on foliage (fNRW) can also be affected by three additional processes: 1) re-evaporation of NRW on foliage (He and Richards, 2015), 2) distillation (Monteith, 1957), and 3) guttation (Hughes and Brimblecombe, 1994; Xu et al., 2019). The role of distillation was quantified in Sect. 4.4, and in the following we argue why the other two additional processes at most had a minor influence on dew formation during all three events investigated here. Re-evaporation should have caused both $\delta^{18}O_{fNRW}$ and $\delta^2H_{fNRW}$ being higher than $\delta^{18}O_{aNRW}$ and $\delta^2H_{aNRW}$ (e.g., He and Richards (2015)), which was not the case in our study: we observed higher $\delta^{18}O_{fNRW}$ but lower $\delta^2H_{fNRW}$ as compared to $\delta^{18}O_{aNRW}$ and $\delta^2H_{aNRW}$ (Fig. 7a, b). Re-evaporation of NRW droplets on foliage might have occurred but was not the dominant process that could have led to the observed isotopic differences between fNRW and aNRW. Guttation, the exudation at leaf edges, is a process without a phase change of liquid water and thus does not involve isotopic fractionation. Hence, $\delta^{18}O$ and $\delta^2H$ of guttation water should be identical to $\delta^{18}O$ and $\delta^2H$ of leaf water. In our study, we found significant among-species differences in $\delta^{18}O$ and $\delta^2H$ of leaf water (Table 2), most likely resulting from species-specific leaf water evaporation and root water uptake, which contrasts with the insignificance of among-species differences in $\delta^{18}O$ and $\delta^2H$ of fNRW. This suggests that plant water only has a minor effect on $\delta^{18}O$ and $\delta^2H$ of fNRW. Furthermore, when the soil water content is much lower than field capacity, as was the case during all three events studied here, guttation hardly occurs (Long, 1955). During all three events SWC was very low (17–20 %) and thus close to the

wilting point (12–14 %), and much lower than field capacity (27–30 %) in the main rooting zone in 0–15 cm soil depth. From these considerations, we conclude that re-evaporation and guttation are of no concern at our site during dry spells, and only distillation constitutes an important component for NRW on foliage besides the dominant NRW from ambient water vapor during the events in our study.

### 5.3 Uncertainty assessment of partitioning non-rainfall water components

The uncertainty of partitioning non-rainfall water components arises from the difficulties of measuring or calculating the distillation amount, although the NRW amount from ambient water vapor can be easily and accurately measured by a hydrometric approach, e.g., using a lysimeter (Jacobs et al., 2006). Distillation is an internal re-cycling of water from soil to plant surfaces (Monteith, 1957), which cannot be captured by a lysimeter because the latter device measures the water budget of plant and soil monoliths (Agam and Berliner, 2006), and thus does not distinguish between water in the soil and water on plant leaves. The EC method is widely used to investigate the water flux dynamics in ecosystem, but its suitability for quantitative NRW estimates can be questioned when an open-path IRGA is used to measure LE in clear and calm nights with dew and radiation fog occurrences. As soon as fog occurs or dew drips to the optical windows of the IRGA, LE measurements become unrealistic and cannot be analyzed quantitatively. The use of a closed-path IRGA that does not suffer from this problem may be a solution but could not be tested at the Chamau site in this study. But even when LE measurements appear to be of high quality, the EC-derived NRW estimates from ambient water vapor, may not be very accurate, as shown by Jacobs et al. (2006) that the EC approach obtained less than one third of NRW amount as compared to NRW amount by lysimeter. Monteith (1957) gave the equations of calculating distillation amount, but a reanalysis of the data he published revealed that only the order of magnitude of distillation (reported as 1–2 mg cm$^{-2}$ h$^{-1}$, which corresponds with 0.01–0.02 mm h$^{-1}$) agreed reasonably with observations, and large uncertainties remained, most likely as a result of untestable assumptions that have to be made about molecular transfer, linear temperature gradient, and saturated vapor at the soil surface for the M57 method as shown in Eq. 19 to be valid (Monteith, 1957).

To overcome this problem, we used a two end-member mixing model as an alternative approach to partition NRW components. We compared the results of partitioning NRW components by our mixing model with estimates computed using M57 approach. In general, with our mixing model we obtain lower values of $f_{distillation}$ in the second half of condensation periods (i.e., 3:00 CET of event 1, and 1:00 CET of event 2; Fig. 10b) in comparison with the results obtained from M57 approach. Especially, in the second half condensation periods of events 2 and 3, when the ambient air reached saturation, the mixing model was not applicable for partitioning NRW into aNRW and distillation fractions because $\delta^{18}O$ and $\delta^2H$ were too similar between aNRW and fNRW. Under these conditions the NRW amount approach yielded 6% of $f_{distillation}$ at minimum. This could be explained by the isotopic exchanges in the soil–plant–atmosphere continuum that attenuated the $\delta^{18}O$ and $\delta^2H$ differences among different water sources driven by molecular diffusivity (Farquhar and Lloyd, 1993; Dawson et al., 2002). During the nights of events 1 and 2, we observed a very large variability of soil water $\delta^{18}O$ and $\delta^2H$ (Fig. 8), suggesting that distillation is not a vapor source with constant $\delta^{18}O$ and $\delta^2H$. Furthermore, the spatial variability of shallow soil water content and its isotopic composition might enlarge the variability of $\delta^{18}O$ and $\delta^2H$ for distillation. This might explain why $f_{distillation}$ obtained from the mixing model differed more substantially from the M57 approach at 1:00 CET of event 2 than that at 23:00 of event 2. One reason might be the shortcoming that we had to assume constant values of $\delta^{18}O$ and $\delta^2H$ of distillation for estimating $f_{distillation}$ during 23:00 and 1:00 CET of event 2. Therefore, we recommend more intensive sampling of NRW on foliage in future studies, e.g., every 30 minutes for $\delta^{18}O_{fNRW}$ and $\delta^2H_{fNRW}$ in Eqs. 13–18. This should help to improve the accuracy of $f_{distillation}$ estimate by the mixing model. Another reason might be that the NRW droplets we took from the different leaves represent the cumulated NRW, while the temporal variability of NRW droplets on foliage might enlarge the uncertainty of NRW partitioning.

The M57 approach was not accurate enough for computing $f_{distillation}$ during the events studied here, because air temperature at 1 cm a.g.l. ($T_{a1cm}$) had to be computed from soil and surface temperatures, as it was not directly measured. Consequently,

the large uncertainty in our $T_{a1cm}$ estimates translates to increased uncertainty in distillation estimates computed via Eq. 19. Thus, we have to assume that our calculated distillation rates using M57 approach are even more than 32% off the levels that Monteith (1957) measured via his filter paper sampling. In the M57 approach as shown in Eq. 22, the stability term $\Phi=1/(1+10 \cdot Ri)$ was used. However, the stability term is sometimes written as $\Phi = [1 - 16 \cdot (z - z_d)/L]^{-0.5} = [1 - 16 \cdot Ri]^{-0.5}$ for $Ri < -0.1$, and $\Phi = [1 + 5 \cdot (z - z_d)/L] = [1 - 5 \cdot Ri]^{-1}$ for $-0.1 \leq Ri \leq 1$ as e.g. in Monteith and Unsworth (2013), which would cause higher condensation rate using Eq. 22 (see Fig. E1 in Appendix E), hence lower relative contribution of distillation in the total NRW than given the term $\Phi=1/(1+10 \cdot Ri)$. In future research, we recommend combining isotopic composition measurements with lysimetric measurements to partition NRW from ambient water vapor and distillation. This would provide a useful benchmark to better evaluate the isotope-based estimates of NRW inputs. The NRW amount from ambient water vapor can be measured directly by a lysimeter as the net water gain of the soil and plant monoliths (Kaseke et al., 2012; Ucles et al., 2013), while distillation is an indirect estimate based on stable water isotope data of the transfer of moisture from one part of the surface (soil surface) to another (foliage) within grassland ecosystems.

### 5.4 **Potential effects of non-rainfall water on local water cycling**

From the perspective of ecological relevance, distillation might be more important than previously thought, although it has no large-scale hydrological significance as it is simply a moisture transfer from the soil to the atmosphere (Monteith, 1957), and was thought to be detrimental by enlarging soil water loss via providing a short-cut for the water transfer (Chaney, 1981). The ecological relevance of distillation can be expected if the transfer of moisture is from one hydrological pool that is inaccessible to plants to another that is actually accessible to plants. For example, distillation could transfer soil-diffusing vapor from layers deeper than the effective rooting zone of grassland to droplets forming or depositing on leaf surfaces or surface soil where it can be accessed by the fine roots. Wang et al. (2017) observed that 0.0092 mm of water was transferred from deeper soil layers to the surface by vapor diffusion in a grassland plot, although it was debated whether the water went onto foliage or was absorbed by the topsoil. The process of vapor diffusion from deeper soil layers to the surface strongly depends on soil properties, and thus might differ from site to site. For event 2 in our study, the nighttime variability of soil water $\delta^{18}O$ and $\delta^2H$ was not only observed in the top 0–15 cm, but also in the deeper soil layers at 15–40 cm (Fig. 8), suggesting that isotopic exchange occurs between deeper soil and topsoil layers. Furthermore, when soil water content was low and close to the wilting point, soil vapor diffusion is expected to become more important for distillation than capillary rise. Theoretically, the soil vapor diffusion rate increases with the decrease of soil water content on the conditions of volumetric soil water content > 10% (Philip and De Vries, 1957; Barnes and Turner, 1998), which makes soil vapor diffusion more important under such conditions. This corresponded to our results of the isotopic composition of soil water that $\delta^{18}O_s$ and $\delta^2H_s$ variability (Fig. 8) was stronger in events 1 and 2 (SD of $\delta^{18}O_s$ and $\delta^2H_s$ was 4.1‰ and 20.1‰ respectively) with a bit lower SWC than that in event 3 (1–3 % lower of SWC, Fig. 1d; SD of $\delta^{18}O_s$ and $\delta^2H_s$ was 2.2‰ and 12.4‰ respectively) with a bit higher SWC. Therefore, future research focusing on the continuous measurements of the isotopic composition ($\delta^{18}O$ and $\delta^2H$) of soil vapor is expected to give more quantitative insights on vapor transfer in soils during nights with dew and radiation fog. According to the results from our mixing model, distillation contributed up to 42% of the total NRW inputs (Fig. 10b), and was the important pathway of NRW inputs during very calm nights ($u_{2m} < 0.7$ m s$^{-1}$; see also Monteith (1957) for $u_{2m} < 0.5$ m s$^{-1}$) besides the condensation of ambient water vapor. According to our mixing model, distillation contributed up to 42% of the total NRW on foliage (Fig. 10b); according to the M57 approach, the distillation amount was 0.04–0.05 mm per night (Fig. 10a) as compared to the NRW from ambient water vapor (0.12–0.50 mm per night) for the three events in our study.

## 6 Conclusion

We investigated the small-scale processes of how fog and dew formation influence the water cycling over a Central European temperate grassland site during representative warm-season nights. Our results revealed different input pathways for dew and radiation fog during three dry intensive observation periods in summer 2018. Dew and radiation fog occurred in clear calm nights with very low wind speed ($u_{2m} < 0.7$ m s$^{-1}$) and weak turbulence, with LE ranging from –26 to 14 W m$^{-2}$. Three primary pathways of NRW gains during the night were investigated in detail: (1) the condensation of atmospheric water vapor to plants, which constitutes a net water gain and might be important during dry spells or droughts; (2) internal recycling by distillation of water vapor from soil onto foliage, thereby re-distributing the water within the ecosystem with no net water gain; and (3) radiation fog droplet deposition, which also leads to a net water gain. Condensation of ambient water vapor during dew and radiation fog was found to be predominantly an equilibrium fractionation process, which was deduced from the rather constant $d_a$ during NRW nights. A decrease of 0.8–1.6‰ $\delta^2$H$_a$ h$^{-1}$ was observed in ambient water vapor, induced by condensation under equilibrium conditions during dew and radiation fog. Local evapotranspiration at high relative humidity from 17:00 CET until sunset caused the lowering of $d_a$ to values in the range of 2.4‰ to 4.8‰ as compared to the higher daytime $d_a$ (12.2‰ to 18.0‰). A further decrease to $d_a$ values in the range of –11.8‰ to –4.7‰ was observed during the occurrence of dew and radiation fog in the night. Dew only formed with unsaturated conditions with a mixed NRW condensing from ambient water vapor and soil-diffusing vapor (distillation). The comparison between the foliage NRW $\delta^{18}$O$_{fNRW}$ and $\delta^2$H$_{fNRW}$ and the equilibrium NRW $\delta^{18}$O$_{aNRW}$ and $\delta^2$H$_{aNRW}$ of ambient water vapor allowed us to trace the source of NRW input pathways during dew formation. Distillation contributed 9–42 % to the total foliage NRW computed from a two end-member mixing model. The dew and radiation fog produced 0.17–0.54 mm d$^{-1}$ NRW gain on foliage, computed from a near-surface vertical temperature gradient method proposed by Monteith in 1957, which constitutes a non-negligible water flux into the canopy compared to evapotranspiration of on average 2.7 mm d$^{-1}$. The strong anti-correlation between $d_a$ and local RH$_0$ suggested an only minor influence of large-scale air advection and highlighted the dominant role of local moisture as a source for ambient water vapor. Our results thus underline the importance of NRW inputs to temperate grasslands during dry spells and reveal the complexity of the local water cycle in such conditions, including different pathways of dew and radiation fog water inputs. In future studies, more intensive and continuous isotope measurements of foliage NRW, ambient water vapor and soil vapor should be combined with direct lysimeter measurements to partition the NRW components from ambient water vapor and soil-diffusing vapor.

**Data availability**

Data was deposited at the ETH Zurich research collection at https://doi.org/10.3929/ethz-b-000465064.

*Author contributions.* YL, AR, WE, and FA designed the project. YL and AR performed the field experiment. YL carried out the laboratory work. YL performed statistical analyses. FA wrote the code of data calibration for the ambient water vapor isotopes. FA and WE commented on the results of the data analysis. YL wrote and revised the manuscript, with contributions and feedbacks by FA, WE, AR, and NB.

*Acknowledgements.* This study was supported by Swiss National Science Foundation (grant 175733, IFDewS project). The Picarro L2130-i analyser was provided by the Atmospheric Dynamics group (Prof. Heini Wernli) at ETH Zurich. We thank Iris Feigenwinter, ETH Zurich, for the quality check of EC and meteorological data as well as measurements of LAI; Dr. Kathrin Fuchs, KIT Garmisch-Partenkirchen (Germany), for her introduction to the Chamau (Chamau) field site; Dr. Lukas Hörtnagl, ETH Zurich, for the method introduction of EC data; Stephanie Westerhuis for the coding of extracting COSMO-1

model data. The Canton of Zug staff are thanked for maintaining the agricultural management of the Chamau field site. The authors also acknowledge MeteoSwiss for the access to the COSMO-1 model data products. We also thank the editor and two anonymous reviewers who provided very insightful and detailed comments that considerably improved this final version.

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

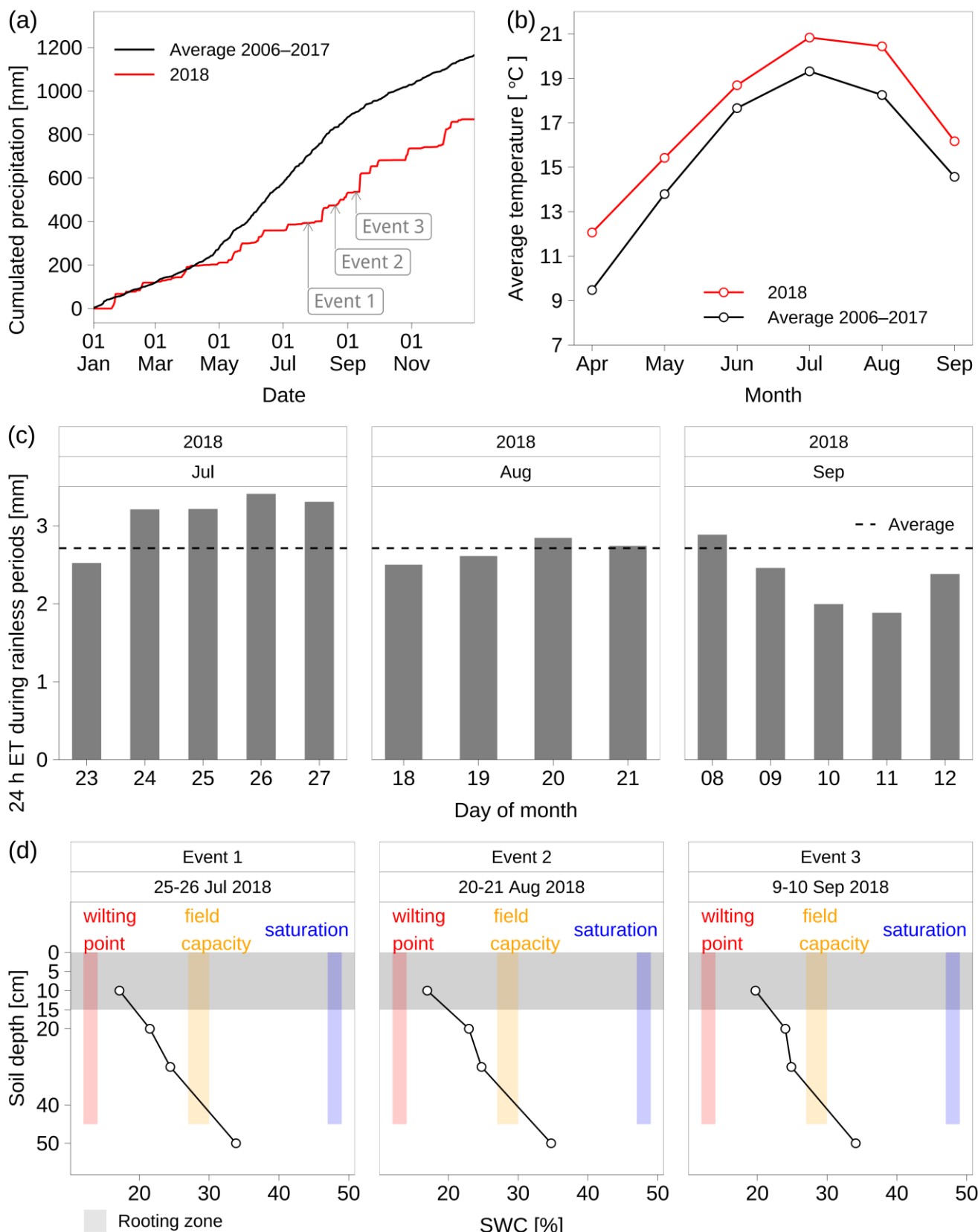

**Figure 1**. Dry and hot summer in 2018. (a) Year-to-date precipitation in 2018 as compared to the average levels over 2006–2017, and the corresponding values before the three events. (b) Average temperature ($T_{a2m}$) from April to September in 2018 as compared to the corresponding average levels over 2006–2017. (c) The 24 h evapotranspiration (ET) during the corresponding rainless periods of the three events. (d) Volumetric soil water content (SWC) at the Chamau site; the wilting point is 12–14 % calculated from Eq. C1 given soil water potential = –1500 kPa and soil texture in Table C1; the field capacity is 27–30 % calculated from Eq. C1 given soil water potential = –33 kPa and soil texture in Table C1; and saturated water content is 47–49 % calculated from Eq. C4 given soil texture in Table C1; the rooting zone is in the top 0–15 cm soil.

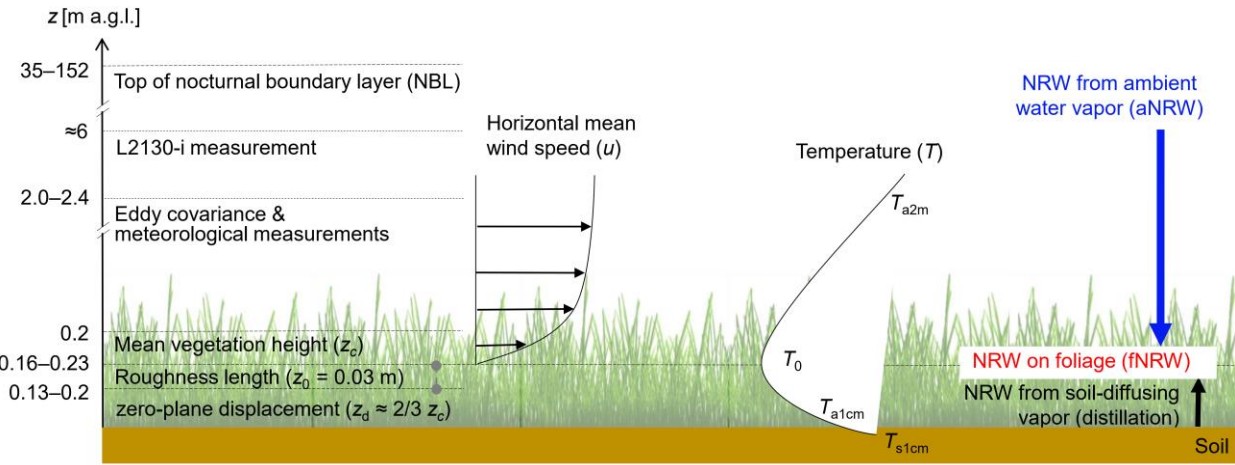

**Figure 2**. Simplified schematics of non-rainfall water (NRW) inputs adapted from Monteith and Unsworth (2013), and Oke (2002): At $z_0 + z_d = 0.16$–0.23 m a.g.l., NRW on foliage (i.e., fNRW) is a mixture of condensate from ambient water vapor (downward) and distillation (i.e., condensate from soil-diffusing vapor, upward). Mean vegetation height was 0.2–0.3 m during the three events; eddy covariance and meteorological measurements were at 2.0–2.4 m a.g.l.; L2130-i measurement was at about 6 m a.g.l.. Horizontal mean wind speed ($u$) was zero at displacement height = 0.16–0.23 m a.g.l.. Temperature ($T$) was measured at 1 cm in soil ($T_{s1cm}$), and 2 m a.g.l. in the atmosphere ($T_{a2m}$); surface temperature ($T_0 = T_{0w}$) was derived from radiation measurement as shown in Eq. 1; air temperature at 1 cm a.g.l. was derived from soil temperature ($T_{s1cm}$) and surface temperature ($T_0 = T_{0w}$) as shown in Eq. 20.


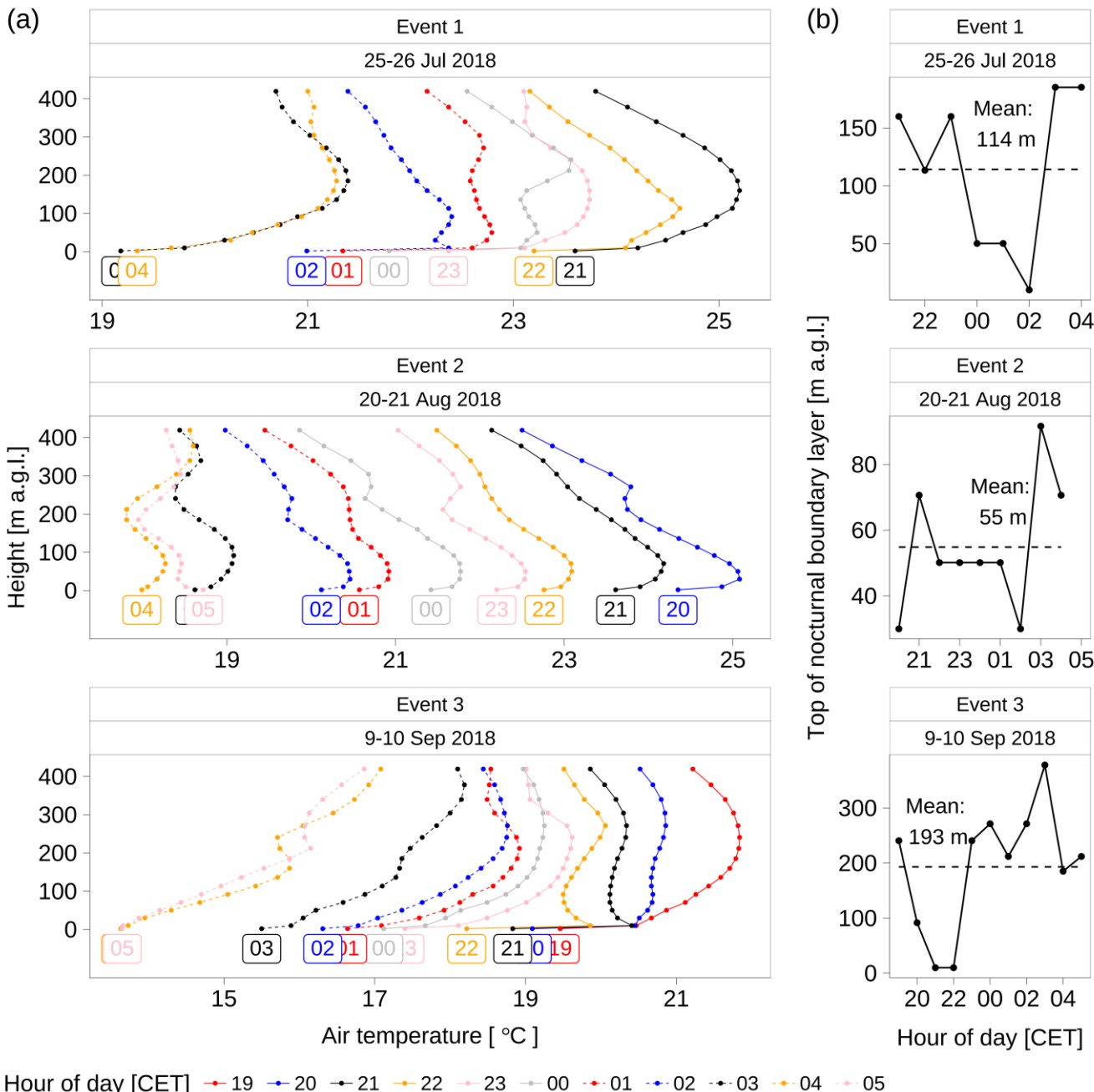

**Figure 3**. Nocturnal boundary layer (NBL) characterised by the vertical profiles of air temperature for the three events interpolated to the location of the Chamau site based on the analysis data  (Doms et al., 2018; Westerhuis et al., 2020) (1.1 km horizontal grid spacing, 80 vertical levels) of the regional numerical weather prediction model COSMO: (a) Hourly air temperature versus height (m a.g.l.); (b) top of nocturnal boundary layer interpreted by the isothermal height, i.e., $\partial T/\partial z = 0$ , where $T$ is air temperature, and $z$ is the height a.g.l..

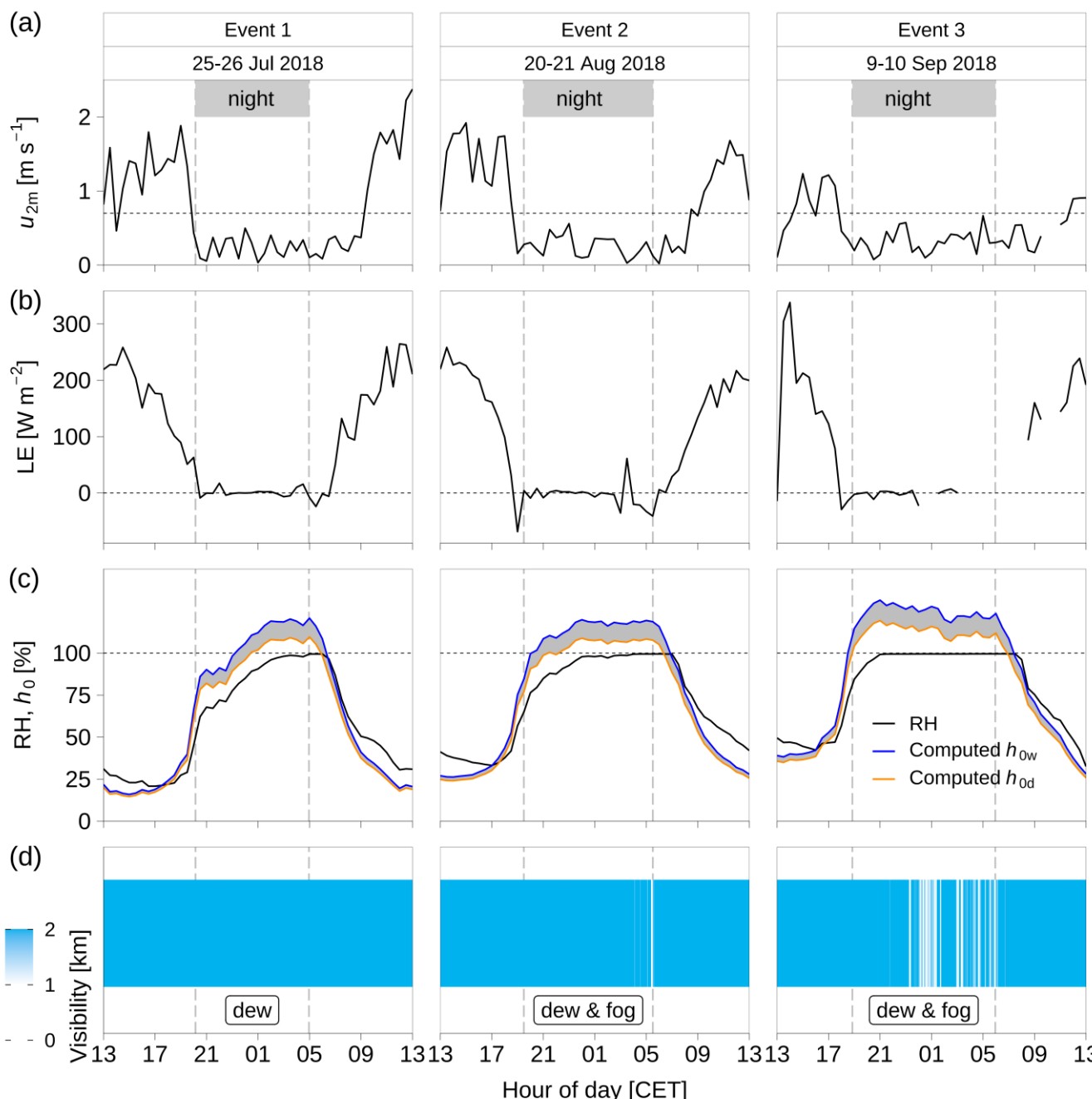

Figure 4. The meteorological and eddy covariance (EC) measurements at the Chamau site. (a) $u_{2m}$, mean wind speed at 2 m a.g.l.; (b) LE, latent heat flux; (c) RH, relative humidity at 2 m a.g.l.; $h_{0w}$ and $h_{0d}$, computed relative humidity with respect to the simulated wet and dry surface temperature; (d) visibility was < 1 km when fog occurred, and visibility was > 1 km with the absence of fog. (a–c) were 30 min average data, and (d) was 1 min data. Vertical dash lines show local sunset and sunrise times. The missing values of LE: As soon as fog occurs or dew drips to the optical windows of the open-path Infrared Gas Analyzer (IRGA), LE measurements become unrealistic and cannot be analyzed quantitatively.

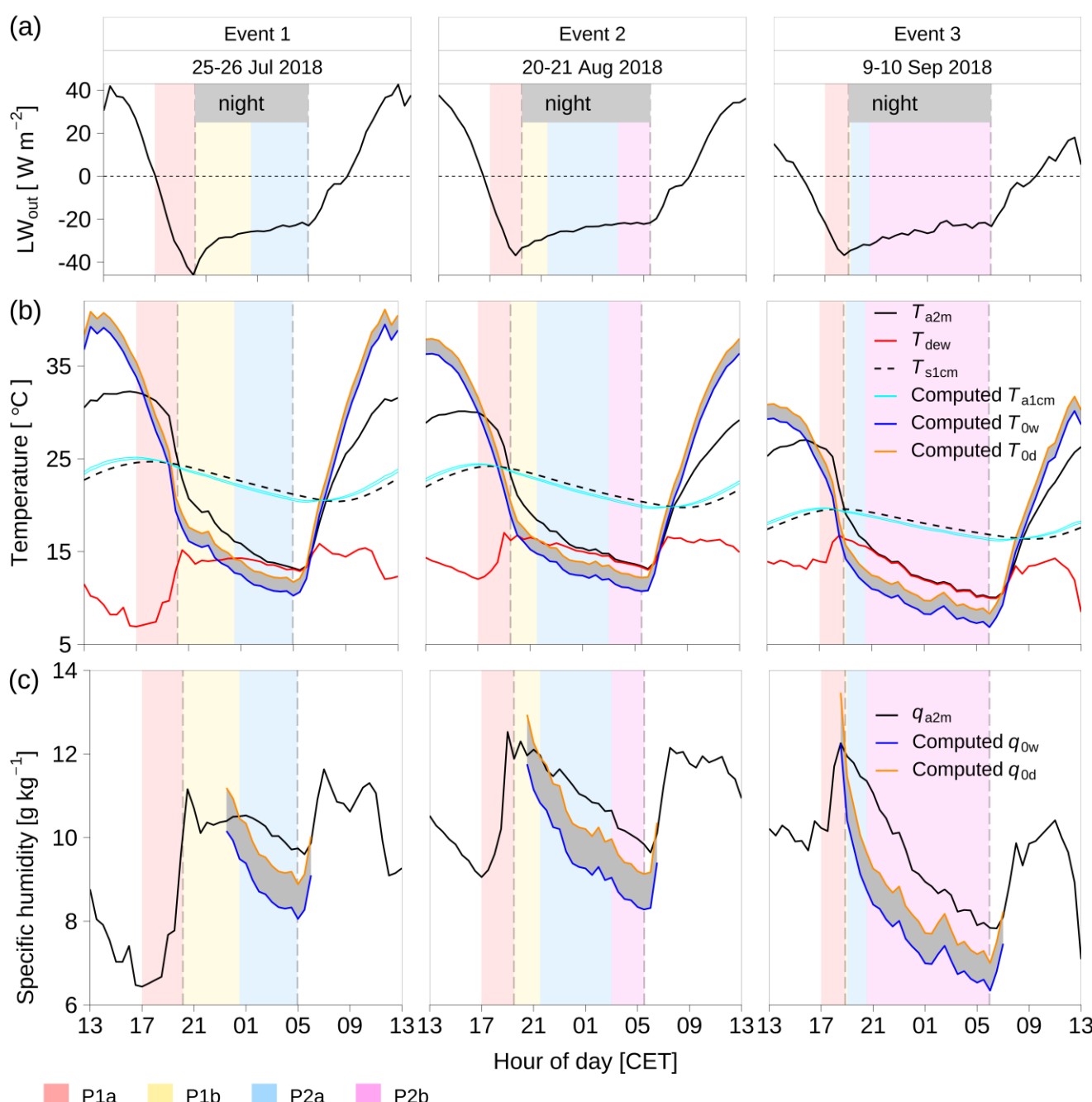

**Figure 5**. The atmospheric and surface conditions at the Chamau site: (a) $LW_{out}$, long-wave outgoing radiation. (b) $T_{a2m}$, air temperature at 2 m a.g.l.; $T_d$, dew-point of the ambient air; $T_{0w}$ and $T_{0d}$, computed wet and dry surface temperature; $T_{s1cm}$, soil temperature at 1 cm below ground; $T_{a1cm}$, computed air temperature at 1 cm a.g.l.. (c) $q_{a2m}$, atmospheric specific humidity at 2.4 m a.g.l.; $q_{0w}$ and $q_{0d}$, computed saturation specific humidity with respect to wet surface temperature $T_{0w}$ and dry surface temperature $T_{0d}$. Vertical dash lines show local sunset and sunrise times. The shaded areas indicated different periods of environmental conditions as described in Sect. 4.1. The P1a period was from around 17:00 CET until sunset with the weakening of turbulence and the increase of specific humidity; P1b period was from sunset until the first sign of condensation with short-term fluctuations of specific humidity; the P2a period was dew formation period in the conditions of relative humidity < 100 %; the P2b period was combined dew and radiation fog period in the conditions of relative humidity = 100 %.


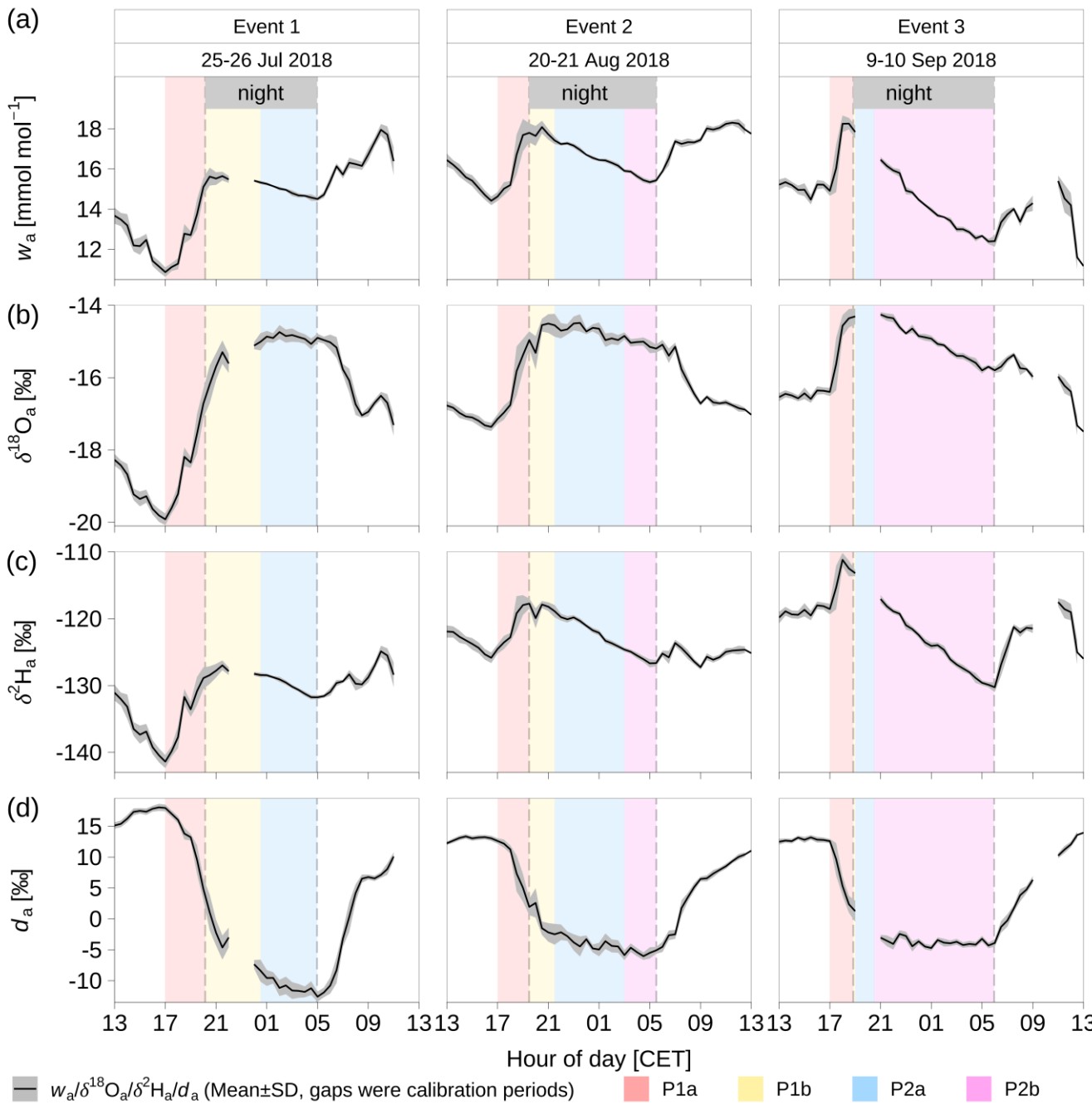

**Figure 6**. The 30 min averages and standard deviations (mean±SD) of the volumetric mixing ratio and isotopic composition for ambient water vapor ($w_a$, $\delta^{18}O_a$, $\delta^2H_a$, and $d_a$). The shaded areas indicated different periods of environmental conditions as described in Sect. 4.1. Data gaps indicate times when the automatic calibration procedure of the spectrometer was active. Vertical dash lines show local sunset and sunrise times. The shaded areas indicated different periods of environmental conditions as described in Sect. 4.1. The P1a period was from around 17:00 CET until sunset with the weakening of turbulence and the increase of specific humidity; P1b period was from sunset until the first sign of condensation with short-term fluctuations of specific humidity; the P2a period was dew formation period in the conditions of relative humidity < 100%; the P2b period was combined dew and radiation fog period in the conditions of relative humidity = 100%.

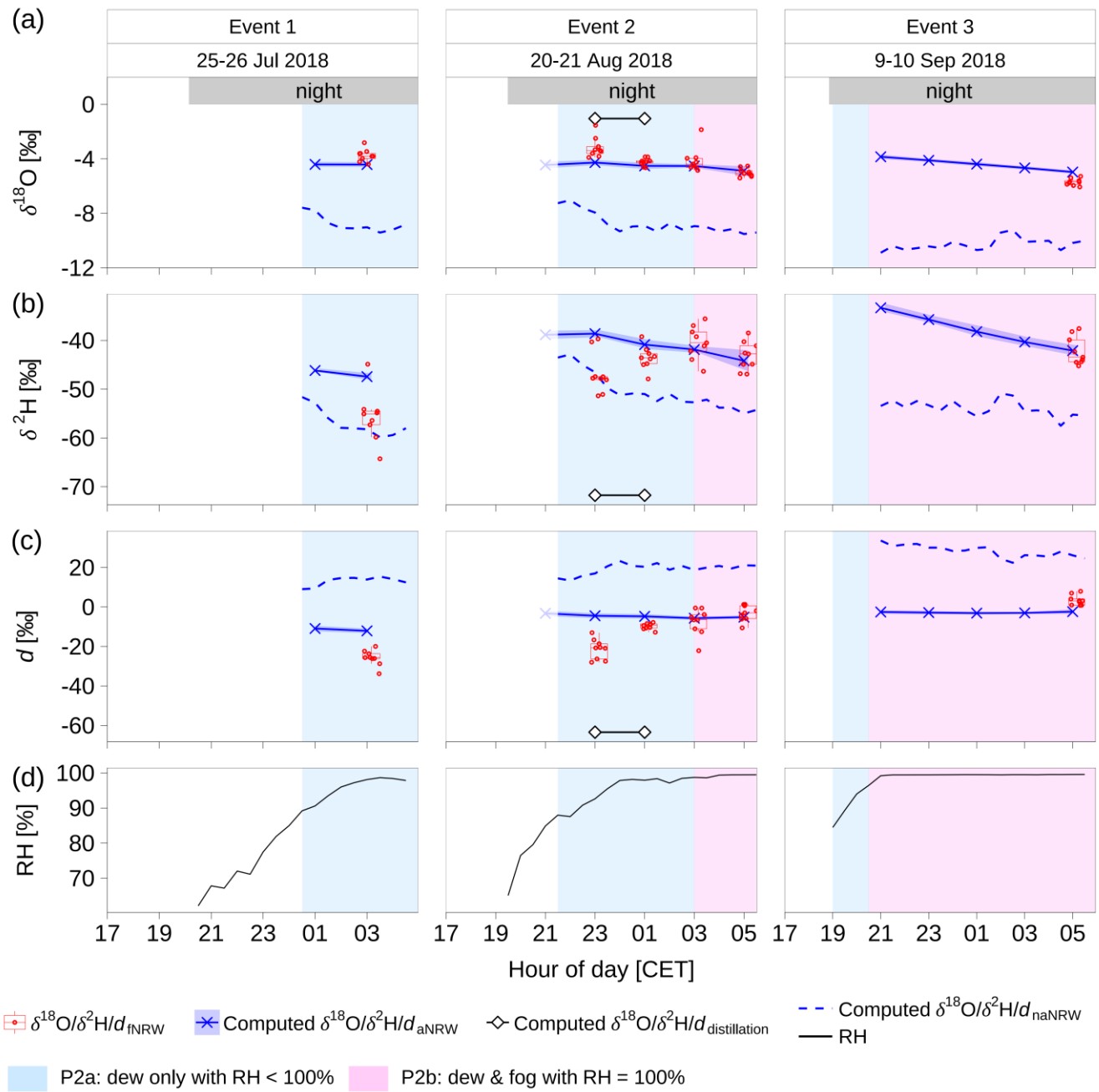

**Figure 7**. The isotopic composition of different non-rainfall water (NRW) components: $\delta^{18}O_{fNRW}$, $\delta^2H_{fNRW}$, and $d_{fNRW}$ for NRW on foliage; $\delta^{18}O_{aNRW}$, $\delta^2H_{aNRW}$, and $d_{aNRW}$ for computed NRW equilibrium liquid from ambient water vapor; $\delta^{18}O_{distillation}$, $\delta^2H_{distillation}$, and $d_{distillation}$ for distillation computed from two end-member mixing model; $\delta^{18}O_{naNRW}$, $\delta^2H_{naNRW}$ and $d_{naNRW}$ for NRW computed from ambient water vapor considering both equilibrium and non-equilibrium factors. The corresponding relative humidity (RH) at 2 m a.g.l. was also shown synchronously. The shaded areas indicated different periods of environmental conditions as described in Sect. 4.1. The P2a period was dew formation period in the conditions of relative humidity < 100%; the P2b period was combined dew and radiation fog period in the conditions of relative humidity = 100%.

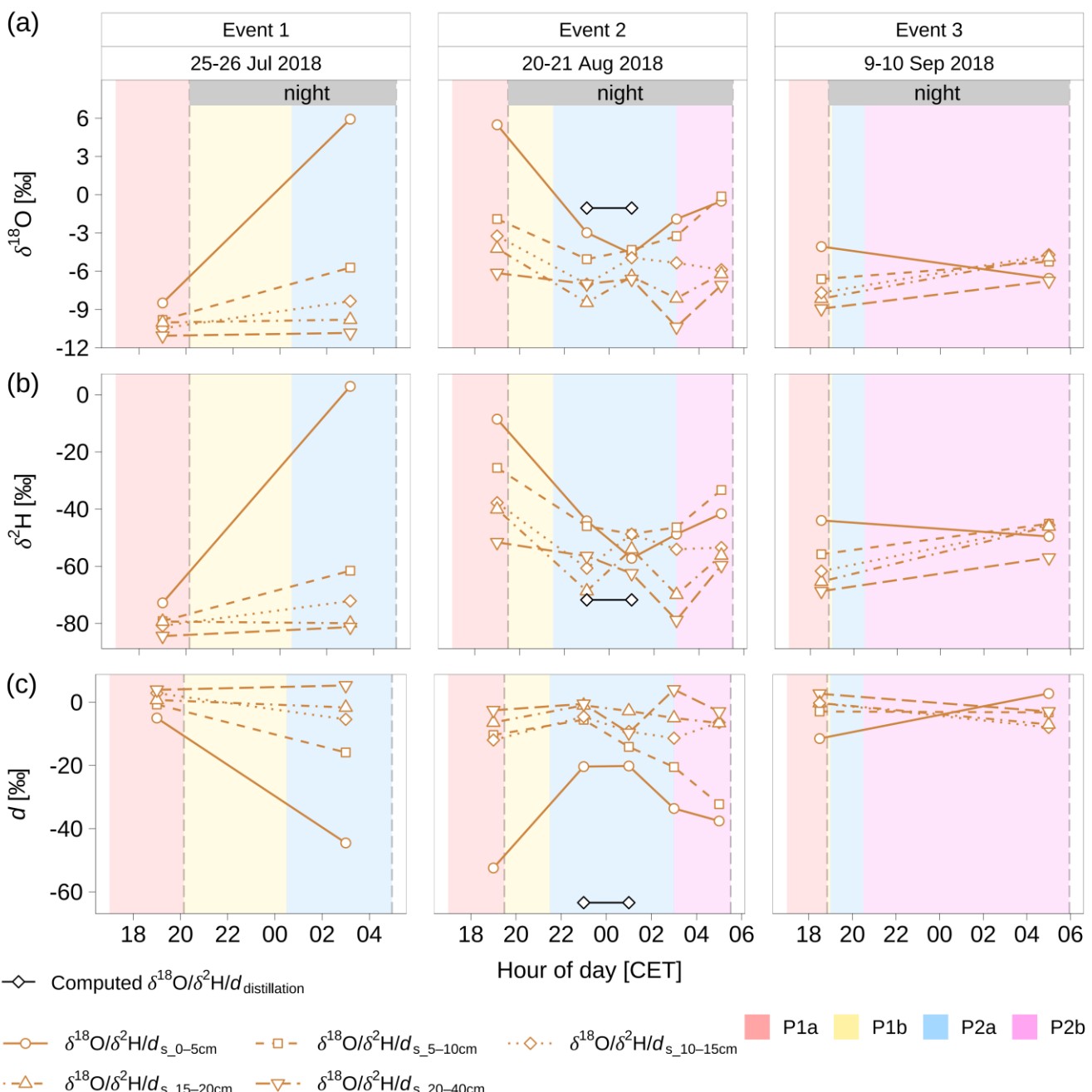

**Figure 8**. The isotopic composition of soil moisture ($\delta^{18}O_s$ and $\delta^2H_s$) at 0–5 cm, 5–10 cm, 10–15 cm, 15–20 cm, 20–40 cm as compared to the isotopic composition of distillation ($\delta^{18}O_{distillation}$ and $\delta^2H_{distillation}$) computed from two end-member mixing model. Vertical dash lines show local sunset and sunrise times. The shaded areas indicated different periods of environmental conditions as described in Sect. 4.1. The P1a period was from around 17:00 CET until sunset with the weakening of turbulence and the increase of specific humidity; P1b period was from sunset until the first sign of condensation with short-term fluctuations of specific humidity; the P2a period was dew formation period in the conditions of relative humidity < 100%; the P2b period was combined dew and radiation fog period in the conditions of relative humidity = 100%.

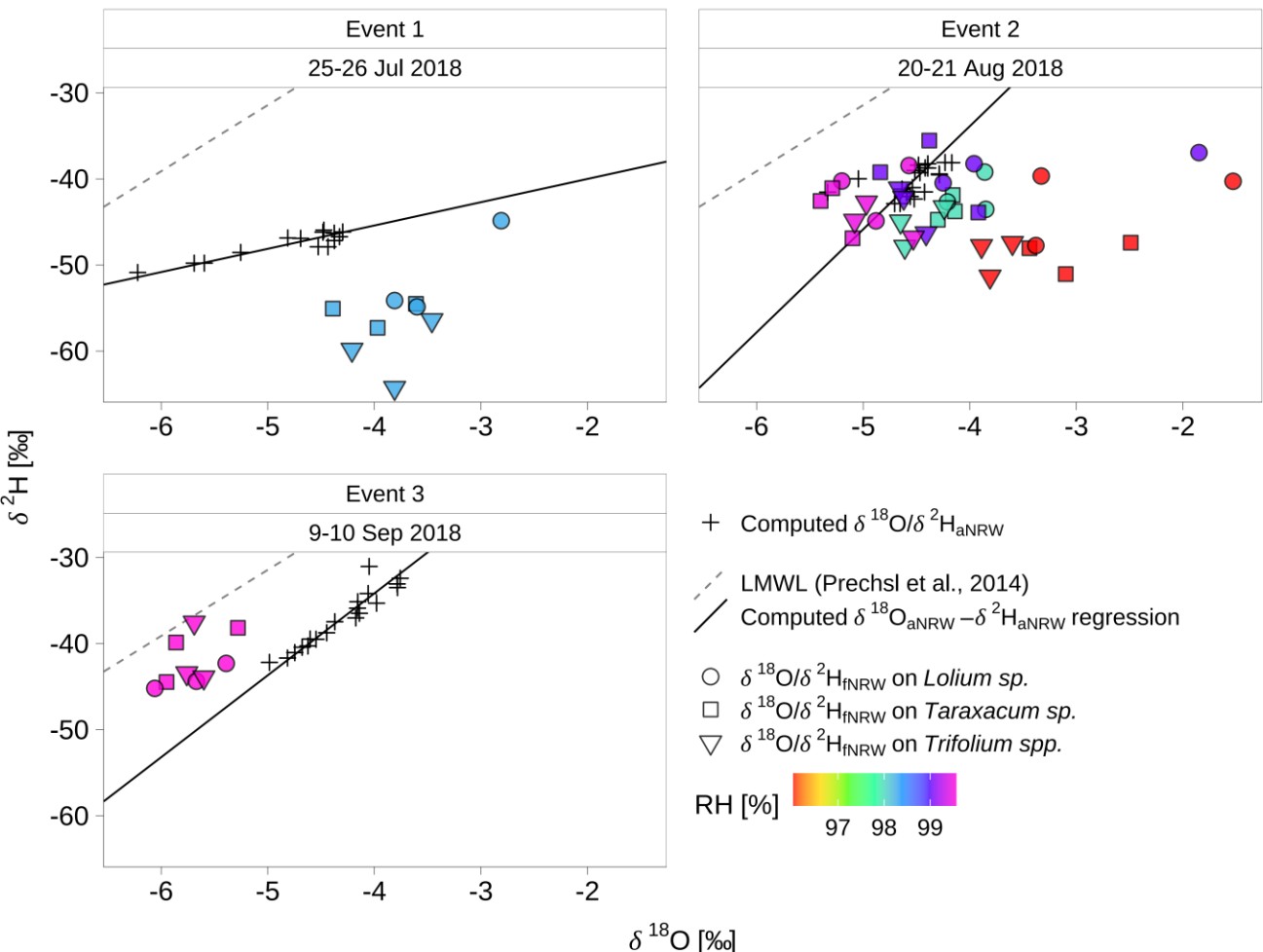

**Figure 9**. The relationship of $\delta^2H_{fNRW} - \delta^{18}O_{fNRW}$ for non-rainfall water (NRW) on foliage with respect to the orthogonal regression of $\delta^2H_{aNRW} - \delta^{18}O_{aNRW}$ for NRW equilibrium liquid from ambient water vapor, and local meteorological water line (LMWL: $\delta^2H = 7.68 \times \delta^{18}O + 6.97$, Prechsl et al. (2014)). The filled colours of $\delta^{18}O_{fNRW}/\delta^2H_{fNRW}$ represented the corresponding relative humidity at 2 m a.g.l. (RH).

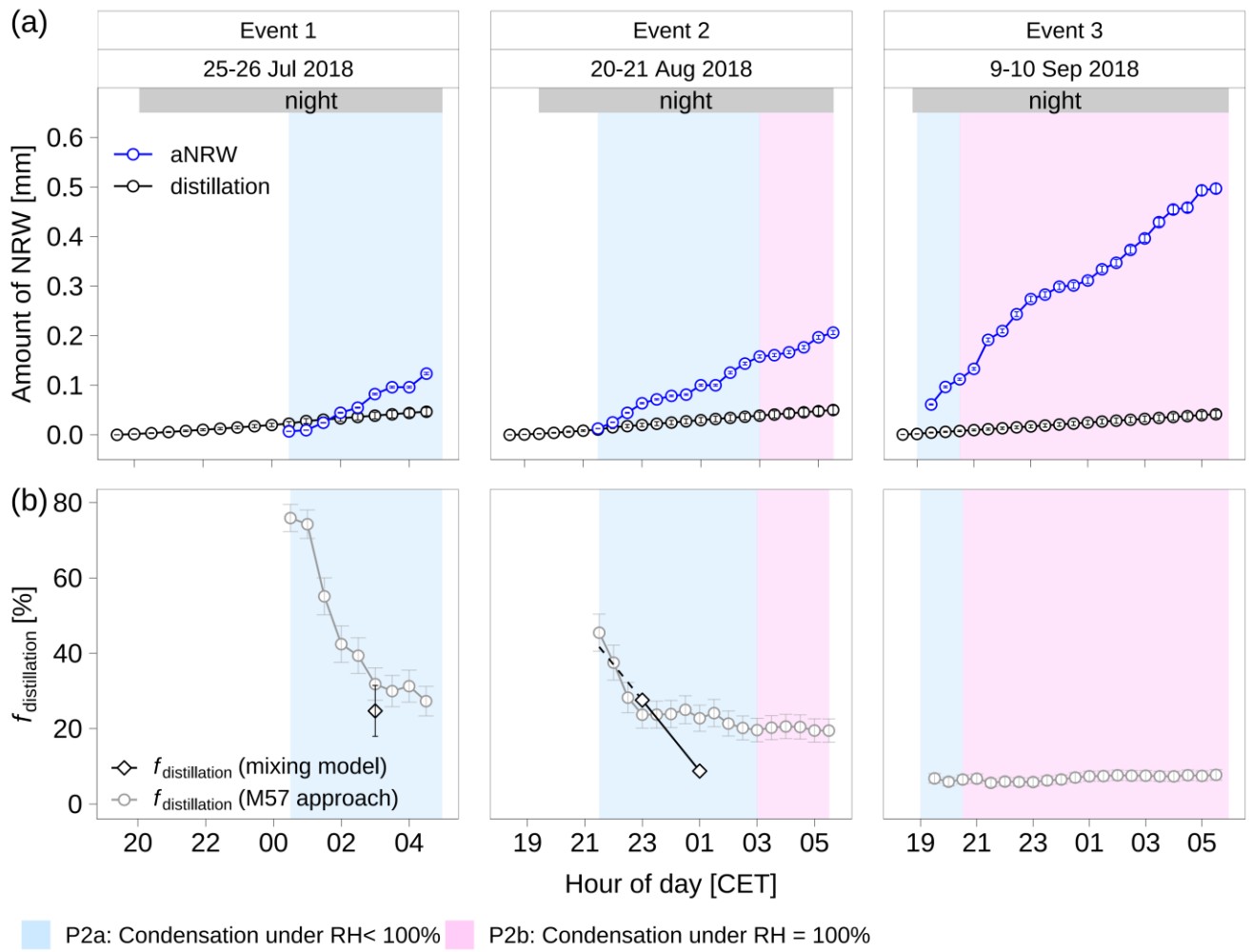

**Figure 10**. Computed amounts of non-rainfall water (NRW), and the contribution of distillation ($f_{distillation}$) in the total NRW on foliage (fNRW): (a) computed amount of NRW condensing from ambient water vapor (aNRW), and computed amount of distillation by M57 approach. (b) Ratio of distillation $f_{distillation}$ in NRW on foliage computed from two end-member mixing model (black), and ratio of distillation $f_{distillation}$ in total NRW by M57 approach as described in Sect. 3.2.5 (grey). The P2a period was dew formation period in the conditions of relative humidity < 100%; the P2b period was combined dew and radiation fog period in the conditions of relative humidity = 100%.

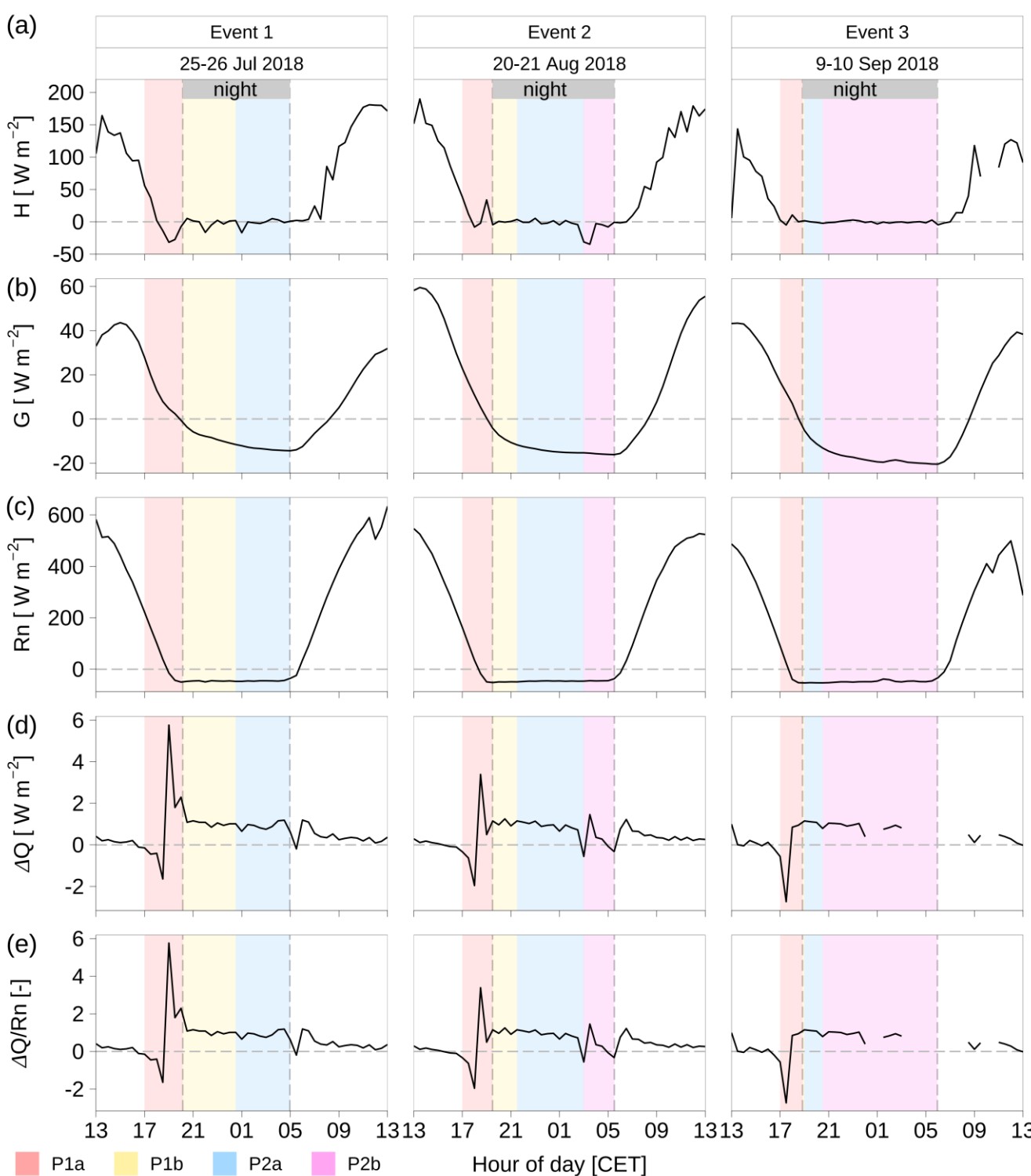

**Figure 11**. The 30 min heat flux measurement during the three events: (a) H is sensible heat flux; (b) G is ground heat flux; (c) Rn, net radiation flux; (d) ΔQ is the budget closure term which accounts for all unmeasured advective fluxes and for the measurement errors of the measured fluxes. (e) ΔQ/Rn is the ratio of budget closure term ΔQ to net radiation flux Rn. Vertical dash lines show local sunset and sunrise times. The shaded areas indicated different periods of environmental conditions as described in Sect. 4.1. The P1a period was from around 17:00 CET until sunset with the weakening of turbulence and the increase of specific humidity; P1b period was from sunset until the first sign of condensation with short-term fluctuations of specific humidity; the P2a period was dew formation period in the conditions of relative humidity < 100 %; the P2b period was combined dew and radiation fog period in the conditions of relative humidity = 100 %.

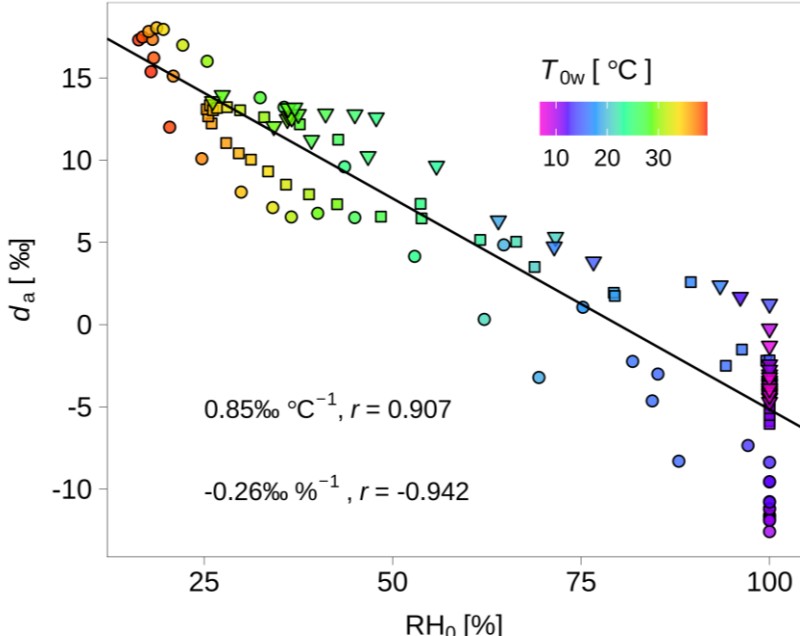

**Figure 12**. The orthogonal regression of deuterium excess of ambient water vapor ($d_a$) with surface relative humidity ($RH_0$) computed from volumetric water vapor mixing ratio ($w_a$) using Eq. 3, and air temperature ($T_{a2m}$) at 2 m a.g.l.. Slopes and Pearson's $r$ for regressions were shown.

**Table 1**. Partitioning the contribution of distillation from a mix of distillation and aNRW. The fNRW means non-rainfall water (NRW) on foliage; aNRW represents either dew or radiation fog, or dew and radiation fog in combination condensed from ambient water vapor; distillation means dew condensed from soil-diffusing vapor; $f_{distillation}$ means the proportion of distillation in total foliage NRW.

| Event | Time | Isotope | fNRW | aNRW | dDew | $f_{dDew}(\%)$ |
|-------|------|---------|------|------|------|------|
| Event 1 | 3:00 CET | $\delta^{18}O$ (‰) | −3.8 | −4.4±0.2 | −1.0 | 18–31 |
| | | $\delta^2H$ (‰) | −55.1 | −47.4±1.7 | −71.8 | |
| | | $d$ (‰) | −25.6 | −12.1±1.3 | −63.4 | |
| Event 2 | 21:30 CET | No sampling, but extrapolating from 23:00 and 1:00 CET | | | | 42 |
| | 23:00 CET | $\delta^{18}O$ (‰) | −3.4 | −4.3±0.2 | −1.0 | 28 |
| | | $\delta^2H$ (‰) | −47.7 | −38.6±0.7 | −71.8 | |
| | | $d$ (‰) | −20.7 | −4.4±1.3 | −63.4 | |
| | 1:00 CET | $\delta^{18}O$ (‰) | −4.2 | −4.5±0.2 | −1.0 | 9 |
| | | $\delta^2H$ (‰) | −43.5 | −40.8±1.0 | −71.8 | |
| | | $d$ (‰) | −9.4 | −4.7±1.1 | −63.4 | |

**Table 2**. Variability of the isotopic composition among species for non-rainfall water on foliage, and leaf water. The 25% quantile, median, and 75% quantile are shown. The different letters (a–b) after the statistical values show the significance of within-species differences using Tukey's honest significant differences (HSD) test.

| Sample | Isotope | Isotopic composition of different species (25% quantile, median, 75% quantile) and Tukey's test results | | |
|--------|---------|---------|---------|---------|
| | | *Lolium sp.* | *Taraxacum sp.* | *Trifolium spp.* |
| NRW on foliage (fNRW) | $\delta^{18}O_{fNRW}$ | (−4.8, −3.9, −3.4) a | (−5.2, −4.3, −3.9) a | (−4.9, −4.6, −4.0) a |
| | $\delta^2H_{fNRW}$ | (−44.9, −42.5, −39.8) a | (−47.9, −44.2, −41.3) a | (−47.9, −45.6, −43.3) a |
| | $d_{fNRW}$ | (−21.8, −8.7, −2.9) a | (−20.4, −9.5, 0.3) a | (−18.1, −10.0, −3.9) a |
| Leaf water | $\delta^{18}O_{leaf}$ | (−4.2, −3.9, −3.6) b | (−4.9, −4.4, −3.4) b | (−4.0, −3.5, −2.1) a |
| | $\delta^2H_{leaf}$ | (−42.7, −38.7, −38.0) b | (−41.2, −38.1, −36.5) b | (−37.7, −36.8, −32.6) a |
| | $d_{leaf}$ | (−11.4, −8.5, −4.6) ab | (−10.4, −5.1, −1.6) a | (−16.2, −12.4, −6.9) b |

**Appendix A: Study site**

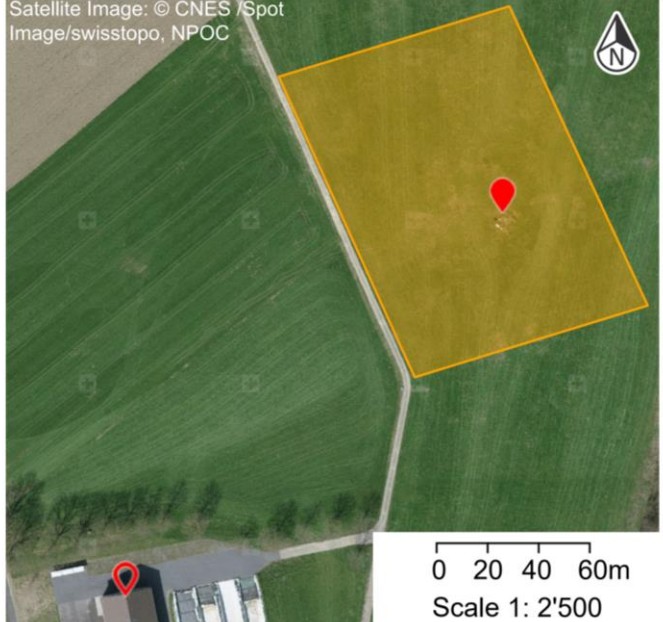

○ L2130-i
● EC & meteorological measurements
▧ Sampling of fNRW, soil, and leaf

**Figure A1**. Measurements and sampling at the Chamau site (Satellite Image: © CNES /Spot Image/swisstopo, NPOC). "L2130-i" represents the isotopic composition and mixing ratio measurements for ambient water vapor; EC, eddy covariance; fNRW, non-rainfall water on foliage.

**Appendix B: Calculating environmental variables**

The eddy covariance fluxes were calculated using the software EddyPro (version 7.0.6, LI-COR (2017)) and following established community guidelines (Aubinet et al., 2012). Eddy covariance raw data were despiked and screened following Vickers and Mahrt (1997). Wind data was rotated (2D rotation, Wilczak et al. (2001)) and time lags between the turbulent wind and $H_2O$ data were compensated using covariance maximization. For spectral corrections, fluxes were corrected for high-pass and low-pass filtering effects (Moncrieff et al., 2005; Fratini et al., 2012) and instrument separation (Horst and Lenschow, 2009). Processed $H_2O$ fluxes were rejected from further analyses (1) if they were found outside a physically plausible range (between -20 and 50 mmol $H_2O$ m$^{-2}$ s$^{-1}$) and (2) if they failed the tests for stationarity and well-developed turbulence (e.g., Foken et al. (2005)).

The saturation specific humidity ($q_0$, g kg$^{-1}$) for wet ($q_{0w}$) and dry ($q_{0d}$) vegetation surfaces was calculated as (Garratt, 1992):

$$q_0 = \frac{622 \cdot e_{s0}}{p - 0.378 \cdot e_{s0}} \text{ ,} \tag{B1}$$

where $p$ in hPa is air pressure, and $e_{s0}$ in hPa is saturation vapor pressure at $T_0$ calculated as (Garratt, 1992):

$$e_{s0} = 6.112 \cdot \exp\left(\frac{17.67 \cdot T_0}{T_0 + 243.5}\right) \text{ .} \tag{B2}$$

The dew point temperature ($T_d$, °C) was calculated as (Garratt, 1992):

$$T_d = 243.5 \cdot \frac{\ln\left(\frac{e_{sa} \cdot RH}{6.112}\right)}{17.67 - \ln\left(\frac{e_{sa} \cdot RH}{6.112}\right)} \text{ ,} \tag{B3}$$

where $e_{sa}$ in hPa is saturation vapor pressure at $T_a$ calculated as (Garratt, 1992):

$$e_{sa} = 6.112 \cdot \exp\left(\frac{17.67 \cdot T_a}{T_a + 243.5}\right) \text{ .} \tag{B4}$$

The evapotranspiration rate (in mm h$^{-1}$) was calculated from the turbulent latent heat flux (LE in W m$^{-2}$) as (Stull, 1988):

$$ET = b \frac{LE}{\lambda \cdot \rho_{H2O}} \text{ ,} \tag{B5}$$

where $\lambda = (2.501 - 0.00237 \cdot T_a) \cdot 10^6$ (Stull, 1988), $\rho_{H2O} = 10^3$ kg m$^{-3}$ is water density, and b is a unit conversion factor ($3.6 \cdot 10^6$ mm m$^{-1}$ s h$^{-1}$). Negative values of ET indicate dew formation below the eddy covariance flux instrumentation.

## Appendix C: Soil characteristics

The soil water potential ($\psi_s$ in kPa) was calculated from soil water content ($\theta_s$) as (Saxton et al., 1986):

$$\theta_s = \exp\left[\frac{\ln(\frac{-\psi_s}{A})}{B}\right] , \tag{C1}$$

where

$$A = 100 \cdot \exp[-4.396 - 0.0715 \cdot (\% \text{ clay}) - 4.880 \cdot 10^{-4} \cdot (\% \text{ sand})^2 - 4.285 \cdot 10^{-5} \cdot (\% \text{ sand})^2 \cdot (\% \text{ clay})] , \tag{C2}$$

and

$$B = -3.140 - 2.22 \cdot 10^{-3} \cdot (\% \text{ clay})^2 - 3.484 \cdot 10^{-5} \cdot (\% \text{ sand})^2 \cdot (\% \text{clay}) , \tag{C3}$$

where (% sand) and (% clay) are percent sand and clay, respectively; wilting point and field capacity were calculated given $\psi_s$ = −1500 kPa, and $\psi_s$ = −33 kPa, respectively (Rai et al., 2017).

The saturated water content was calculated as (Saxton et al., 1986):

$$\theta_{s\_saturation} = 0.332 - 7.251 \cdot 10^{-4} \cdot (\% \text{ sand}) + 0.1276 \cdot \log_{10} (\% \text{ clay}) . \tag{C4}$$

**Table C1**. Wilting point, field capacity, and saturated water content of soil in volumetric soil water content calculated from soil texture by Roth (2006) at the Chamau site using the methods by Saxton et al. (1986). Wilting point and field capacity were calculated from Eqs. (C1–C3) given the soil water potential $\psi_s$ = −1500 kPa, and $\psi_s$ = −33 kPa, respectively. Saturated water content was calculated from Eq. C4.

| Profile | Depth | % sand | % clay | Wilting point ($\psi_s$ = −1500 kPa) | Field capacity ($\psi_s$ = −33 kPa) | Saturated water content |
|---------|-------|--------|--------|--------------------------------------|-------------------------------------|-------------------------|
| 1 | 0–20 | 35.8 | 19.0 | 12% | 27% | 47% |
| 2 | 0–15 | 25.4 | 24.4 | 14% | 30% | 49% |

**Appendix D. Roughness length computed from wind speed and friction velocity in neutral nights**

In relatively windy nights when the leaf surfaces remained dry (Monteith, 1957; Garratt, 1992), the roughness length ($z_0$) at the Chamau site was computed from wind speed ($u_{2m}$) and friction velocity ($u*$) following Monin–Obukhov similarity theory (Monteith, 1957; Garratt, 1992) as shown in Eq. 9. We selected two relatively windy nights, i.e., neutral atmospheric stratification (Panofsky, 1984), from 2018-06-22 to 2018-06-23, and from 2018-07-01 to 2018-07-02 to calculate roughness length $z_0$. During these two nights, no precipitation occurred, and the latent heat flux (LE) was purely upward (i.e., no

condensation), therefore leaf surfaces remained dry. The average of roughness length $z_0$ was thus 0.03 m (Table D1). No harvest occurred since these two nights till the three events (see Sect. 3.2), and the grassland height was 0.2–0.3 m, therefore the grassland growth causes minor change of $z_0$.

**Table D1**. Computing the roughness length ($z_0$) at the Chamau site from wind speed ($u_{2m}$) and friction velocity ($u*$) in neutral nights.

| Date (yyyy-mm-dd) | Sunset (CET) | Sunrise (CET) | Time (CET) | Precipitation (mm) | Latent heat flux (LE, W m$^{-2}$) | Wind speed ($u_{2m}$, m s$^{-1}$) | Friction velocity ($u*$, m s$^{-1}$) | Roughness length ($z_0$, m) |
|---|---|---|---|---|---|---|---|---|
| 2018-06-22 | 20:26 | | 21:30 | 0 | 27 | 1.0 | 0.13 | 0.10 |
| | | | 22:00 | 0 | 31 | 0.8 | 0.09 | 0.04 |
| | | | 22:30 | 0 | 36 | 1.5 | 0.15 | 0.04 |
| | | | 23:00 | 0 | 25 | 1.7 | 0.14 | 0.02 |
| | | | 23:30 | 0 | 35 | 1.7 | 0.15 | 0.02 |
| 2018-06-23 | | 4:30 | 0:00 | 0 | 18 | 1.2 | 0.12 | 0.04 |
| | | | 0:30 | 0 | 31 | 1.5 | 0.15 | 0.03 |
| | | | 1:00 | 0 | 22 | 2.0 | 0.15 | 0.01 |
| | | | 1:30 | 0 | 17 | 1.6 | 0.15 | 0.03 |
| | | | 2:00 | 0 | 11 | 1.2 | 0.13 | 0.05 |
| | | | 2:30 | 0 | 7 | 0.6 | 0.05 | 0.03 |
| | | | 3:00 | 0 | 22 | 0.5 | 0.02 | 0.00 |
| 2018-07-01 | 20:25 | | 21:00 | 0 | 35 | 1.1 | 0.08 | 0.01 |
| | | | 21:30 | 0 | 43 | 0.9 | 0.08 | 0.02 |
| | | | 22:00 | 0 | 34 | 1.4 | 0.16 | 0.05 |
| | | | 22:30 | 0 | 30 | 1.6 | 0.17 | 0.05 |
| | | | 23:00 | 0 | 28 | 1.8 | 0.16 | 0.02 |
| | | | 23:30 | 0 | 24 | 2.0 | 0.17 | 0.02 |
| 2018-07-02 | | 4:34 | 0:00 | 0 | 21 | 1.7 | 0.15 | 0.02 |
| | | | 0:30 | 0 | 19 | 1.7 | 0.16 | 0.03 |
| | | | 1:00 | 0 | 14 | 1.4 | 0.14 | 0.03 |
| | | | 1:30 | 0 | 24 | 1.4 | 0.16 | 0.06 |
| | | | 2:00 | 0 | 22 | 1.1 | 0.10 | 0.03 |
| | | | 2:30 | 0 | 42 | 1.1 | 0.07 | 0.00 |
| | | | 3:00 | 0 | 40 | 1.2 | 0.11 | 0.03 |

## Appendix E. Semi-empirical forms of stability parameter

The stability parameter $\Phi$ is proportional to Richardson number $Ri$ with numerous semi-empirical forms (Garratt, 1992). In our results we followed the term $\Phi = 1/(1+10 \cdot Ri)$ as suggested in Monteith's (1957) original publication that we refer to. However, in Monteith and Unsworth (2013), $\Phi$ is given as:

$$\Phi = [1 - 16 \cdot (z - z_\mathrm{d})/L]^{-0.5} = [1 - 16 \cdot Ri]^{-0.5} \text{ for } Ri < -0.1 , \tag{E1}$$

$$\Phi = [1 + 5 \cdot (z - z_\mathrm{d})/L] = [1 - 5 \cdot Ri]^{-1} \text{ for } -0.1 \leq Ri \leq 1 . \tag{E2}$$

In Fig. E1 we assess the effect that a replacement of Monteith's (1957) original approach (Fig. E1a,b) has if it is replaced by the variant found in Monteith and Unsworth (2013) (Fig. E1c,d). The change does not affect our estimates for distillation (red curves in Fig. E1a,c) but increases the cumulative NRW amount gained over each of the three events studied (blue curves in Fig. E1a,c) by 34%, 75%, and 43%, respectively.

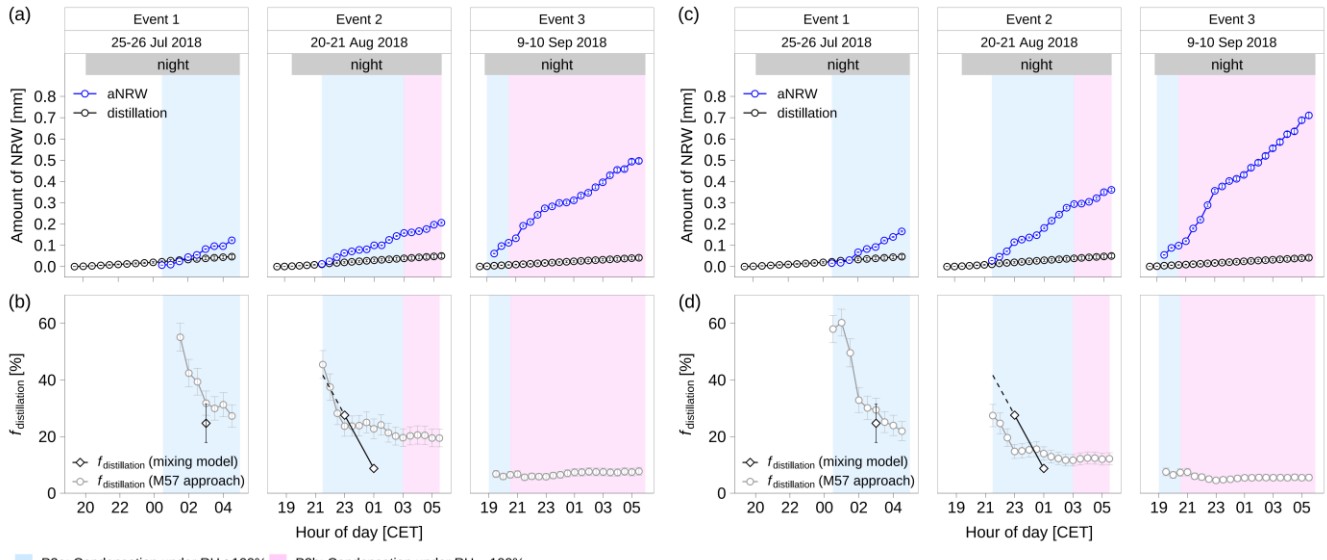

Figure E1. The non-rainfall water amount and distillation contribution given different semi-empirical forms of stability parameter $\Phi$. (a) and (b) are the results given $\Phi = 1/(1+10 \cdot Ri)$ following Monteith (1957). (c) and (d) are the results given $\Phi = [1 - 16 \cdot (z - z_\mathrm{d})/L]^{-0.5} = [1 - 16 \cdot Ri]^{-0.5}$ for $Ri < -0.1$, and $\Phi = [1 + 5 \cdot (z - z_\mathrm{d})/L] = [1 - 5 \cdot Ri]^{-1}$ for $-0.1 \leq Ri \leq 1$ following Monteith and Unsworth (2013). The P2a period was dew formation period in the conditions of relative humidity < 100%; the P2b period was combined dew and radiation fog period in the conditions of relative humidity = 100%.
