# Peer review of "The role of dew and radiation fog inputs in the local water cycling of a temperate grassland during dry spells in Central Europe"

_Hydrology and Earth System Sciences, 2020_

## Referee Comment (RC1) · Anonymous Referee #1 · 29 Nov 2020

The manuscript presents an interesting topic on non-rainfall water. The authors analyses for 3 events the water in the atmosphere and on the plants of an temperate grassland in Central Europe. The authors report data from a well-equipped test site and showed based on observation that dew formation and fog deposition are an overlooked part of the water cycle at such locations. The manuscript is overall well written, but the structure of the subchapter sometimes makes it difficult to follow the red line, and how this helps to answer the formulated aims of the manuscript. Several aspects in the manuscript require a further improvement; clarification and especially a broader discussion of their results including results on the third objective (see further details in "General and Specific comments"). I want here to emphasize that it was a very interesting read and that the topic is of current interest for readership of HESS. I recommend a major revision and encourage the authors to carefully rewrite, revise and improve their manuscript.

General comments:

1) A lot of subchapters and abbreviations makes it sometime difficult to follow the red line of the manuscript. I suggest restructuring the chapter/section in order to answer to aims/objectives of the investigation.

2) I recommend adding a much broader discussion on the formation of NRW, including the parallel condensation of water by soil distillation and dew in the introduction.

3) NRW during prolong drought periods. Please use a common definition on the periods during the measurements e.g. term drought or hot days.

4) For the third objective, there is no data shown in the manuscript that could give new insight here in the results section and authors only discuss potential impact of NRW on ecological functions. Please clarify by adding further points in results section and describe how this was done (M&M section) that justifies the mention the third objective. E.g. the authors could include soil moisture observations during events (result section). Then discuss based on this results their ecological relevance in the corresponding discussion section.

5) There is the need to show the latent heat flux measured with the EC-tower in this manuscript, which might help to clarify some points referring the observations on fNRW or dDew. It would be also helpful to see if EC-station can even indicate the formation of dew at night.

6) Discussion on the outcomes of the results are very short and partially parts of 4.1 should be shift into result section.

7) Add result on potential NRW section into the results section and explain in addition the used methods in the M&M section.

Specific comments:

L7: NRW is more than dew and fog. Thus, I recommend using here in the text: [. . .] (hereafter NRW) mostly formed from dew and fog [. . .]

L10: I recommend changing: condensation of soil-diffusing vapor to condensation of water vapor evaporating from the soil in the canopy (i.e. soil distillation), [. . .]. The processes described here by the authors is related to the term dewrise or soil distillation, whereby I recommend sticking with the latter term also use in Monteith (1957) within this manuscript.

L22: [. . .] (2) of soil-diffusing vapor. Please clarify that water from soil distillation was not measured in this study, but was determine/assumed as end member.

L22: Please clarify the sentence why a potential of 0.06 – 0.39 mm per night are comparable to 2.8 mm daytime ET. Even after reading the entire manuscript it wasn't clear to me how the authors came up to this statement and values.

L28: [. . .] water deposition. I recommend to change it to: [. . .] water condensation and deposition. Please differentiate in the manuscript for condensation (dew) and fog (deposition).

L30: [. . .] (hereafter NRW) inputs, namely dew and fog. Please name first all possible contributor to NRW on the soil or canopy surface: dew, fog, water vapor adsorption, soil distillation, and guttation.

L51: Please make clear that the authors refer here to the crop water use efficiency (WUE = ANPP/ET) as in other WUE definition only transpiration are used

L88: [. . .] onto foliage, [. . .]. Please change: onto the plant or soil surface.

L87-89: but water on plants can also stem from guttation. Please discuss this here and add also info on this at a later stage of the manuscript, how this might affect the results of the study, because water from this might be isotopically different, from other water

sources in the plant-soil-atmospheric continuum (e.g. ambient water vapor, soil water, plant water).

L92: delete After

L103: [. . .] (3) assess the potential ecological relevance of NRW inputs. The authors report here observation for three events, but no further observations that could allow to make some statement on ecological relevance of NRW. I could not find any method used here to realize this in the Material & Method section and no results are reported within this manuscript on this point. Only in the discussion section, authors discuss potential impact of NRW on ecological functions! Please clarify the point that justifies this objective and how the authors answer this within the manuscript.

L112-116: The authors report that rainfall amount in 2018 was ∼297 mm less than the long-term annual rainfall. In the next sentence, they report that during the 6 months period (April – September) the monthly rainfall, which was 81 mm, were reduced by 38% (49 mm). Something went wrong here, because 49 mm less rainfall per month (April-September) would mean a reduction of 60.5 % per month. I am also wondering that these reported values would mean that during the other months the rainfall was similar to the long terms values? As 6*49 = 294 mm and the total difference between 2018 and the long-term values was 297 mm.

L117-118: The authors discuss their results in the light of a prolonged drought, but looking at the 3 measurement campaigns only the first event was within a month with less rainfall, because in August monthly rainfall was similar to long-term rainfall, and the monthly rainfall in September was with ∼130mm much larger as the long term mean rainfall (∼80 mm). If the authors want to relate their NRW results to term drought (especially important for the ecological relevance part), they need first to define this! Perhaps use better the definition of hot days during the extreme year 2018, instead of the term drought, as only the month in July showed a severe rainfall deficit and not the months August and September.

L125-127: I recommend reformulating this sentence. The info in the brackets are larger than the rest of the sentence. Please change this for the whole manuscript, as relevant information should be mentioned directly in the sentence rather than in brackets.

L132-133: Please reformulate this sentence: The EC measurements were processed to 30 min averages for evapotranspiration rate (mm h -1), [. . .], as half hourly values are not hourly values and please report it as actual evapotranspiration. By the way, I could found any results showing hourly actual evapotranspiration from EC-measurements in mm/h in the result section. Please show for the three events also half-hourly actual evapotranspiration in the Figure.

L152: It is not quite clear to me how the leaf water sample was taken. As this measurements are essential for the investigation I recommend to add some more sentences to clarify how the authors collected the water from the leaves and when (time before sunset, which is in summer already very early). What does it mean replicated fNRW samples? From where of the plant canopy sample were taken? Can the authors exclude from the form appearance of the water that it actual stems from guttation instead from dew formation? For event 2, bihourly samples were taken. Therefore, my question is, if the authors collected the water from the same leaves or from leaves of different plants during this event, which would make a difference for the collected water. Can the authors also say something on the plant species for which water was collected within each event and between the events?

L157: Not clear, what was measured here? [. . .] in soil moisture (hereafter $\delta$ s ). In addition I couldn't found anything on that measurements in the result section.

L166: Is it possible that the heating of the tube affected measurements?

L184: Please explain this more in detail

L207: Please add also here a statement about guttation, e.g. under the assumption that guttation did not occur during the events. . ..

[Figure]

L202:224: Please explain this more in detail what was done here to determine the four unknowns in the eq. 2-4.

L226: Please reformulate: [. . .] In unspecified explicit,[. . .]

L228: Could the authors also add the info why this type of regression was used here?

L247: add info where the reader can see this i.e. [. . .] levels (see Fig.xxx). I wonder why the authors do not show in addition to temperature and humidity the measured radiation variables from the EC-station.

L247-248: I recommend adding here the info that T0 was estimated and not measured.

L250: Was this before or after sunset for the specific event? Perhaps add text in Fig.5a and b that vertical lines shown are the times of sunset and sunrise. Also add in the figure caption what the vertical lines stands for.

L260: In the first event qa decrease is very low in comparison to the other events! This event was also with the month of the large rainfall deficits. Are there any estimate or measurements of NRW amount available? E.g., showing the measured latent heat flux from the EC-tower or lysimeter, leaf wetness sensor or estimates based on any model that predicts dew formation.

L271: Please explain the gaps in Fig. 6 a-d during P1b

L276: Please refer to $\delta$2Ha $\delta$18Oa here instead of $\delta$a

L288: Looking at Fig. 6 a and b, there a partially large difference between fNRW and aNRW especially for first and second, but also to some extent for the third event. The authors report later that much of the dew comes from the soil itself and not atmosphere so I would not expect that fNRW and aNRW are identical! Please describe results more carefully here and discuss it later.

L293: [. . .] The relationships between the isotopic compositions of fNRW and aNRW were related to RH [. . .] please add in Fig. 7 a, b, c the RH on the second y-axis. As it

is difficult to follow results until L300 without seeing measured isotopes and RH in one plot.

L302: Please explain the deviation of aNRW from the LMWL in Fig. 8. Does the position of aNRW below the LMWL means that aNRW stems from local ET water?

L306: but for event 2-sampled fNRW under ~97 to 98 are similar to that of aNRW. Others show large spread (deep purple triangles)? Is there no other reason that could explain the isotopic position of the samples fNRW that are much below the eq. line? E.g. nighttime evaporation processes of dew water on the leaf canopy. Would be good to check here the latent heat flux of the EC tower measurements for these times. Please add here for the discussion findings from Chen et al. (2019) (see Fig. 5), where data for soil, dewrise and dew water as well as vapor are shown.

L301-310: The authors mention in L157 the measurement of soil moisture (hereafter $\delta s$). I couldn't found a description of the data in the results section (already mentioned). Please add this here and describe it. This could clarify in Fig.8 where the water came from soil or evaporation of dew from canopy!

-we will add the isotopic compositions of soil moisture.

L301-310: another point might here that a mix of guttation water with dew might lead to a shift in the isotopic composition. It would at least fit as the deviation was seen for both events for the first sampled foliage water! Please at least mention it and discuss possible affects of guttation on stable isotope composition in the discussion e.g. see Xu et al. (2019).

L311-320: Not clear to me how the authors finally estimate the contribution of dew or soil distillation on the collected dew water, when the amount of dew, fog or from soil distillation are unknown for the events! This should be clearly describe in the M&M section.

L323: As $\delta$aNRW are simulations, the uncertainty of the used assumptions to determine $\delta$aNRW in the two end-member mixing model should be included and naNRW as

well as dDew with naNRW should be reported in the result section.

L323-329: these are results and the used method of e.g. Wen et al. 2012 should be describe in the Material and Method section and results should be shown in the results section!

L323-344: I recommend enlarging the discussion about the result here in a much broader context. Compare results with previous studies and discuss possible effects from e.g. guttation or dew re-evaporation on the sampled isotopic composition fNRW and the method on the partitioning of NRW inputs using a two end-member mixing model.

L351: Not sure about this reported values here. A) Please clarify how dDew was potentially 22-83% according to Monteiht 1957? B) Please report methods used here in the manuscript in the M&M section and not adding this info a Table caption (i.e. Table 2). C) More in detail, it is unclear how the authors come up with different times for dDew and aNRW. D) In addition, I recommend to use eq. mentioned in Monteith 1957 to calculate potential contribution of soil distillation and dew, present this result in the result section and compare it with the latent heat flux observations an than discuss it in this section 4.2.

L353: Not clear to me how NRW gain is comparable to average ET of 2.8 mm? There were no results on actual NRW water, the authors only report potential NRW+soil distillation which were somehow taken from report rates in Monteith 1957. At least soil distillation is soil dependent and also depends on the canopy or? If soil is bare we might see evaporation instead of soil distillation. This means reported values are location dependent!

L371-372: this values should be reported in the result section and added to the other plots to better distinguish from where water collected on leaves are coming from. From the M&M section it was also not clear how and when this samples were taken e.g. suction cups or destructive? Please add also this missing part in the M&M section

L380: From my perspective, the reported results (until now) are not a direct indicator that soil evaporation is synchronously happen with condensation. Please reformulate this in a more careful way. Perhaps it would be worth to calculate potential dew and soil distillation based on the eq. from Monteith 1957.

L389: Not clear to me why water that comes actually from the soil is not accessible for roots? Vapor transport might be largest during very dry conditions. However, this was only the case for first event. The other two event were observed during months with higher rainfall amounts than long-term average values.

L391: Please discuss somewhere that the amount of water transferred by vapor transport from soil depends on soil properties. L395: From which soil depths is this water coming from? Is this deeper than the effective rooting zone of the grassland? Would be an important point here to discuss, as only deeper than the roots zone located water would actual lead to a benefit of dDew for plants.

L401-403: the authors reported the estimated wilting point of the soil in the M&M section. Would be worth to mention this somewhere in the Results section to see if soil was actually near the wilting point during the 3 events, which would emphasize that NRW could reduce water stress during this time and discuss this point, e.g. Groh et al. (2018) that the occurrence of dew during times with water stress might alleviate drought stress for plant.

L416: My recommendation for this section is to present less individual results and to focus more on answering the question/objectives of the study and its impact in a broader context.

Chen, G., Sun, L.Z., Auerswald, K., 2019. Effects of Wilting and Dew on the Water Isotope Composition of Detached Grass in Temperate Grassland. Journal of Agricultural and Food Chemistry, 67(34): 9460-9467, 10.1021/acs.jafc.9b02978.

Groh, J., Slawitsch, V., Herndl, M., Graf, A., Vereecken, H., Pütz, T., 2018.

Determining dew and hoar frost formation for a low mountain range and alpine grassland site by weighable lysimeter. Journal of Hydrology, 563: 372-381, 10.1016/j.jhydrol.2018.06.009.

Xu, Y., Yi, Y., Yang, X., Dou, Y., 2019. Using Stable Hydrogen and Oxygen Isotopes to Distinguish the Sources of Plant Leaf Surface Moisture in an Urban Environment. Water, 11(11): 2287.
* * *

---

## Referee Comment (RC2) · Anonymous Referee #2 · 15 Dec 2020

Review of the manuscript hess-2020-493

The role of dew and radiation fog inputs in the local water cycling of a temperate grassland in Central Europe

by Yafei Li et al.

Summary This study investigates the role of fog and dew deposition in the water budget of a grassland in Switzerland. The authors aim to distinguish different pathways of the liquid water sources, e.g. fog deposition, dew deposition from the atmosphere to the surface, and dew deposition from the soil upwards towards the vegetation. The study uses isotopic composition of H and O in the water vapour in the atmosphere and the

liquid water. I think the authors did an tremendous effort in performing a measurement campaign to measure these components during three different nights and in understanding the pathways. This is also an interesting new approach. My main criticism about this manuscript is that the description of all the isotopic ratio's and compositions is written in a too much technical way. The reader is offered a number of values without interpretation what it mean related to the three proposed pathways. In the current shape the paper is only interesting for experts in isotopic signatures and does not serve the wider fog research community, while I think this huge research effort deserves this wider audience. More detailed comments have been listed below.

Recommendation: Major revisions required

Remarks

Ln 7: "In a warmer climate, non-rainfall water (hereafter NRW) formed from dew and fog potentially plays an increasingly important role in temperate grassland ecosystems under the scarcity of precipitation over prolonged periods". Please reword. I find this a confusing sentence, since warmer should be compared to a reference (warmer than....) and secondly I do not see the rationale that in a climate with high temperatures the relative contribution of occult precipitation will increase. Under climate change the hydrological cycle is expected to accelerate, which means more precipitation and thus less relative contribution by occult precipitation. Please rephrase.

Ln 11: remove "at all"

Ln 13 : the abstract misses a statement why isotopes are needed to identify the pathways. I would say that if I install eddy covariance, a fog collector and a microlysimeter, I can also obtain the mechanisms contributing to the NRW budgets. So motivate why a more difficult method is needed.

Ln 10-20: an interpretation should be provided what a certain permille for a certain isotope means. The reader is now overloaded with values without guidance about the

interpretation. In such a way the paper is only interesting for a small incrowd.

Ln 34-35: cite in chronological order, here and throughout the whole manuscript.

Ln 80-85: please add a few lines what are the physical reasons why local evaporation and entrainment at the PBL differ so much in d. This will help the non-involved reader.

Ln91: in height: please be more precise. Do you mean in the soil?

Ln 104: please specify in more detail the what is meant by ecological relevance and how you will measure that.

Section 2.2.1: please add which software was used for the flux processing and with which settings.

Ln 130: I am quite concerned about the height of the flux measurement since 2.4 m is very close to the surface, which means that there will be a relatively large "flux loss". Please specify how much this is and whether it will influence your results.

Ln 130: What happens to the contribution in the transport of the turbulence that happens below 2.4 m and is as such not seen by the EC sensor? Since the site is that the bottom of a valley I can imagine that thin katabatic flows are present from the valley walls to the valley and that they generate small scale turbulence. Does ignoring this component affect your conclusions. Please reflect and if possibly quantify.

Ln 142: The equation is incomplete. The upwelling LW_up flux consists of sigma*T^4 +(1-emiss)+LW_down and the latter component is missing. This would not have been a problem if the emissivity of the surface would equal 1, but you explicitly report it amounts to 0.98. Please recalculate your results.

Ln 199: why wasn't potential temperature gradient used for the PBL height determination?

Ln 200-202: I think it is this method should be reconsidered. The NBL depth can vary spatially enormously, especially in complex terrain where the experiment was done

(i.e. a valley) while the ECMWF product is at 30 km spatial resolution. Furthermore the vertical grid spacing of ECMWF is too coarse to detect the NBL height properly. Also the reported values are very high for nights where you can expect fog or dew. As a rule of thumb one can use that the NBL depth amounts to 700* u_star (friction velocity). That would mean that here the u_star would be 1 m/s and that is really really high for nights with fog or dew.

Ln 200-202: concerning Figure 3 I doubt whether the interpretation is correct since I think at the y axis the height above sea level is shown. The surface inversion should be at the surface (i.e. 0 m) right? Not at 650 m above ground level. This can also change the story about my previous point.

Ln 211: "while in saturated conditions, fNRW was a mix of aDew and aFog". I disagree on this since it is very hard to create fog in a night with a lot of dew at the same time. Dew takes out water vapour so fog in inhibited to develop. This contrasts with your statement.

Ln 213: typo: is -> as

Ln 222: It is good that you are honest about your assumption. But how realistic is the assumption. Could you spend a few words on it?

Ln 248: net longwave radiation loss: can you be more quantitative? Was it -80, -50 or -10 W/m2

Figure 5: the top of panel b can be at 12 of 15 g/kg.

Section 3.2-3.4 are hard to follow and only useful for specialists in isotope measurements. The numbers a presented as a flood of values without discussion or interpretation what they mean. I did not get so much from these sections.

Ln 354: "This amount of NRW gain was comparable with the average evapotranspiration rate of 2.8 mm day-1 (daytime) during ...". I do not understand what the authors want to say with this statement. How is dew at night comparable with evaporation

during the day. The mechanisms are completely different!

Ln 377: " minor influence of large-scale air advection": this is in complete contrast to the large diurnal cycle of specific humidity that is clearly driven by katabatic flows, as shown by the authors.

Ln 393: u-> u2m

Figure 10: I am not sure both panels are meaningful since in the definition of RH, the temperature plays an important role through the denominator in RH =q/q_sat(T). So I have the feeling we look twice at the same effect.

Formula B1: Perhaps I overlook something but I have the feeling that equation B1 is wrong when I compare it to Equation 3.19 in Campbell and Norman (1998). In CN98, the vapour concentration should be entered in mol/mol, but here in Pa. Please check, and check whether this affects your results.

---

## Author Comment (AC1) · 22 Dec 2020

Author comment on RC1 Anonymous Referee #1 The manuscript presents an interesting topic on non-rainfall water. The authors analyses for 3 events the water in the atmosphere and on the plants of an temperate grassland in Central Europe. The authors report data from a well-equipped test site and showed based on observation that dew formation and fog deposition are an overlooked part of the water cycle at such locations. The manuscript is overall well written, but the structure of the subchapter sometimes makes it difficult to follow the red line, and how this helps to answer the formulated aims of the manuscript. Several aspects in the manuscript require a further

improvement; clarification and especially a broader discussion of their results including results on the third objective (see further details in "General and Specific comments"). I want here to emphasize that it was a very interesting read and that the topic is of current interest for readership of HESS. I recommend a major revision and encourage the authors to carefully rewrite, revise and improve their manuscript.

→ → We thank the reviewer for her/his constructive comments (i.e., using Monteith (1957) equation to compute distillation rate, and adding more details in M&M) and positive feedback. We provide our answers point-by-point below.← ←

General comments: 1) A lot of subchapters and abbreviations makes it sometime difficult to follow the red line of the manuscript. I suggest restructuring the chapter/section in order to answer to aims/objectives of the investigation.

→ → we will remove third level titles in section 3.1; we will combine sections 3.2 and 3.3; we will combine part of section 4.1 into results section, and merge sections 4.1 and 4.3. we will try to remove some of the abbreviations (i.e., write them out) to make the reading more fluent. ← ←

2) I recommend adding a much broader discussion on the formation of NRW, including the parallel condensation of water by soil distillation and dew in the introduction.

→ → we will add some additional material on NRW formation pathways in the introduction. ← ←

3) NRW during prolong drought periods. Please use a common definition on the periods during the measurements e.g. term drought or hot days.

→ → we preferred "prolonged dry periods", because we addressed consecutive days without rainfall. ← ←

4) For the third objective, there is no data shown in the manuscript that could give new insight here in section results and authors only discuss potential impact of NRW on ecological functions. Please clarify by adding further points in section results and

describe how this was done (M&M section) that justifies the mention the third objective. E.g. the authors could include soil moisture observations during events (section result). Then discuss based on this results their ecological relevance in the corresponding discussion section.

→ → we will add data on the isotopic composition of soil water, and soil water content, and discuss them in relation to our third objective. ← ←

5) There is the need to show the latent heat flux measured with the EC-tower in this manuscript, which might help to clarify some points referring the observations on fNRW or dDew. It would be also helpful to see if EC-station can even indicate the formation of dew at night.

→ → we will add latent heat flux, although it tells the similar story as FH2O (Ed-dyPro_maual, Equation 5-101, 5-102). In calm dew nights, the uncertainty associated with EC measurements are large, because some of the underlying assumptions are not fulfilled. For example, in Figure4b, at around sunset in event 1, there was a downward flux, but condensation has not started yet (Figure5a, surface temperature had not cooled down below the dew point), so this might not be condensation, but the drainage of more humid air from aloft. Moreover, at around sunrise, there was a bigger download flux (Figure 4b), this might be entrainment from free troposphere. ← ←

6) Discussion on the outcomes of the results are very short and partially parts of 4.1 should be shift into result section.

→ → we will move part of section 4.1 into the section results. ← ←

7) Add result on potential NRW section into the results section and explain in addition the used methods in the M&M section.

→ → we will add the result on potential NRW into sections M&M and results following the methods in Monteith (1957). ← ←

Specific comments: L7: NRW is more than dew and fog. Thus, I recommend using

here in the text: [. . .] (hereafter NRW) mostly formed from dew and fog [. . .]

→ → we will rewrite our sentence. ← ←

L10: I recommend changing: condensation of soil-diffusing vapor to condensation of water vapor evaporating from the soil in the canopy (i.e. soil distillation), [. . .]. The processes described here by the authors is related to the term dewrise or soil distillation, whereby I recommend sticking with the latter term also use in Monteith (1957) within this manuscript.

→ → here we mean both capillary rise and gaseous transport of soil moisture, therefore we tried to get rid of using "evaporating". Especially, when soil moisture is very low, the gaseous transport is dominant (L399-400). ← ←

L22: [...] (2) of soil-diffusing vapor. Please clarify that water from soil distillation was not measured in this study, but was determine/assumed as end member.

→ → we will do as suggested. ← ←

L22: Please clarify the sentence why a potential of 0.06 − 0.39 mm per night are comparable to 2.8 mm daytime ET. Even after reading the entire manuscript it wasn0t clear to me how the authors came up to this statement and values.

→ → we will rewrite our sentence to make it more understandable. The aim with this statement was to put the obtained NRW input in relation to daytime ET to underline its importance in the diel near-surface moisture budget. ← ←

L28: [...] water deposition. I recommend to change it to: [...] water condensation and deposition. Please differentiate in the manuscript for condensation (dew) and fog (deposition).

→ → 'water deposition' includes both dew formation and fog droplet deposition. ← ←

L30: [...] (hereafter NRW) inputs, namely dew and fog. Please name first all possible contributor to NRW on the soil or canopy surface: dew, fog, water vapor adsorption,

soil distillation, and guttation.

→ → we will add all the possible types of NRW. We will mention that our study was conducted in the absence of precipitation, and low soil moisture availability, thus guttation occurring under high soil water content is not applicable in our case study. ← ←

L51: Please make clear that the authors refer here to the crop water use efficiency (WUE = ANPP/ET) as in other WUE definition only transpiration are used

→ → we will do as suggested. ← ←

L88: [...] onto foliage, [...]. Please change: onto the plant or soil surface.

→ → we will do as suggested. ← ←

L87-89: but water on plants can also stem from guttation. Please discuss this here and add also info on this at a later stage of the manuscript, how this might affect the results of the study, because water from this might be isotopically different, from other water sources in the plant-soil-atmospheric continuum (e.g. ambient water vapor, soil water, plant water).

→ → as answered for L30. ← ←

L92: delete After

→ → we will rewrite our sentence. ← ←

L103: [...] (3) assess the potential ecological relevance of NRW inputs. The authors report here observation for three events, but no further observations that could allow to make some statement on ecological relevance of NRW. I could not find any method used here to realize this in the Material & Method section and no results are reported within this manuscript on this point. Only in the discussion section, authors discuss potential impact of NRW on ecological functions! Please clarify the point that justifies this objective and how the authors answer this within the manuscript.

→ → we will add the effect of NRW inputs on the isotopic compositions of soil water. ← ←

L112-116: The authors report that rainfall amount in 2018 was 297 mm less than the long-term annual rainfall. In the next sentence, they report that during the 6 months period (April – September) the monthly rainfall, which was 81 mm, were reduced by 38% (49 mm). Something went wrong here, because 49 mm less rainfall per month (April-September) would mean a reduction of 60.5 % per month. I am also wondering that these reported values would mean that during the other months the rainfall was similar to the long terms values? As 6*49 = 294 mm and the total difference between 2018 and the long-term values was 297 mm.

→ → average level during 2006-2017 = 81+49=130mm, level in 2018 = 81 mm, 49 mm/130mm = 38%; 297 mm less is for the whole year, the calculated 294 mm is from April to September. ← ←

L117-118: The authors discuss their results in the light of a prolonged drought, but looking at the 3 measurement campaigns only the first event was within a month with less rainfall, because in August monthly rainfall was similar to long-term rainfall, and the monthly rainfall in September was with 130mm much larger as the long term mean rainfall (80 mm). If the authors want to relate their NRW results to term drought (especially important for the ecological relevance part), they need first to define this! Perhaps use better the definition of hot days during the extreme year 2018, instead of the term drought, as only the month in July showed a severe rainfall deficit and not the months August and September.

→ → we will use accumulated precipitation instead to better show the drought in 2018. The September precipitation was higher because of the heavy rain after our event 3. ← ←

L125-127: I recommend reformulating this sentence. The info in the brackets are larger than the rest of the sentence. Please change this for the whole manuscript, as relevant

information should be mentioned directly in the sentence rather than in brackets.

→ → we will rewrite our sentences. ← ←

L132-133: Please reformulate this sentence: The EC measurements were processed to 30 min averages for evapotranspiration rate (mm h -1), [...], as half hourly values are not hourly values and please report it as actual evapotranspiration. By the way, I could found any results showing hourly actual evapotranspiration from EC-measurements in mm/h in the result section. Please show for the three events also half-hourly actual evapotranspiration in the Figure.

→ → we will do as suggested. ← ←

L152: It is not quite clear to me how the leaf water sample was taken. As this measurements are essential for the investigation I recommend to add some more sentences to clarify how the authors collected the water from the leaves and when (time before sunset, which is in summer already very early). What does it mean replicated fNRW samples? From where of the plant canopy sample were taken? Can the authors exclude from the form appearance of the water that it actual stems from guttation instead from dew formation? For event 2, bihourly samples were taken. Therefore, my question is, if the authors collected the water from the same leaves or from leaves of different plants during this event, which would make a difference for the collected water. Can the authors also say something on the plant species for which water was collected within each event and between the events?

→ → droplets on leaf surfaces were taken in the nighttime. It was randomly sampled. It was the short-statured grassland with 10-20 cm of the vegetation height. We took triplicates from three species (Lolium sp. (long-narrow leaf, higher plants), Taraxacum sp. (long-wide leaf, shorter plants), and Trifolium spp. (short-wide leaf, some are shorter and others are higher), thus 9 replicates in total. There was no significant difference of the droplet samples from different species. We will give more details in M&M. As answered for L30, we will add data of soil water content, and exclude the

confusion of guttation. ← ←

L157: Not clear, what was measured here? [...] in soil moisture (hereafter $\delta$s ). In addition I couldn't found anything on that measurements in the result section.

→ → We will add isotopic composition of soil water. ← ←

L166: Is it possible that the heating of the tube affected measurements?

→ → we tested the effect of heating with tap water: raw tap water ($\delta$18O= -11.4 $\pm$ 0.1‰ $\delta$2H = -81.1 $\pm$ 0.9‰ $n = 9); tap water after vacuum extraction (\delta18O= -11.2 \pm 0.2$‰ $\delta$2H = -82.1 $\pm$ 1.8‰ $n = 9). There was no significant difference of \delta2H (1.0‰ $p > 0.05$) between raw tap water and extracted tap water, and the difference of \delta2H between raw tap water and extracted tap water was within measurement uncertainty of $\delta$2H (better than $\pm$1.0‰ L159) for IRMS. There was difference of $\delta$18O (0.2‰ $p < 0.05$) between raw tap water and extracted tap water, but much smaller than the observed \delta18O difference between fNRW and aNRW under unsaturated conditions (0.6‰ 0.9‰ and 0.3‰ for 03:00 CET of event 1, 23:00 and 01:00 CET of event 2 respectively, Fig.7a; Table 1). ← ←

L184: Please explain this more in detail

→ → We will add more details on how we calibrated the data. ← ←

L207: Please add also here a statement about guttation, e.g. under the assumption that guttation did not occur during the events....

→ → we will mention in section introduction that our events were in the absence of precipitation and low soil moisture availability to get rid of the confusion by guttation. ← ←

L202:224: Please explain this more in detail what was done here to determine the four unknowns in the eq. 2-4.

→ → we will give more details: with one time of sampling, we have 3 equations and

4 unknown values; with two times of sampling, we have 6 equations, and 8 unknown values; but we assumed $\delta18OdDew$ and $\delta2HdDew$ were constant for this two times of sampling, therefore, unknown values became 6, which can be solved with 6 equations. ← ←

L226: Please reformulate: [...] In unspecified explicit,[...]

→ → we will reformulate it. ← ←

L228: Could the authors also add the info why this type of regression was used here?

→ → we will add more details from Gat 1981 to explain this. Because $\delta18O$ and $\delta2H$ are always dependent each other, therefore orthogonal regression is recommended. ← ←

L247: add info where the reader can see this i.e. [...] levels (see Fig.xxx). I wonder why the authors do not show in addition to temperature and humidity the measured radiation variables from the EC-station.

→ → we will refer to the figures wherever needed. We will add radiation variables, although latent heat flux and evaporation rate tell the similar story as FH2O (linearly correlated with each other). ← ←

L247-248: I recommend adding here the info that T0 was estimated and not measured.

→ → we will note that T0 was computed values. ← ←

L250: Was this before or after sunset for the specific event? Perhaps add text in Fig.5a and b that vertical lines shown are the times of sunset and sunrise. Also add in the figure caption what the vertical lines stands for.

→ → Yes, we will add the legends that vertical dash-lines represent "sunset/sunrise". ← ←

L260: In the first event qa decrease is very low in comparison to the other events! This

event was also with the month of the large rainfall deficits. Are there any estimate or measurements of NRW amount available? E.g., showing the measured latent heat flux from the EC-tower or lysimeter, leaf wetness sensor or estimates based on any model that predicts dew formation.

→ → we will add latent heat flux, although FH2O has already shown the transition of evaporation and condensation. We do have micro-lysimeter and leaf wetness measurements, but unfortunately the micro-lysimeter and leaf wetness data was not available in 2018. But we will calculate condensation rate according to Monteith (1957) as suggested below. ← ←

L271: Please explain the gaps in Fig. 6 a-d during P1b

→ → we will note in Figure 6 that "gaps were calibration periods". ← ←

L276: Please refer to $\delta$2Ha, $\delta$18Oa here instead of $\delta$a

→ → we will do as suggested. ← ←

L288: Looking at Fig. 6 a and b, there a partially large difference between fNRW and aNRW especially for first and second, but also to some extent for the third event. The authors report later that much of the dew comes from the soil itself and not atmosphere so I would not expect that fNRWand aNRW are identical! Please describe results more carefully here and discuss it later.

→ → we will do as suggested. ← ←

L293: [...] The relationships between the isotopic compositions of fNRW and aNRW were related to RH [...] please add in Fig. 7 a, b, c the RH on the second y-axis. As it is difficult to follow results until L300 without seeing measured isotopes and RH in one plot.

→ → we will do as suggested. ← ←

L302: Please explain the deviation of aNRW from the LMWL in Fig. 8. Does the

position of aNRW below the LMWL means that aNRW stems from local ET water?

→ → yes, aNRW below the LMWL means that aNRW stems from local ET water. We will add the information as suggested. ← ←

L306: but for event 2-sampled fNRW under 97 to 98 are similar to that of aNRW. Others show large spread (deep purple triangles)? Is there no other reason that could explain the isotopic position of the samples fNRW that are much below the eq. line? E.g. nighttime evaporation processes of dew water on the leaf canopy. Would be good to check here the latent heat flux of the EC tower measurements for these times. Please add here for the discussion findings from Chen et al. (2019) (see Fig. 5), where data for soil, dewrise and dew water as well as vapor are shown.

→ → in L312-313 we mentioned, $\delta$18O and $\delta$2H of fNRW were higher and lower than aNRW respectively. Re-evaporation can occur, but should have caused both $\delta$18O and $\delta$2H of fNRW being higher. We will address this statement into discussion to make it more understandable. ← ←

L301-310: The authors mention in L157 the measurement of soil moisture (hereafter $\delta$s ). I couldn't found a description of the data in the results section (already mentioned). Please add this here and describe it. This could clarify in Fig.8 where the water came from soil or evaporation of dew from canopy!

→ → we will add the isotopic compositions of soil moisture. ← ←

L301-310: another point might here that a mix of guttation water with dew might lead to a shift in the isotopic composition. It would at least fit as the deviation was seen for both events for the first sampled foliage water! Please at least mention it and discuss possible affects of guttation on stable isotope composition in the discussion e.g. see Xu et al. (2019).

→ → we will address in section introduction that our events were in the absence of precipitation and low soil availability, and remove the confusion of guttation; but guttation

in discussion might distract readers from our storyline. ← ←

L311-320: Not clear to me how the authors finally estimate the contribution of dew or soil distillation on the collected dew water, when the amount of dew, fog or from soil distillation are unknown for the events! This should be clearly describe in the M&M section.

→ → we will calculate distillation rate following Monteith (1957), and add this in sections M&M and results. ← ←

L323: As aNRW are simulations, the uncertainty of the used assumptions to determine aNRW in the two end-member mixing model should be included and naNRW as well as dDew with naNRW should be reported in the result section.

→ → we split NRW under the assumption of equilibrium fractionation. We think include naNRW into results will distract readers from our storyline. But we will rearrange our structure. ← ←

L323-329: these are results and the used method of e.g. Wen et al. 2012 should be describe in the Material and Method section and results should be shown in the results section!

→ → as answered for L323. ← ←

L323-344: I recommend enlarging the discussion about the result here in a much broader context. Compare results with previous studies and discuss possible effects from e.g. guttation or dew re-evaporation on the sampled isotopic composition fNRW and the method on the partitioning of NRW inputs using a two end-member mixing model.

→ → we will add a broader discussion. ← ←

L351: Not sure about this reported values here. A) Please clarify how dDew was potentially 22-83% according to Monteiht 1957? B) Please report methods used here

in the manuscript in the M&M section and not adding this info a Table caption (i.e. Table 2). C) More in detail, it is unclear how the authors come up with different times for dDew and aNRW. D) In addition, I recommend to use eq. mentioned in Monteith 1957 to calculate potential contribution of soil distillation and dew, present this result in the result section and compare it with the latent heat flux observations an than discuss it in this section 4.2.

→ → as recommended, we will use equation in Monteith (1957) to estimate distillation rate. ← ←

L353: Not clear to me how NRW gain is comparable to average ET of 2.8 mm? There were no results on actual NRW water, the authors only report potential NRW+soil distillation which were somehow taken from report rates in Monteith 1957. At least soil distillation is soil dependent and also depends on the canopy or? If soil is bare we might see evaporation instead of soil distillation. This means reported values are location dependent!

→ → we will rewrite this part. ← ←

L371-372: this values should be reported in the result section and added to the other plots to better distinguish from where water collected on leaves are coming from. From the M&M section it was also not clear how and when this samples were taken e.g. suction cups or destructive? Please add also this missing part in the M&M section

→ → we will add the contents as suggested. ← ←

L380: From my perspective, the reported results (until now) are not a direct indicator that soil evaporation is synchronously happen with condensation. Please reformulate this in a more careful way. Perhaps it would be worth to calculate potential dew and soil distillation based on the eq. from Monteith 1957.

→ → we will calculate distillation rate as suggested. ← ←

L389: Not clear to me why water that comes actually from the soil is not accessible for

roots? Vapor transport might be largest during very dry conditions. However, this was only the case for first event. The other two event were observed during months with higher rainfall amounts than long-term average values.

→ → the events were during 4-5 consecutive days without precipitation. The confusion would be removed by adding accumulated precipitation and soil water content. ← ←

L391: Please discuss somewhere that the amount of water transferred by vapor transport from soil depends on soil properties.

→ → we will do as suggested. ← ←

L395: From which soil depths is this water coming from? Is this deeper than the effective rooting zone of the grassland? Would be an important point here to discuss, as only deeper than the roots zone located water would actual lead to a benefit of dDew for plants.

→ → we will add soil water content, and isotopic composition of soil water, and discuss more details. ← ←

L401-403: the authors reported the estimated wilting point of the soil in the M&M section. Would be worth to mention this somewhere in the Results section to see if soil was actually near the wilting point during the 3 events, which would emphasize that NRW could reduce water stress during this time and discuss this point, e.g. Groh et al. (2018) that the occurrence of dew during times with water stress might alleviate drought stress for plant.

→ → we will add soil water content, and isotopic composition of soil water. ← ←

L416: My recommendation for this section is to present less individual results and to focus more on answering the question/objectives of the study and its impact in a broader context.

→ → we will rewrite our conclusion. ← ←

Chen, G., Sun, L.Z., Auerswald, K., 2019. Effects of Wilting and Dew on the Water Isotope Composition of Detached Grass in Temperate Grassland. Journal of Agricultural and Food Chemistry, 67(34): 9460-9467, 10.1021/acs.jafc.9b02978. Groh, J., Slawitsch, V., Herndl, M., Graf, A., Vereecken, H., Pütz, T., 2018. Determining dew and hoar frost formation for a low mountain range and alpine grassland site by weighable lysimeter. Journal of Hydrology, 563: 372-381, 10.1016/j.jhydrol.2018.06.009. Xu, Y., Yi, Y., Yang, X., Dou, Y., 2019. Using Stable Hydrogen and Oxygen Isotopes to Distinguish the Sources of Plant Leaf Surface Moisture in an Urban Environment. Water, 11(11): 2287. Interactive comment on Hydrol. Earth Syst. Sci. Discuss., https://doi.org/10.5194/hess-2020- 493, 2020.

→ → Monteith, J. L.: Dew, Quarterly Journal of the Royal Meteorological Society, 83, 322-341, https://doi.org/10.1002/qj.49708335706, 1957. ← ←

---

## Author Comment (AC2) · 22 Dec 2020

Author comment on RC2

Anonymous Referee #2 Review of the manuscript hess-2020-493 The role of dew and radiation fog inputs in the local water cycling of a temperate grassland in Central Europe by Yafei Li et al. Summary This study investigates the role of fog and dew deposition in the water budget of a grassland in Switzerland. The authors aim to distinguish different pathways of the liquid water sources, e.g. fog deposition, dew deposition from the atmosphere to the surface, and dew deposition from the soil upwards towards the vegetation. The study uses isotopic

composition of H and O in the water vapour in the atmosphere and the liquid water. I think the authors did an tremendous effort in performing a measurement campaign to measure these components during three different nights and in understanding the pathways. This is also an interesting new approach. My main criticism about this manuscript is that the description of all the isotopic ratio's and compositions is written in a too much technical way. The reader is offered a number of values without interpretation what it mean related to the three proposed pathways. In the current shape the paper is only interesting for experts in isotopic signatures and does not serve the wider fog research community, while I think this huge research effort deserves this wider audience. More detailed comments have been listed below. Recommendation: Major revisions required

→ → We thank the reviewer for her/his constructive comments (i.e., clarifying the motivation why isotopic method is needed, and revising the manuscript for wider audience) and positive feedback. We provide our answers point-by-point below. ← ←

Remarks Ln 7: "In a warmer climate, non-rainfall water (hereafter NRW) formed from dew and fog potentially plays an increasingly important role in temperate grassland ecosystems under the scarcity of precipitation over prolonged periods". Please reword. I find this a confusing sentence, since warmer should be compared to a reference (warmer than....) and secondly I do not see the rationale that in a climate with high temperatures the relative contribution of occult precipitation will increase. Under climate change the hydrological cycle is expected to accelerate, which means more precipitation and thus less relative contribution by occult precipitation. Please rephrase.

→ → we will rewrite as suggested. We addressed that NRW is important during consecutive no-rain days. ← ←

Ln 11: remove "at all"

→ → we will rewrite as suggested. ← ←

Ln 13 : the abstract misses a statement why isotopes are needed to identify the pathways. I would say that if I install eddy covariance, a fog collector and a microlysimeter, I can also obtain the mechanisms contributing to the NRW budgets. So motivate why a more difficult method is needed.

→ → we will rewrite the statement why isotopes are needed.

On the one hand, micro-lysimeter can quantify the condensation from ambient water vapor, but cannot quantify distillation. Because distillation is the internal cycle from one part (soil) to the other (leaf surfaces) (Monteith, 1957). That is why we addressed that isotopic measurements could be combined with micro-lysimeter to quantify distillation amount if we know the mixing rate of distillation and condensation from ambient water vapor using isotopic splitting, and the condensation amount from ambient water vapor using micro-lysimeter. On the other hand, EC measurements are uncertain during calm nights (friction velocity u* ≤ 0.1 m s-1 (Jacobs et al., 2006)). As shown in Jacobs et al. (2006), dew amounts by EC measurement was much smaller than the values from micro-lysimeter. Similarly, during the three nights in our study with dew formation and radiation fog deposition, u* was smaller than 0.06 m s-1. Furthermore, as shown in Figure4b, FH2O showed an abrupt downward flux, but this might be cold air drainage instead of condensation, because surface has not cooled down below dew point as shown in Figure5a; abrupt downward flux was also observed at around sunrise, which might be entrainment from free troposphere. The uncertainty of EC measurements will be quantified with the energy budget closure following Eugster and Siegrist (2000). ← ←

Ln 10-20: an interpretation should be provided what a certain permille for a certain isotope means. The reader is now overloaded with values without guidance about the interpretation. In such a way the paper is only interesting for a small incrowd.

→ → we will interpret our results in a broader way. ← ←

Ln 34-35: cite in chronological order, here and throughout the whole manuscript.

→ → we will do as suggested. ← ←

Ln 80-85: please add a few lines what are the physical reasons why local evaporation and entrainment at the PBL differ so much in d. This will help the non-involved reader.

→ → we will do as suggested. d value decreased with stronger non-equilibrium evaporation because of the varied diffusive velocity for different water molecules (1H2H16O: 1H1H18O = 0.9723: 0.9755). Continental evaporation is mostly non-equilibrium fractionation process. Local (continental) evaporation experienced stronger evaporation as compared to entrainment from free troposphere, thus had lower d.

The value d = $\delta$2H–8* $\delta$18O; at equilibrium fractionation, $\Delta\delta$18O: $\Delta\delta$2H =1:8, hence d keeps rather constant; at non-equilibrium fractionation, $\Delta\delta$18O: $\Delta\delta$2H > 1:8, therefore evaporation would cause the decrease of d. ← ←

Ln91: in height: please be more precise. Do you mean in the soil?

→ → means a.g.l.; we will revise it. ← ←

Ln 104: please specify in more detail the what is meant by ecological relevance and how you will measure that.

→ → ecological relevance means the effect of NRW inputs on plants and soil moisture. We will revise in the next version. ← ←

Section 2.2.1: please add which software was used for the flux processing and with which settings.

→ → we will refer that eddypro processing was used. ← ←

Ln 130: I am quite concerned about the height of the flux measurement since 2.4 m is very close to the surface, which means that there will be a relatively large "flux loss". Please specify how much this is and whether it will influence your results.

→ → we will specify this as recommended. But "flux loss" would not affect our results,

because we used isotopes instead of EC data to quantify our results. As recommended by the first reviewer (RC1), we will use the equation by Monteith (1957) to calculate distillation rate, and then the condensation rate of ambient water vapor. This condensation rate from isotopic splitting will be compared with the condensation rate by EC measurement to analyze the uncertainty of EC measurements in dew and radiation fog nights. ← ←

Ln 130: What happens to the contribution in the transport of the turbulence that happens below 2.4 m and is as such not seen by the EC sensor? Since the site is that the bottom of a valley I can imagine that thin katabatic flows are present from the valley walls to the valley and that they generate small scale turbulence. Does ignoring this component affect your conclusions. Please reflect and if possibly quantify.

→ → we will quantify the effect of katabatic flows using energy budge closure following Eugster and Siegrist (2000). ← ←

Ln 142: The equation is incomplete. The upwelling LW_up flux consists of sigma*TËȨ4 +(1-emiss)+LW_down and the latter component is missing. This would not have been a problem if the emissivity of the surface would equal 1, but you explicitly report it amounts to 0.98. Please recalculate your results.

→ → we used longwave-outgoing radiation instead of LW_up here, therefore no LW_down is needed here. But we will recheck this. ← ←

Ln 199: why wasn't potential temperature gradient used for the PBL height determination?

→ → we will use potential temperature gradient instead. ← ←

Ln 200-202: I think it is this method should be reconsidered. The NBL depth can vary spatially enormously, especially in complex terrain where the experiment was done (i.e. a valley) while the ECMWF product is at 30 km spatial resolution. Furthermore the vertical grid spacing of ECMWF is too coarse to detect the NBL height properly. Also

the reported values are very high for nights where you can expect fog or dew. As a rule of thumb one can use that the NBL depth amounts to 700* u_star (friction velocity). That would mean that here the u_star would be 1 m/s and that is really really high for nights with fog or dew.

→ → we will use COSMO model instead. The resolution is 4 km (meridional) × 6 km (zonal) over Switzerland (Westerhuis et al., 2020). ← ←

Ln 200-202: concerning Figure 3 I doubt whether the interpretation is correct since I think at the y axis the height above sea level is shown. The surface inversion should be at the surface (i.e. 0 m) right? Not at 650 m above ground level. This can also change the story about my previous point.

→ → There was a mistake in the computation of the vertical height. The ECMWF model will be replaced by COSMO model as answered above. ← ←

Ln 211: "while in saturated conditions, fNRW was a mix of aDew and aFog". I disagree on this since it is very hard to create fog in a night with a lot of dew at the same time. Dew takes out water vapour so fog in inhibited to develop. This contrasts with your statement.

→ → We stated in L58-64 that this is radiation fog. As shown in Figure4d, intermittent radiation fog occurred at our site. Not only events 2 and 3, combined dew and radiation fog is often observed at the CH-CHA site. It is true that dew takes out water vapor from near surface atmosphere (Figure5b), but both air temperature and surface temperature cooled down (Figure5a). This causes an increase of relative humidity at surface temperature (Figure4c).

The inhibition of dew on fog might be true in the first hour of dew, as mentioned by Monteith (1957). Because "sufficient latent heat would be released to raise the temperature of the leaves above the dew point, preventing condensation until further cooling had taken place." But with the further cooling down of surface temperature and air temperature, the latent heat did not warm the temperature above dew-point (Figure5a). ←
←

Ln 213: typo: is -> as

→ → we will change as suggested. ← ←

Ln 222: It is good that you are honest about your assumption. But how realistic is the assumption. Could you spend a few words on it?

→ → we will give more statement as suggested. We will calculate distillation rate following Monteith (1957), and then the condensation rate will be calculated from splitting ratio, which will be compared with previous research. ← ←

Ln 248: net longwave radiation loss: can you be more quantitative? Was it -80, -50 or -10 W/m2 Figure 5: the top of panel b can be at 12 of 15 g/kg. Section 3.2-3.4 are hard to follow and only useful for specialists in isotope measurements. The numbers a presented as a flood of values without discussion or interpretation what they mean. I did not get so much from these sections.

→ → we will revise as suggested, and restructured our results and discussion. ← ←

Ln 354: "This amount of NRW gain was comparable with the average evapotranspiration rate of 2.8 mm day-1 (daytime) during ...". I do not understand what the authors want to say with this statement. How is dew at night comparable with evaporation during the day. The mechanisms are completely different!

→ → we want to give general concept how much is this NRW inputs, but we will rewrite to get rid of confusion. ← ←

Ln 377: " minor influence of large-scale air advection": this is in complete contrast to the large diurnal cycle of specific humidity that is clearly driven by katabatic flows, as shown by the authors.

→ → we will clarify this point. We mean the synoptic-scale flow has a minor influence because of the anticyclonic influence. Katabatic flows are density driven flows of mesoscale extent, induced by the local topography and the regional thermodynamic conditions in a situation with weak large-scale influence. ← ←

Ln 393: u-> u2m Figure 10: I am not sure both panels are meaningful since in the definition of RH, the temperature plays an important role through the denominator in RH =q/q_sat(T). So I have the feeling we look twice at the same effect. Formula B1: Perhaps I overlook something but I have the feeling that equation B1 is wrong when I compare it to Equation 3.19 in Campbell and Norman (1998). In CN98, the vapour concentration should be entered in mol/mol, but here in Pa. Please check, and check whether this affects your results.

→ → we will revise our figures, and wind speed abbreviation. - no units for both our equation B1 and Equation 3.19 in Campbell and Norman (1998), it is just ratio. In supplement we showed how we rewrite this equation. ← ←

→ →

Eugster, W. and Siegrist, F.: The influence of nocturnal $CO_2$ advection on $CO_2$ flux measurements, Basic and Applied Ecology, 1, 177-188, https://doi.org/10.1078/1439-1791-00028, 2000.

Jacobs, A. F. G., Heusinkveld, B. G., Kruit, R. J. W., and Berkowicz, S. M.: Contribution of dew to the water budget of a grassland area in the Netherlands, Water Resources Research, 42, https://doi.org/10.1029/2005WR004055, 2006.

Monteith, J. L.: Dew, Quarterly Journal of the Royal Meteorological Society, 83, 322-341, https://doi.org/10.1002/qj.49708335706, 1957.

Westerhuis, S., Fuhrer, O., Cermak, J., and Eugster, W.: Identifying the key challenges for fog and low stratus forecasting in complex terrain, Quarterly Journal of the Royal

[Figure]

Meteorological Society, 146, 3347-3367, https://doi.org/10.1002/qj.3849, 2020.

← ←

Please also note the supplement to this comment:
https://hess.copernicus.org/preprints/hess-2020-493/hess-2020-493-AC2-
supplement.pdf

––––––––––––––––––––––––––

[Figure]

**Supplement:**

Ln 393: u-> u2m

Figure 10: I am not sure both panels are meaningful since in the definition of RH, the temperature plays an important role through the denominator in RH =q/q_sat(T). So I have the feeling we look twice at the same effect.

Formula B1: Perhaps I overlook something but I have the feeling that equation B1 is wrong when I compare it to Equation 3.19 in Campbell and Norman (1998). In CN98, the vapour concentration should be entered in mol/mol, but here in Pa. Please check, and check whether this affects your results.

$\rightarrow$ $\rightarrow$ we will revise our figures, and wind speed abbreviation.

- no units for both our equation B1 and Equation 3.19 in Campbell and Norman (1998), it is just ratio. Here we showed how we rewrite this equation:

$$q_0 = \frac{0.622 \cdot C_{va}}{1 - 0.378 \cdot C_{va}} = \frac{0.622 \cdot \frac{e_{s0}}{p}}{1 - 0.378 \cdot \frac{e_{s0}}{p}} = \frac{0.622 \cdot e_{s0}}{p - 0.378 \cdot e_{s0}} \left(\frac{kg}{kg}\right) = \frac{0.622 \cdot e_{s0}}{p - 0.378 \cdot e_{s0}} \left(\frac{kg}{kg}\right) * 1000 \left(\frac{g}{kg}\right)$$

$$= \frac{622 \cdot e_{s0}}{p - 0.378 \cdot e_{s0}} \left(\frac{g}{kg}\right)$$

---

## Author Response (AR1)

**Editor Remarks**

Your manuscript has been reviewed by two anonymous referees, both of which I consider experts in this field. Both referees agree in their generally positive assessment of the focus and quality of the data and analysis. However they also raise a number of issues related to the presentation and interpretation that need to be addressed before the manuscript can be considered for final publication. I agree with the referees that this is an important study, but the work needs to be accessible to a broader audience in order to have an impact, and to fit within the scope of HESS (HESS aims to serve not only the hydrological science community but all earth and life scientists). I classify these as major, mainly because the referees suggest to include some additional analysis. However based on your response I think the revision should be doable, and I generally agree with your answers. I am looking forward to receiving a revised version of your manuscript, and please feel free to contact me in case of questions in the meantime.

We thank both Reviewers and the Editor for their critical but constructive assessment. We think that we could address the most critical aspects and provide our response to the critique in blue color after each Reviewer or Editor comments. In the very few minor aspects where we cannot agree we provide our viewpoint. We hope that with these revisions our manuscript can now be accepted for publication in HESS. We appreciate the time and care that the Reviewers have taken and thus have substantially revised our manuscript. We also expanded the text to address the more general readership of HESS as noted by the Editor and Reviewers.

**The main changes of the revised version are as follow:**

[1]. As suggested by Reviewer #1, we reduced the abbreviations in our manuscript. "dDew" and "dFog" were both called "aNRW" in the new version; "dis" was changed to its full name "distillation". See L320-324, and Fig. 2.

[2]. To assess the reasonability of the results for our two end-member mixing model, and also as suggested by reviewer #1, we added the method of Monteith (1957) (M57 approach in the revised version) to estimate the contribution of distillation, which was close to the contribution of distillation by our mixing model. See in Methods Sections 3.2.4, and 3.2.5; in Results Section 4.4, and Discussion Sections 5.2 and 5.3; Fig. 10.

[3]. We added the Background Section 2 to provide some basic information on stable isotopes for non-isotopic readers (Section 2.1), and explain why we did not consider guttation in our two end-member mixing model (Section 2.2).

[4]. We added more information on the uncertainty of eddy covariance measurement in dew and fog nights; see the Introduction Section 1 at L84-94.

[5]. Instead of assuming roughness length as 0.1 times the vegetation height, we now calculate the roughness length $z_0$ from eddy-covariance (EC) measurement under neutral stability during the nights. But this did not dramatically change our results (change from 0.02–0.03 m to 0.03 m). See details in Section 3.2.3 (L300-306) and Appendix D (L1116-1126).

[6]. Instead of reporting monthly precipitation which caused the confusion that August and September 2018 were not dry periods, we plotted the year-to-date precipitation (Fig. 1a; L215-217) and volumetric soil water content (and wilting point, field capacity, and rooting depths; Fig. 1d; L224-229) to better show the dry conditions of the corresponding three events.

[7]. We removed the third-level section titles in Results Section 4.1 to make it more readable.

[8]. We merged the interpretation of isotopic signals (former Discussion Section 4.1) into new Results Section 4.3 to make it easier to be followed by non-isotopic readers.

[9]. We added the schematics of temperature profile into Fig. 1.

[10].    As commented by Reviewer #2, we replaced the ECMWF model with the much finer resolved COSMO model to interpret the top of nocturnal boundary layer (see Fig. 3).

[11].    We replaced the $H_2O$ flux with latent heat flux as suggested by Reviewer #1. Basically $H_2O$ flux, LE and ET show the same information in different units, so we eliminated $H_2O$ flux and only kept LE (see Fig. 4b), but also ET which has the same units as the estimates of dew and fog water gains during the nights. We hope that this reduces the confusion that obviously concerned Reviewer #1.

[12].    We added the long-wave outgoing radiation to Fig. 5a to better show the long-wave radiation loss during surface cooling period; we also added the air temperature at 1 cm a.g.l. (computed from soil temperature $T_{s1cm}$ and grassland surface temperature $T_{0w}$ using Eq. 20, and which was for calculating distillation rate in Eq. 19) into Fig. 5c.

[13].    We removed former Table 2, and the corresponding results were moved to new Fig. 10a.

[14].    We removed former Table 3, and the corresponding results were moved to new Fig. 1c.

[15].    We added new Table 2 for comparing the among-species difference of leaf water isotopic composition and non-rainfall water isotopic composition on foliage.

**Reply to RC1**

Anonymous Referee #1

The manuscript presents an interesting topic on non-rainfall water. The authors analyses for 3 events the water in the atmosphere and on the plants of an temperate grassland in Central Europe. The authors report data from a well-equipped test site and showed based on observation that dew formation and fog deposition are an overlooked part of the water cycle at such locations. The manuscript is overall well written, but the structure of the subchapter sometimes makes it difficult to follow the red line, and how this helps to answer the formulated aims of the manuscript. Several aspects in the manuscript require a further improvement; clarification and especially a broader discussion of their results including results on the third objective (see further details in "General and Specific comments"). I want here to emphasize that it was a very interesting read and that the topic is of current interest for readership of HESS. I recommend a major revision and encourage the authors to carefully rewrite, revise and improve their manuscript.

We thank the reviewer for her/his constructive comments and positive feedback. We provide our answers point-by-point below.

General comments:

1) A lot of subchapters and abbreviations makes it sometime difficult to follow the red line of the manuscript. I suggest restructuring the chapter/section in order to answer to aims/objectives of the investigation.

➢ We added Background Section 2 to improve the story line of our paper.
➢ We removed the third level sections in the Results section, and merged the environmental conditions into one second-level section (Section 4.1; L377-415).
➢ The former Discussion subsection 4.1 "4.1 Fractionation during condensation of ambient water vapor " (former version) was moved to the Results section as new subsection 4.3 (L445-504).

2) I recommend adding a much broader discussion on the formation of NRW, including the parallel condensation of water by soil distillation and dew in the introduction.

➢ We added Section 5.2 "5.2 Processes affecting non-rainfall water on foliage" for a broader discussion on the formation of NRW (NRW from ambient water vapor, distillation, guttation, and re-evaporation; L573-591).
➢ We added the measurement or quantification of NRW from ambient water vapor and distillation in the Introduction section (L84-128); the guttation effect is now described in the Background section 2.2 (L159-172).

3) NRW during prolong drought periods. Please use a common definition on the periods during the measurements e.g. term drought or hot days.

➢ We changed "prolonged dry periods" into "dry spells" (L7). Here we address the consecutive rainless days.

4) For the third objective, there is no data shown in the manuscript that could give new insight here in section results and authors only discuss potential impact of NRW on ecological functions. Please clarify by adding further points in section results and describe how this was done (M&M section) that justifies the mention the third objective. E.g. the authors could include soil moisture observations during events (section result). Then discuss based on this results their ecological relevance in the corresponding discussion section.

> The third aim is actually more a long-term perspective/goal of our research. In this paper we make a first step in that direction, and thus we agree that this can be misinterpreted by the readers. We therefore decided to remove the third aim from this paper (L129-130) in the form of a specific aim of this one manuscript, and now mention our overarching goal in a more general way at the end of the Introduction (L133–134). We nevertheless added the data of soil water content (Fig. 1d) and the isotopic composition of leaf water (Table 2, new version) and soil water (new Fig. 9) for discussing subsections "5.2 Processes affecting non-rainfall water on foliage (L573-591)", and "5.4 Potential effects of non-rainfall water on local water cycling" (L637-L662). The samplings of leaf water and soil water were added in Methods Section 3.2.1 (L236-241).

5) There is the need to show the latent heat flux measured with the EC-tower in this manuscript, which might help to clarify some points referring the observations on fNRW or dDew. It would be also helpful to see if EC-station can even indicate the formation of dew at night.

> We replaced the $H_2O$ flux by latent heat flux (Fig. 4b). These variables essentially provide the same information but in different physical units with the H2O flux representing a water vapor flux and the latent heat flux representing the corresponding energy flux (EddyPro_manual: https://www.licor.com/documents/1ium2zmwm6hl36yz9bu4, Equation 5-101, 5-102, see screenshot below).

$$F_{3,h_2o} = E_3 \cdot 10^{-3} \cdot M_{h_2o} \qquad \text{5-101}$$

$$LE_3 = \lambda \cdot E_3 \qquad \text{5-102}$$

> In calm dew nights, the uncertainty associated with EC measurements are large, because some of the underlying assumptions are not fulfilled, and in this specific case we used an open-path instrument that becomes inaccurate when fog occurs or dew drips to the optical window of the sensor.

6) Discussion on the outcomes of the results are very short and partially parts of 4.1 should be shift into result section.

> The former Discussion section 4.1 was merged into the Results section as new section 4.3. The outcomes were discussed in new Sections 5.2 and 5.3.

7) Add result on potential NRW section into the results section and explain in addition the used methods in the M&M section.

> The NRW amount is now computed using the method of Monteith (1957) (M57 approach), see Methods, new section 3.2.5; in Results, section 4.4 (L513-517).

**Specific comments:**

L7: NRW is more than dew and fog. Thus, I recommend using here in the text: […] (hereafter NRW) mostly formed from dew and fog […]

> All the NRW components are now listed in the Introduction (L44-47).
NRW inputs include a number of components: (1) dew formation (Monteith, 1957); (2) fog deposition (Dawson, 1998); (3) water vapor adsorption (Agam and Berliner, 2006); (4) rime ice deposition (Hindman et al., 1983); (5) hoar frost(Monteith and Unsworth, 2013); and (6) guttation (Long, 1955).

L10: I recommend changing: condensation of soil-diffusing vapor to condensation of water vapor evaporating from the soil in the canopy (i.e. soil distillation), […]. The processes described here by the authors is related to the term dewrise or soil distillation, whereby I recommend sticking with the latter term also use in Monteith (1957) within this manuscript.

➢ We simplified this statement by using the term "distillation of water vapor from soil" (L10) as suggested by Monteith (1957) here and replaced "condensation of soil vapor diffusion" by the same term throughout the text, where correct and appropriate. In cases where we emphasis the concentration of vapor (a) from the ambient atmosphere in combination with (b) from the soil (which is considered by the term "distillation"), we kept the wording "from ambient water vapor and soil-diffusing vapor".

L22: [...] (2) of soil-diffusing vapor. Please clarify that water from soil distillation was not measured in this study, but was determine/assumed as end member.

➢ We added that distillation is meant as computed from two end-member mixing model: "We employed a simple two end-member mixing model using $\delta^{18}O$ and $\delta^2H$ to quantify the NRW inputs from these two different sources." (L22-23).

L22: Please clarify the sentence why a potential of 0.06 – 0.39 mm per night are comparable to 2.8 mm daytime ET. Even after reading the entire manuscript it wasn't clear to me how the authors came up to this statement and values.

➢ The idea of this statement was to provide an overall local water budget perspective and compare the NRW input to the canopy during nights with the loss due to ET. We agree with the reviewer that the wording was confusing, the ET estimate is from the entire day, including the night. To clarify, we reworded this statement as "The dew and radiation fog potentially produced 0.10–0.41 mm $d^{-1}$ NRW gain on foliage, thereby constituting a non-negligible water flux to the canopy, as compared to the daily evapotranspiration of 2.8 mm $d^{-1}$." (L25-26). Please note that our estimate has slightly increased in the revisions using Monteith (1957) equation.

L28: [...] water deposition. I recommend to change it to: [...] water condensation and deposition. Please differentiate in the manuscript for condensation (dew) and fog (deposition).

➢ To clarify and better relate this statement to the title of our manuscript, we reworded to "…including different pathways of dew and radiation fog water inputs." (L26–28).

L30: [...] (hereafter NRW) inputs, namely dew and fog. Please name first all possible contributor to NRW on the soil or canopy surface: dew, fog, water vapor adsorption, soil distillation, and guttation.

➢ We added all the NRW components in the Introduction (L44-47), but to simplify legibility of the text for the wider audience of HESS we replaced the explicity reference to dew and fog in the statement that this reviewer refers to, which now reads: "…non-rainfall water (hereafter NRW) inputs from various sources (see below) may become essential for the vegetation…" (L36-38).

L51: Please make clear that the authors refer here to the crop water use efficiency (WUE = ANPP/ET) as in other WUE definition only transpiration are used

➢ We added the definition of WUE used in this context: "(crop WUE, defined as gross carbon uptake per unit water lost)" (L68).

L88: [...] onto foliage, [...]. Please change: onto the plant or soil surface.

➢ We think distillation is the condensation of soil-diffusing vapor onto foliage but not at the soil surface, because soil surface is warmer than the adjacent air (e.g., 1 cm air) if there is vegetation covering the ground, but foliage is cooler than the adjacent air (1

cm air). This is the basic assumption that Monteith (1957) makes with his Eq. (3): that the ground surface temperature $T(0)$ is warmer than the $T(1)$ temperature in the air at 1 cm above ground. His Table 3 provides the corresponding data that show that $T(0)$ was always warmer than $T(1)$, and we adhere to this concept here.

L87-89: but water on plants can also stem from guttation. Please discuss this here and add also info on this at a later stage of the manuscript, how this might affect the results of the study, because water from this might be isotopically different, from other water sources in the plant-soil-atmospheric continuum (e.g. ambient water vapor, soil water, plant water).

➢ Guttation is now montioned in the Background Section 2.2.

L92: delete After

➢ We replaced "After" with "Since" (L100) because the statement should remain "Since Monteith (1957)… research has rarely focused on …".

L103: [...] (3) assess the potential ecological relevance of NRW inputs. The authors report here observation for three events, but no further observations that could allow to make some statement on ecological relevance of NRW. I could not find any method used here to realize this in the Material & Method section and no results are reported within this manuscript on this point. Only in the discussion section, authors discuss potential impact of NRW on ecological functions! Please clarify the point that justifies this objective and how the authors answer this within the manuscript.

➢ The third objective is actually the overarching goal of our research rather than the objective of this specific paper. Therefore we removed the third objective in this paper and phrased it a more general way to clarify (L129-134).

L112-116: The authors report that rainfall amount in 2018 was 297 mm less than the long-term annual rainfall. In the next sentence, they report that during the 6 months period (April – September) the monthly rainfall, which was 81 mm, were reduced by 38% (49 mm). Something went wrong here, because 49 mm less rainfall per month (April-September) would mean a reduction of 60.5 % per month. I am also wondering that these reported values would mean that during the other months the rainfall was similar to the long terms values? As 6*49 = 294 mm and the total difference between 2018 and the long-term values was 297 mm.

➢ We apologize for the unclear wording. The actual calculation was: average level during 2006-2017 = 81 + 49 = 130 mm, level in 2018 = 81 mm, 49 mm/130mm = 38%; 297 mm less is for the whole year, the calculated 294 mm is from April to September. But this feedback showed us, that we should better only report conditions before our events, because whatever precipitation was recorded after eacn vent did not have an influence on the respective events. Thus, in the revised version, we change the monthly precipitation into "year-to-date precipitation" before the three events (Fig. 1a; L214-217) to better show the dry conditions during our three events.

L117-118: The authors discuss their results in the light of a prolonged drought, but looking at the 3 measurement campaigns only the first event was within a month with less rainfall, because in August monthly rainfall was similar to long-term rainfall, and the monthly rainfall in September was with 130mm much larger as the long term mean rainfall (80 mm). If the authors want to relate their NRW results to term drought (especially important for the ecological relevance part), they need first to define this! Perhaps use better the definition of hot days during the extreme year 2018, instead of the term drought, as only the month in July showed a severe rainfall deficit and not the months August and September.

➢ This point was well taken. In the revised version, we changed the monthly precipitation information to "year-to-date precipitation" before the three events (Fig. 1a; L214-217) to better show the dry conditions during our three events.

L125-127: I recommend reformulating this sentence. The info in the brackets are larger than the rest of the sentence. Please change this for the whole manuscript, as relevant information should be mentioned directly in the sentence rather than in brackets.

➢ The changes are given in L224-229. Moreover, throughout the text we incorporated the text in parentheses into the text where this was increasing legibility of the text. See e.g. L99–100, 121–122, 223.

L132-133: Please reformulate this sentence: The EC measurements were processed to 30 min averages for evapotranspiration rate (mm h -1), [...], as half hourly values are not hourly values and please report it as actual evapotranspiration. By the way, I could found any results showing hourly actual evapotranspiration from EC-measurements in mm/h in the result section. Please show for the three events also half-hourly actual evapotranspiration in the Fig..

➢ We reworded to "The EC measurements at 20 Hz were processed to 30 min averages …Evapotranspiration (ET in mm $h^{-1}$) was derived from LE (see Appendix B)" to clarify this point (L179-185).
➢ We also added information on how ET was calculated from LE, see Appendix B: Eq. B5 (L1098).

L152: It is not quite clear to me how the leaf water sample was taken. As this measurements are essential for the investigation I recommend to add some more sentences to clarify how the authors collected the water from the leaves and when (time before sunset, which is in summer already very early). What does it mean replicated fNRW samples? From where of the plant canopy sample were taken? Can the authors exclude from the form appearance of the water that it actual stems from guttation instead from dew formation? For event 2, bihourly samples were taken. Therefore, my question is, if the authors collected the water from the same leaves or from leaves of different plants during this event, which would make a difference for the collected water. Can the authors also say something on the plant species for which water was collected within each event and between the events?

➢ We modified the text accordingly, which now reads: "NRW droplets on foliage (fNRW) were absorbed in triplicates with cotton balls from the leaf surfaces of *Lolium sp*. with long and narrow leaves, and taller vegetation; *Taraxacum sp*. with long and wide leaves, and shorter vegetation; and *Trifolium spp*. with short and wide leaves, and both shorter and taller vegetation. The fNRW samples were taken at the end of the nights of events 1 and 3 (once sampling per event), but bi-hourly during the night of event 2 (i.e., four times of sampling in event 2). Simultaneously, leaf samples were taken in triplicates for the three species after softly drying the leaf surfaces with tissue paper." (Secion 3.2.1 in L232-237).
➢ The sampling time was the corresponding time of fNRW values as shown in Fig. 7.
➢ Among-species difference was insignificant in NRW samples, but was significant in leaf samples (Table 2); guttation is now discussed at L573-591.

L157: Not clear, what was measured here? [...] in soil moisture (hereafter δs ). In addition I couldn't found anything on that measurements in the result section.

➢ The information for the isotopic composition of soil water is added in Methods Section 3.2.1 (L237-241), and Fig. 8; in Results Section 4.3 (L468-472, L482-483); in Discussion, Section 5.1 (L543-545), Section 5.3 (L620-623), Section 5.4 (L647-649, L652-655).

L166: Is it possible that the heating of the tube affected measurements?

➢ The inlet tube is heated to prevent condensation in the tubing and minimize the response time of the inlet system (L256, see also detailed discussion in Sturm and Knohl (2010) and Aemisegger et al. (2012)), thus heating must affect the measurements in a positive way, i.e. reducing the error associated with sampling, that is the goal of heating an inlet tube. We are not aware of any

negative effects that heating would have on the measurements, because the cavity temperature of the Picarro L2130 laser spectrometer is controlled at 80°C, thus if the inlet is heated, the Picarro instrument uses less heat to bring the sample to the desired (noncondensing) temperature inside the cavity. Therefore, preheating the sample gas can only be beneficial.

L184: Please explain this more in detail

5 ➢ We rephrased this to include the details in Section 3.2.2. See L279-296.

L207: Please add also here a statement about guttation, e.g. under the assumption that guttation did not occur during the events....

➢ Guttation is now mentioned in Background Section 2.2. We however are not convinced that guttation plays a role in our investigation and phrased this at the end of Section 2: " In all our samples, however, the isotopic composition of dew water was not related to the plant species from the surfaces of which the water was collected, which allowed us to exclude guttation as a
10 relevant process during dry-spell periods." (L170–172). Hence, no further emphasis on guttation was added in the remainder of the manuscript.

L202:224: Please explain this more in detail what was done here to determine the four unknowns in the eq. 2-4.

➢ See Section 3.2.4 at L319-346. We rewrote the former Eqs. 2-4 into new Eqs. 10-12, and the two-point equations Eqs. 13-18.

L226: Please reformulate: [...] In unspecified explicit,[...]

15 ➢ See L388: "We report means ± SD, unless specified differently.".

L228: Could the authors also add the info why this type of regression was used here?

➢ We now use orthogonal regression for all the linear regression (L374-375). Least-square orthogonal regression is a better approach for $\delta 18O$ and $\delta 2H$ regression (Gat (1981), page 123).

> In the development of a linear relationship between $\delta^{18}O$ and $\delta D$ and in its use for hydrological interpretations, the common practice, so far, has been to designate $\delta D$ as the dependent variable and $\delta^{18}O$ as the independent one. In view of the fact that the analytical errors involved in $\delta^{18}O$ and $\delta D$-data are usually comparable relative to the variations of the two isotopes, a linear regression based upon least-squared orthogonal deviations would be a better approach (reduced major axis line, Kermack and Haldane, 1950).

20 ➢ Orthogonal regression assumes that uncertainties exist in both the x- and y-variable, whereas the ordinary least-squares regression assumes that there is no uncertainty in the x-values, and all uncertainty is found in the y-values, which is not the case here for any of our observations.

L247: add info where the reader can see this i.e. [...] levels (see Fig.xxx). I wonder why the authors do not show in addition to temperature and humidity the measured radiation variables from the EC-station.

25 ➢ LE is now added in Fig. 4b replacing former $F_{H2O}$. Fig. 4 citation is added (L378-381). The measured radiation variables from the EC-station are now shown in Fig. 5a (long-wave outgoing radiation $LW_{out}$), and Fig. 11 (sensible heat flux H, ground heat flux G, net radiation Rn).

L247-248: I recommend adding here the info that $T_0$ was estimated and not measured.

➤ We changed this sentence to: "… the vegetation surface temperature $T_0$ derived from radiation measurement remained cooler than air temperature $T_{a2m}$ at 2 m a.g.l., although both gradually decreased (Fig. 5b)." (L392-394).

L250: Was this before or after sunset for the specific event? Perhaps add text in Fig.5a and b that vertical lines shown are the times of sunset and sunrise. Also add in the Figure caption what the vertical lines stands for.

➤ The nighttime period is now clearly indicated in all the related Figs (Figs 4, 5, 6, 8, 11). For clarity, we also indicate the meaning of the vertical lines in the caption to Figs 4, 5, 6, 8, 11: "Vertical dash lines show local sunset and sunrise times."

L260: In the first event $q_a$ decrease is very low in comparison to the other events! This event was also with the month of the large rainfall deficits. Are there any estimate or measurements of NRW amount available? E.g., showing the measured latent heat flux from the EC-tower or lysimeter, leaf wetness sensor or estimates based on any model that predicts dew formation.

➤ We replaced $F_{H2O}$ by LE in Fig. 4b. We do also have micro-lysimeter and leaf wetness measurements from 2020, but unfortunately the micro-lysimeter and leaf wetness data from 2018 are not available. Thus, we calculated the NRW amount according to the equations by Monteith (1957) as suggested below. See methods in Section 3.2.5 (L347-367).

L271: Please explain the gaps in Fig. 6 a-d during P1b

➤ The gaps are due to calibration periods, which is now noted in legend of Fig. 6.

L276: Please refer to $\delta^2 H_a$, $\delta^{18} O_a$ here instead of $\delta_a$

➤ We change all $\delta$ into $\delta^2 H$, $\delta^{18} O$ (e.g., L418, etc.).

L288: Looking at Fig. 6 a and b, there a partially large difference between fNRW and aNRW especially for first and second, but also to some extent for the third event. The authors report later that much of the dew comes from the soil itself and not atmosphere so I would not expect that fNRWand aNRW are identical! Please describe results more carefully here and discuss it later.

➤ We now compare fNRW and aNRW case by case In Section 4.3 at L446-459.

L293: [...] The relationships between the isotopic compositions of fNRW and aNRW were related to RH [...] please add in Fig. 7 a, b, c the RH on the second y-axis. As it is difficult to follow results until L300 without seeing measured isotopes and RH in one plot.

➤ Relative humidity was added in Fig. 7d.

L302: Please explain the deviation of aNRW from the LMWL in Fig. 8. Does the position of aNRW below the LMWL means that aNRW stems from local ET water?

➤ Yes, aNRW below the LMWL means that aNRW stems from local ET water. We added this at L473-475 (this is now Fig. 9).

L306: but for event 2-sampled fNRW under 97 to 98 are similar to that of aNRW. Others show large spread (deep purple triangles)? Is there no other reason that could explain the isotopic position of the samples fNRW that are much below the eq. line? E.g. nighttime evaporation processes of dew water on the leaf canopy. Would be good to check here the latent heat flux of the EC tower measurements for these times. Please add here for the discussion findings from Chen et al. (2019) (see Fig. 5), where data for soil, dewrise and dew water as well as vapor are shown.

➢ Potentially, re-evaporation and guttation could be the reasons. Discussion on re-evaporation and guttation was added in Section 5.2 (L573-591).

L301-310: The authors mention in L157 the measurement of soil moisture (hereafter $\delta s$ ). I couldn't found a description of the data in the results section (already mentioned). Please add this here and describe it. This could clarify in Fig.8 where the water came from soil or evaporation of dew from canopy!

➢ The isotopic composition of soil water was added in new Fig. 8, and addressed in the Results section (L468-472, L482-483), and discussed in 5.1 (L543-545), Section 5.3 (L620-623), Section 5.4 (L647-649, L652-655). We also discuss the effect of re-evaporation of dew in Section 5.2 (L578-582).

L301-310: another point might here that a mix of guttation water with dew might lead to a shift in the isotopic composition. It would at least fit as the deviation was seen for both events for the first sampled foliage water! Please at least mention it and discuss possible affects of guttation on stable isotope composition in the discussion e.g. see Xu et al. (2019).

➢ Discussion on guttation was added in Section 5.2 (L573-591).

L311-320: Not clear to me how the authors finally estimate the contribution of dew or soil distillation on the collected dew water, when the amount of dew, fog or from soil distillation are unknown for the events! This should be clearly describe in the M&M section.

➢ The contribution of aNRW and distillation was obtained from the two end-member mixing model in Section 3.2.4 (L318-346).
➢ As a comparison, in the revised version, the contribution of aNRW and distillation are also obtained from the Monteith (1957) approach (M57), see Section 3.2.5 (L347-367). The corresponding results are compared in Section 4.4 (L505-523).

L323: As aNRW are simulations, the uncertainty of the used assumptions to determine aNRW in the two end-member mixing model should be included and naNRW as well as dDew with naNRW should be reported in the result section.

➢ The results of naNRW are now incorporated in Fig. 7, and mentioned in Results section 4.3 (L488-495).

L323-329: these are results and the used method of e.g. Wen et al. 2012 should be describe in the Material and Method section and results should be shown in the results section!

➢ The methods were moved to methods Section 3.2.2 (L288-296).

L323-344: I recommend enlarging the discussion about the result here in a much broader context. Compare results with previous studies and discuss possible effects from e.g. guttation or dew re-evaporation on the sampled isotopic composition fNRW and the method on the partitioning of NRW inputs using a two end-member mixing model.

➢ A discussion on guttation and re-evaporation effect was added in Section 5.2 (L573-591).

L351: Not sure about this reported values here. A) Please clarify how dDew was potentially 22-83% according to Monteiht 1957? B) Please report methods used here in the manuscript in the M&M section and not adding this info a Table caption (i.e. Table 2). C) More in detail, it is unclear how the authors come up with different times for dDew and aNRW. D) In addition, I recommend to use eq. mentioned in Monteith 1957 to calculate potential contribution of soil distillation and dew, present this result in the result section and compare it with the latent heat flux observations an than discuss it in this section 4.2.

➢ As recommended, we now use the Monteith (1957) approach (M57) to calculate NRW amounts, see methods Section 3.2.5 (L347-367), and Fig. 10a.

L353: Not clear to me how NRW gain is comparable to average ET of 2.8 mm? There were no results on actual NRW water, the authors only report potential NRW+soil distillation which were somehow taken from report rates in Monteith 1957. At least soil distillation is soil dependent and also depends on the canopy or? If soil is bare we might see evaporation instead of soil distillation. This means reported values are location dependent!

➢ We agree with the reviewer that the wording was confusing. We compared the NRW input to the canopy with ET to put the NRW flux in perspective of the overall canopy water budget. The dew and radiation fog produced 0.10–0.41 mm d$^{-1}$ NRW gain on foliage computed from M57 approach, which, compared to evapotranspiration of on average 2.8 mm d$^{-1}$, constitutes a non-negligible water flux into the canopy. See adapted text at L514-516, and L682-684.

L371-372: this values should be reported in the result section and added to the other plots to better distinguish from where water collected on leaves are coming from. From the M&M section it was also not clear how and when this samples were taken e.g. suction cups or destructive? Please add also this missing part in the M&M section

➢ The information for the isotopic composition of soil water is added in Methods Section 3.2.1 (L237-241), and Fig. 8; in Results Section 4.3 (L468-472, L482-483); in Discussion, Section 5.1 (L543-545), Section 5.3 (L620-623), Section 5.4 (L647-649, L652-655).

L380: From my perspective, the reported results (until now) are not a direct indicator that soil evaporation is synchronously happen with condensation. Please reformulate this in a more careful way. Perhaps it would be worth to calculate potential dew and soil distillation based on the eq. from Monteith 1957.

➢ As recommended, we now use the Monteith (1957) approach (M57) to calculate NRW amounts, see methods Section 3.2.5 (L347-367), and Fig. 10a. We discussed the source of distillation in a more careful way in Section 5.4 (L637-662).

L389: Not clear to me why water that comes actually from the soil is not accessible for roots? Vapor transport might be largest during very dry conditions. However, this was only the case for first event. The other two event were observed during months with higher rainfall amounts than long-term average values.

➢ The soil vapor transport from the deeper soil to the rooting zone is now discussed in Section 5.4 (L638-649).
➢ The monthly precipitation was changed to year-to-date precipitation in Fig. 1a, which shows the dry spells during the three events. Volumetric soil water content (SWC) also is also shown in Fig. 1d, which showed the close-to-wilting-point SWC in the main rooting zone (0-15 cm).

L391: Please discuss somewhere that the amount of water transferred by vapor transport from soil depends on soil properties.

➢ We added a statement in Section 5.4: "The process of vapor diffusion from deeper soil layers to the surface strongly depends on soil properties, and thus might differ from site to site" (L646-647). We now also discuss the fact that soil vapor diffusion becomes more important when SWC is close to wilting point, see Section 5.4 (L649-655). We calculated the wilting point from soil texture, which indirectly showed the effect of soil properties on soil vapor diffusion. But we do not have another site as comparison, therefore we did not discuss more about the effect of soil properties on soil vapor diffusion.

L395: From which soil depths is this water coming from? Is this deeper than the effective rooting zone of the grassland? Would be an important point here to discuss, as only deeper than the roots zone located water would actual lead to a benefit of dDew for plants.

➢ Main rooting zone was addedin the revised version in Fig. 1d, and is discussed in Section 5.4 (L640-650).

L401-403: the authors reported the estimated wilting point of the soil in the M&M section. Would be worth to mention this somewhere in the Results section to see if soil was actually near the wilting point during the 3 events, which would emphasize that NRW could reduce water stress during this time and discuss this point, e.g. Groh et al. (2018) that the occurrence of dew during times with water stress might alleviate drought stress for plant.

➢ Wilting point and field capacity was added in the revised version in Fig. 1d, in Section 3.2 (L224-227), in the Appendix C, and is discussed in Section 5.4 (L649-655).

L416: My recommendation for this section is to present less individual results and to focus more on answering the question/objectives of the study and its impact in a broader context.

➢ We revised the conclusion (L664-690).

**Reply to RC2**

Anonymous Referee #2

This study investigates the role of fog and dew deposition in the water budget of a grassland in Switzerland. The authors aim to distinguish different pathways of the liquid water sources, e.g. fog deposition, dew deposition from the atmosphere to the surface, and dew deposition from the soil upwards towards the vegetation. The study uses isotopic composition of H and O in the water vapour in the atmosphere and the liquid water. I think the authors did an tremendous effort in performing a measurement campaign to measure these components during three different nights and in understanding the pathways. This is also an interesting new approach. My main criticism about this manuscript is that the description of all the isotopic ratio's and compositions is written in a too much technical way. The reader is offered a number of values without interpretation what it mean related to the three proposed pathways. In the current shape the paper is only interesting for experts in isotopic signatures and does not serve the wider fog research community, while I think this huge research effort deserves this wider audience. More detailed comments have been listed below. Recommendation: Major revisions required.

➢ We thank the reviewer for her/his constructive comments (i.e., clarifying the motivation why isotopic method is needed, and revising the manuscript for wider audience) and the positive feedback. We provide our answers point-by-point below.

**Remarks**

Ln 7: "In a warmer climate, non-rainfall water (hereafter NRW) formed from dew and fog potentially plays an increasingly important role in temperate grassland ecosystems under the scarcity of precipitation over prolonged periods". Please reword. I find this a confusing sentence, since warmer should be compared to a reference (warmer than....) and secondly I do not see the rationale that in a climate with high temperatures the relative contribution of occult precipitation will increase. Under climate change the hydrological cycle is expected to accelerate, which means more precipitation and thus less relative contribution by occult precipitation. Please rephrase.

➢ We reworded it as "During dry spells, non-rainfall water (hereafter NRW) mostly formed from dew and fog potentially plays an increasingly important role in temperate grassland ecosystems with ongoing global warming." (L7-8)

Ln 11: remove "at all"

➢ Removed. (L11)

Ln 13 : the abstract misses a statement why isotopes are needed to identify the pathways. I would say that if I install eddy covariance, a fog collector and a microlysimeter, I can also obtain the mechanisms contributing to the NRW budgets. So motivate why a more difficult method is needed.

➢ We added the following statement to the Abstract to clarify why stable isotopes are needed: " Stable isotopes provide additional information on the pathways from water vapor to liquid water (dew and fog) that are cannot be measured otherwise " (L16-17).
➢ We now also address the uncertainty of EC measurements in the Introduction at L84-94, which also is a reason why we think stable isotopes provide additional information that is otherwise not available.
➢ The difficulties of partitioning distillation and aNRW is addressed in the Introduction Ln110-128, and Section 5.3 (Ln592-636).

Ln 10-20: an interpretation should be provided what a certain permille for a certain isotope means. The reader is now overloaded with values without guidance about the interpretation. In such a way the paper is only interesting for a small incrowd.

> We revised abstract to make it more attractive to non-isotope specialists and removed isotope specific information (Ln7-28) and thus do not report the per mil information anymore.

Ln 34-35: cite in chronological order, here and throughout the whole manuscript.

> Citation was changed in chronological order (e.g., L31-32).

Ln 80-85: please add a few lines what are the physical reasons why local evaporation and entrainment at the PBL differ so much in *d*. This will help the non-involved reader.

> For example, at the local scale, as compared to the higher *d* vapor of the entrainment from free troposphere, local evaporation is a vapor source with lower *d*, because soil water vapor isotopes at the evaporation front had lower *d* value (Parkes et al., 2017). See L154-156.

Ln91: in height: please be more precise. Do you mean in the soil?

> Changed to "a.g.l.", see L116.

Ln 104: please specify in more detail the what is meant by ecological relevance and how you will measure that.

> The third aim is actually more a long-term perspective/goal of our research. In this paper we make a first step in that direction, and thus we agree that this can be misinterpreted by the readers. We therefore decided to remove the third aim from this paper (L129-130) in the form of a specific aim of this one manuscript, and now mention our overarching goal in a more general way at the end of the Introduction (L133–134). We nevertheless added the data of soil water content (Fig. 1d) and the isotopic composition of leaf water (Table 2, new version) and soil water (new Fig. 9) for discussing subsections "5.2 Processes affecting non-rainfall water on foliage (L573-591)", and "5.4 Potential effects of non-rainfall water on local water cycling" (L637-L662). The samplings of leaf water and soil water were added in Methods Section 3.2.1 (L236-241).

Section 2.2.1: please add which software was used for the flux processing and with which settings.

> For initiated readers we have added more details about the settings to Appendix B (L1080-1087) and refer to it in the main text (L179-181).

Ln 130: I am quite concerned about the height of the flux measurement since 2.4 m is very close to the surface, which means that there will be a relatively large "flux loss". Please specify how much this is and whether it will influence your results.

> To understand your concern we would need to know your background, because our concern is when people measure too high above the surface in complex terrain, because this would miss important parts of the fluxes. We therefore have tried to explain this in more detail (see L314-318) based on our earlier publication by Eugster and Merbold (2015) where the reader can find even more details on this aspect. The added text is:
> "During the three events in this study, the NBL top was at 152 m, 35 m, and 105 m a.g.l., respectively (Fig. 3). Therefore, the EC measurement setup at 2.4 m a.g.l. are expected to have captured roughly 99% of the expected flux (Eugster and Merbold, 2015). The roughness sublayer (1–3 times the vegetation height according to Oke (2002)) was at 0.2–0.9 m at the Chamau site, therefore the EC instruments were installed well above the roughness sublayer."

Ln 130: What happens to the contribution in the transport of the turbulence that happens below 2.4 m and is as such not seen by the EC sensor? Since the site is that the bottom of a valley I can imagine that thin katabatic flows are present from the valley walls

to the valley and that they generate small scale turbulence. Does ignoring this component affect your conclusions. Please reflect and if possibly quantify.

- ➢ See Ln314-318.
- ➢ "A measurement height of 2.4 m is a good compromise as was discussed in detail e.g. by Eugster and Merbold (2015), Section 2.3: if a greater height above short-statured grassland vegetation is chosen, then the "flux loss" mentioned by the reviewer will be larger at night (not smaller!), hence the measurement height must always be considered in relation to the roughness sublayer height of the vegetation. With 2.4 m we are well above the roughness sublayer, but we agree that over tall vegetation (forests) it is often not possible to measure outside the roughness layer, hence the distance of the sensors above a forest canopy must be much larger than that over a grassland."
- ➢ Please recall that the turbulent flux linearly decreases from the surface to the inversion layer (e.g.Stull (1988), and Eugster and Merbold (2015)), so the issue increases with increasing measurement height. Thus, we respectfully disagree on this aspect with the reviewer and did our best to clarify this in the text.

Ln 142: The equation is incomplete. The upwelling LW_up flux consists of sigma*T^4 +(1-emiss)+LW_down and the latter component is missing. This would not have been a problem if the emissivity of the surface would equal 1, but you explicitly report it amounts to 0.98. Please recalculate your results.

- ➢ See equation 7.3.2b in Stull (1988), page 259:

$$ I{\uparrow}_s \ = \ \varepsilon_{IR} \ \sigma_{SB} \ T^4 \qquad\qquad (7.3.2b) $$

where $\sigma_{SB} = 5.67 \times 10^{-8}$ W·m$^{-2}$·K$^{-4}$ is the *Stefan-Boltzmann constant.* The infrared *emissivity,* $\varepsilon_{IR}$, is in the range 0.9 to 0.99 for most surfaces.

- ➢ And also equation 5.18 in Garratt (1992), page 123:

surface, with ....

Turning now to the surface fluxes, the upwards longwave flux $R^u_{L0}$ (terrestrial radiation) results from longwave emission from the surface. For grey-body emission, it is given by

$$ R^u_{L0} = - \varepsilon_s \sigma T_0{}^4 \qquad\qquad (5.18) $$

where $T_0$ is the surface temperature. Most natural surfaces are almost black, with $\varepsilon_s$ greater than 0.9; values for a range of surfaces are shown in Appendix Table A8. The downwards longwave flux $R^d_{L0}$ results from emission of infrared radiation from the whole atmosphere, and from clouds. Low clouds, in particular, are dominant since they tend to radiate as black bodies at their cloud-base temperatures. This downwards flux is readily measured, but its computation is by no means straightforward, even in the absence of clouds. Under clear skies, it is strongly dependent on the vertical distributions of temperature and humidity throughout the troposphere and particularly within the ABL.

.... the downwards longwave flux $R^d_{L0}$, both in

➢ We however agree with the reviewer that for net radiation the difference between LW_up and LW_down would be the correct approach, but not for back-calculating surface temperatures, where it is essential to use gross fluxes. From LW_down one could calculate sky/cloud base temperature, and from LW_up one can calculate Earth surface temperature.

Ln 199: why wasn't potential temperature gradient used for the PBL height determination?

➢ According to Stull (1988, page 503) potential temperature would be used to determine the top of the stable boundary layer, i.e. the height where the lapse rate is adiabatic, whereas the inversion top, i.e. the height where the lapse rate is isothermal, is deduced from absolute, not potential temperature. A change in the concept would require a change in the wording, but isothermal is easier to understand than adiabatic for broader readers.

The height, h, of the top of the SBL (i.e., the SBL *depth*) is more difficult to quantify, because in many cases the SBL blends smoothly into the residual layer (RL) aloft without a strong demarcation at its top. Thus, many of the definitions of SBL depth that appeared in the literature are based on relative comparisons of SBL state aloft to near-surface state. For example, h can be defined as the lowest height where:

- $\partial\overline{\theta}/\partial z = 0$      (stable layer top; i.e., the height where the lapse rate is adiabatic)
- $\partial\overline{T}/\partial z = 0$      (inversion top; i.e., the height where the lapse rate is isothermal)
- TKE $= 0$      (top of the turbulent layer, or mixed layer if one exists)
- TKE $= 0.05$ TKE    (height where turbulence is 5% of its surface value)
- $\overline{u'w'} = 0$      (top of the stress layer)
- $\overline{u'w'} = 0.05\ \overline{u'w'}_s$   (height where stress is 5% of its surface value)
- $\overline{M}$ is maximum      (the nocturnal jet level)
- $\overline{M} = \overline{G}$      (bottom of free atmosphere, where winds are geostrophic)
- sodar returns disappear      (top of the layer with temperature fluctuations)

Ln 200-202: I think it is this method should be reconsidered. The NBL depth can vary spatially enormously, especially in complex terrain where the experiment was done (i.e. a valley) while the ECMWF product is at 30 km spatial resolution. Furthermore the vertical grid spacing of ECMWF is too coarse to detect the NBL height properly. Also the reported values are very high for nights where you can expect fog or dew. As a rule of thumb one can use that the NBL depth amounts to 700* u_star (friction velocity). That would mean that here the u_star would be 1 m/s and that is really really high for nights with fog or dew.

➢ We replaced the ECMWF-based estimates by COSMO model data with the resolution of 1.1 km (meridional) × 1.1 km (zonal) over Switzerland and 60 vertical levels (Doms et al., 2018; Westerhuis et al., 2020).
➢ See Section 3.2.3 (L311-315).

Ln 200-202: concerning Fig. 3 I doubt whether the interpretation is correct since I think at the y axis the height above sea level is shown. The surface inversion should be at the surface (i.e. 0 m) right? Not at 650 m above ground level. This can also change the story about my previous point.

➢ Thank you for this careful inspection. There was indeed a mistake in the computation of the vertical height, for which we apologize.

➤ We replaced the ECMWF-based estimates by COSMO model data with the resolution of 1.1 km (meridional) × 1.1 km (zonal) over Switzerland and 60 vertical levels (Doms et al., 2018; Westerhuis et al., 2020).

➤ See Section 3.2.3 (L311-315).

Ln 211: "while in saturated conditions, fNRW was a mix of aDew and aFog". I disagree on this since it is very hard to create fog in a night with a lot of dew at the same time. Dew takes out water vapour so fog in inhibited to develop. This contrasts with your statement.

➤ This is a misunderstanding, we completely agree with the reviewer, but realize that our text was not clear enough. We thus reworded as "while with saturated conditions, fNRW was originating from dew or from fog (aNRW), which could lead to a mixture of water from both sources over the course of a night when dew and fog occur intermittently". See Ln322-324.

Ln 213: typo: is -> as

➤ Thanks for pointing out that our wording was not clear; we actually mean the following (see L324-325) and thus reworded this statement accordingly:

➤ "Dew forming in unsaturated conditions is a mixture of aNRW and distillation but lacks contribution from fog deposition." (L325-326)

Ln 222: It is good that you are honest about your assumption. But how realistic is the assumption. Could you spend a few words on it?

➤ We added the M57 approach to partition aNRW and distillation, which allows us to compare the M57 with our mixing model. See methods in Sections 3.2.4, and 3.2.5, and results in Fig. 10 and Section 4.4, and discussion in Section 5.3.

Ln 248: net longwave radiation loss: can you be more quantitative? Was it -80, -50 or -10 W/m2

➤ We added long-outgoing radiation, and short-wave ingoing radiation, see Fig. 5a, b, and Section 4.1 (L391).The values we report in Section 4.1 are –36 W m$^{-2}$ long-wave radiation loss at sunset.

Fig. 5: the top of panel b can be at 12 of 15 g/kg.

➤ We changed it as suggested, see new Fig. 5c.

Section 3.2-3.4 are hard to follow and only useful for specialists in isotope measurements. The numbers a presented as a flood of values without discussion or interpretation what they mean. I did not get so much from these sections.

➤ We add the interpretation of the data into results, see Section 4.2, 4.3 and 4.4 in revised version.

Ln 354: "This amount of NRW gain was comparable with the average evapotranspiration rate of 2.8 mm day-1 (daytime) during ...". I do not understand what the authors want to say with this statement. How is dew at night comparable with evaporation during the day. The mechanisms are completely different!

➤ We reworded it to get rid of confusion. We actually compared 24-hour averages, otherwise it would indeed not make sense, we completely agree. See L514-516.
"The dew and radiation fog potentially produced 0.10–0.41 mm d$^{-1}$ NRW gain on foliage, which, compared to evapotranspiration of on average 2.8 mm d$^{-1}$, constitutes a non-negligible water flux into the canopy."

Ln 377: " minor influence of large-scale air advection": this is in complete contrast to the large diurnal cycle of specific humidity that is clearly driven by katabatic flows, as shown by the authors.

➢ We reword it, see L528-531.
➢ With "large-scale air advection" we mean the advection with the synoptic scale flow (order of 1000 km in horizontal distance), which was very small, given the weak pressure gradient over Europe in the context of central European anticyclones during all three events. Katabatic flows are density driven flows of mesoscale extent, induced by the local topography and the regional thermodynamic conditions in a situation with weak synoptic-scale influence. Thus, we agree that the large diurnal cycle of specific humidity includes local katabatic drainage flow, but this is the typical meteorological setting during stationary fair weather conditions with small large-scale advection only (which does not mean small local-scale advection). We added references to clarify this aspect of katabatic drainage flows.

Ln 393: $u \rightarrow u_{2m}$

➢ Changed, see L659.

Fig. 10: I am not sure both panels are meaningful since in the definition of RH, the temperature plays an important role through the denominator in RH =q/q_sat(T). So I have the feeling we look twice at the same effect.

This is now Fig. 12 in the revised manuscript and only shows one panel. The two primary environmental controls of dexcess ($d_a$) are known from theory and empirical data to be surface relative humidity and to a smaller extent surface temperature as established in the literature, e.g. Craig and Gordon (1965), Pfahl and Wernli (2008), Welp et al. (2012), and Aemisegger et al. (2014), Aemisegger and Sjolte (2018) and many others. But we agree that temperature and relative humidity are not independent variables, hence we decided to only show one panel, which is the most relevant information for isotope specialists.

Formula B1: Perhaps I overlook something but I have the feeling that equation B1 is wrong when I compare it to Equation 3.19 in Campbell and Norman (1998). In CN98, the vapour concentration should be entered in mol/mol, but here in Pa. Please check, and check whether this affects your results.

➢ There are no units for both our equation B1 and Equation 3.19 in Campbell and Norman (1998), page 47. It is just ratio. Here I show how we rewrite the equation:

$$q_0 = \frac{0.622 \cdot C_{va}}{1 - 0.378 \cdot C_{va}} = \frac{0.622 \cdot \frac{e_{s0}}{p}}{1 - 0.378 \cdot \frac{e_{s0}}{p}} = \frac{0.622 \cdot e_{s0}}{p - 0.378 \cdot e_{s0}} \left(\frac{kg}{kg}\right) = \frac{0.622 \cdot e_{s0}}{p - 0.378 \cdot e_{s0}} \left(\frac{kg}{kg}\right) * 1000 \left(\frac{g}{kg}\right)$$

$$= \frac{622 \cdot e_{s0}}{p - 0.378 \cdot e_{s0}} \left(\frac{g}{kg}\right)$$

The specific humidity $q$ is the mass of water vapor divided by the mass of moist air, and is related to the mole fraction by:

$$q = \frac{0.622 C_{va}}{1 - 0.378 C_{va}} . \qquad (3.19)$$

The units of $r$ and $q$ usually are g/kg. Table 3.2 compares values of $r$ and $q$ with sample values of the other variables representing moisture in air.

[revised manuscript text omitted]

---

## Referee Report (RR1)

Review of hess-2020-493.R1

The authors have done a rigorous revision of the paper and provided a thorough response to the reviewer remarks. The manuscript reads much better now and is more accessible to a wider audience. There are a number of minor remaining aspects that need to be addressed before the manuscript can be published.

Remarks:

Ln 10: ...the soil....

Ln 179: add a period after (2008)

Ln 196: Still I think the formula used here is incomplete. Although I know the books by Stull and Garratt, a more recent book by Moene and Van Dam mentions to include the upwelling component that acts as reflecting part of the downwelling radiation in their formula 2.28:

Typical values for the surface emissivity can be found in Table 2.1. For most surface types $\varepsilon_s$ is between 0.9 and 0.99. This implies that there is *some* reflection of longwave radiation.[2] Hence, the total upwelling longwave radiation is the sum of the emitted radiation, $L_e^\uparrow$ (see earlier) and the reflected radiation:

$$L^\uparrow = L_e^{\ \uparrow} + (1 - \varepsilon_s) L^\downarrow \qquad (2.28)$$

Your conclusions will not necessarily change, but I think it would be good to have insight in the impact of the reflecting part.

Ln 313: I think the approach using COSMO is a good alternative compared to ECMWF data. Nevertheless all models do have difficulties with representing nocturnal boundary layer (depth)s, so perhaps a few words mentioning this uncertainty would be meaningful here. Also a few words of discussion about whether a different definition of the nocturnal PBL height would have affected your results would be illustrative and may help to defend the robustness of your results.

Ln 361: following as follows (Monteith, 1957). Reword to "as follows" or "following M57"

Ln 363: horizontal wind speed gradient => reword, it is the vertical gradient of the horizontal wind.

Equation 22: The exchange function 1/(1+10*Ri) is rather under debate in the field of stable boundary layer research. I would say the used function is very effective in transport, more than is traditionally used in micrometeorology, where (1-5*Ri)^2 is used (without diffusion for Ri>0.2). With that function much less diffusion occurs (see red line, blue line is your function). Please provide an assessment whether this affects your results.

[Figure]

Equation 23: Maybe I missed it, but how did you calculate or measure the partial derivatives du/dz and dT/dz? As you do not have tower measurements you cannot measure them. In case you would have a tower, then you can only use the bulk richardson number.

Figure 3: Since COSMO is now used the reference to Hersbach et al., 2020 is not relevant anymore?

Figures: I thick it is good for the readability to list in the captions also what P1a,b, and P2a,b are. Now the reader has to go back to the start of the paper to rediscover.

---

## Author Response (AR2)

**Response to Editor and Reviewers' comments**

We thank the editor and reviewer for their careful reassessment of our revised manuscript. We did our best to benefit from all feedback to further improve our manuscript. We provide our responses below in blue colour and hope that with these modifications our manuscript can now be accepted for publication in HESS.

**Main changes**

Besides the changes commented by two reviewers, the main changes are as below:

- we changed Eq. 22 and Eq. 23 to get rid of using wind speed and temperature gradients that we assumed in the old version, see methods at L372-380, Figure 10, and results at L526-535. For example, the amount of aNRW was changed from "0.07 – 0.38 mm" to "0.16–0.57 mm ". We added the discussion of stability term in Eq. 22, see L647-652. This did not change our analysis and conclusion.

**Editor comment**

Your revised manuscript has now been seen by two referees that have previously also reviewed your original submission. I am happy to see that both referees agree in their assessment that the manuscript has improved substantially. However they both also list a number of comments that need to be addressed before I can consider the manuscript for final publication. If you manage to address all remaining issues, I will not iterate the manuscript again with the reviewers, but I will evaluate the next iteration myself in order to save time.

Looking to forward to receiving your revised version, which should include a point-by-point response to the issues raised.

Report #1

The authors have done a rigorous revision of the paper and provided a thorough response to the reviewer remarks. The manuscript reads much better now and is more accessible to a wider audience. There are a number of minor remaining aspects that need to be addressed before the manuscript can be published.
Remarks:
Ln 10: ...the soil....

Changed as suggested at L10.

Ln 179: add a period after (2008)

Changed as suggested. A blank space was added after "(2008)". L179

Ln 196: Still I think the formula used here is incomplete. Although I know the books by Stull and Garratt, a more recent book by Moene and Van Dam mentions to include the upwelling component that acts as reflecting part of the downwelling radiation in their formula 2.28:

> Typical values for the surface emissivity can be found in Table 2.1. For most surface types $\varepsilon_s$ is between 0.9 and 0.99. This implies that there is *some* reflection of longwave radiation.[2] Hence, the total upwelling longwave radiation is the sum of the emitted radiation, $L_e^{\uparrow}$ (see earlier) and the reflected radiation:
>
> $$L^{\uparrow} = L_e^{\uparrow} + (1 - \varepsilon_s) L^{\downarrow} \qquad (2.28)$$

Your conclusions will not necessarily change, but I think it would be good to have insight in the impact of the reflecting part.

We replace $LW_{out}$ by $LW_{surface}$ in Eq. (1) and added the information that we now derived $LW_{surface}$ using the first-order reflection of down-welling longwave radiation to conform with Moene and van Dam (2014). This changes the dew yield by ca. 0.002 mm per night of a total of 0.04–0.05 mm, which means that this modification changes our previous estimates by 4–5%. This is within the overall experimental uncertainty and thus did not change our findings. L196-201.

Ln 313: I think the approach using COSMO is a good alternative compared to ECMWF data. Nevertheless all models do have difficulties with representing nocturnal boundary layer (depth)s, so perhaps a few words mentioning this uncertainty would be meaningful here. Also a few words of discussion about whether a different definition of the nocturnal PBL height would have affected your results would be illustrative and may help to defend the robustness of your results.

We added that "The top of NBL is difficult to quantify, because in many cases the NBL does not have a strong demarcation at its top. Therefore, many definitions of the NBL are based on relative comparisons of the stable boundary layer state aloft to near-surface state (Stull, 1988)." L317-319

We then further clarified (L326–329) that: "Here we simply use NBL as a background information on atmospheric stability, but did not use it for nocturnal boundary layer budgets (Denmead et al., 1996) as was done by Stieger et al. (2015) at this exact same site, and thus the uncertainty in the exact value extracted for the NBL top from the COSMO-1 model output has no influence on our dew estimates."

Denmead, O. T., Raupach, M. R., Dunin, F. X., Cleugh, H. A., and Leuning, R.: Boundary layer budgets for regional estimates of scalar fluxes, Global Change Biology, 2, 255-264, https://doi.org/10.1111/j.1365-2486.1996.tb00077.x, 1996.

Stieger, J., Bamberger, I., Buchmann, N., and Eugster, W.: Validation of farm-scale methane emissions using nocturnal boundary layer budgets, Atmos. Chem. Phys., 15, 14055-14069, https://doi.org/10.5194/acp-15-14055-2015, 2015.

Ln 361: following as follows (Monteith, 1957). Reword to "as follows" or "following M57"

Changed as suggested. L372.

Ln 363: horizontal wind speed gradient => reword, it is the vertical gradient of the horizontal wind.

We replace Eq. 22

$$F = \frac{\kappa^2 \cdot z^2 \cdot (\frac{\partial u}{\partial z}) \cdot (\frac{\partial \chi}{\partial z})}{1 + \sigma \cdot Ri} \quad \text{by} \quad F = \frac{\kappa^2 \cdot z_{2m} \cdot u \cdot (\frac{\partial \chi}{\partial z})}{\ln(\frac{z}{z_0})} \cdot \Phi$$

therefore the term d$u$/d$z$ was removed from new Eq. (22). L372-380.

Equation 22: The exchange function 1/(1+10*Ri) is rather under debate in the field of stable boundary layer research. I would say the used function is very effective in transport, more than is traditionally used in micrometeorology, where (1-5*Ri)^2 is used (without diffusion for Ri>0.2). With that function much less diffusion occurs (see red line, blue line is your function). Please provide an assessment whether this affects your results.

[Figure]

We agree that using the term $(1 - 5\,Ri)^2$ would substantially change the results, therefore we addressed this issue at L651-653 "In future research, we recommend combining isotopic composition measurements with lysimetric measurements to partition NRW from ambient water vapor and distillation. This would provide a useful benchmark to better evaluate the isotope-based estimates of NRW inputs."

In addition, we added the discussion of the uncertainty arising from the stability term (L647-651) "In the M57 approach as shown in Eq. 22, the stability term $\Phi=1/(1+10\cdot Ri)$ was used. However, the stability term is sometimes written as $\Phi = [1 - 16 \cdot (z - z_d)/L]^{-0.5} =[1 - 16 \cdot Ri]^{-0.5}$ for $Ri < -0.1$, and $\Phi = [1 + 5 \cdot (z - z_d)/L] =[1 - 5 \cdot Ri]^{-1}$ for $-0.1 \leq Ri \leq 1$ as e.g. in Monteith and Unsworth (2013), which would cause higher condensation rates when using Eq. 22 (see Fig. E1 in Appendix E), hence lower relative contribution of distillation in the total NRW than given the term $\Phi=1/(1+10\cdot Ri)$."

Equation 23: Maybe I missed it, but how did you calculate or measure the partial derivatives du/dz and dT/dz? As you do not have tower measurements you cannot measure them. In case you would have a tower, then you can only use the bulk Richardson number.

Thanks for pointing out this. In the old version,  we assumed that $u$ =0 at $z = z_0 + z_d$ to try to approximate the partial derivatives used by M57 as closely as possible from our available measurements. We however agreed that this

assumption caused very large uncertainty of the condensation rate in Eq. 22. Because we now have eddy covariance flux measurement available, we determined $Ri$ from the Monin-Obukhov (1954) stability parameter z/L that is directly measured by eddy covariance. Therefore, we replaced the old Eq. 22

$$F = \frac{\kappa^2 \cdot z^2 \cdot \left(\frac{\partial u}{\partial z}\right) \cdot \left(\frac{\partial \chi}{\partial z}\right)}{1 + \sigma \cdot Ri} \qquad \text{by} \qquad F = \frac{\kappa^2 \cdot z_{2m} \cdot u \cdot \left(\frac{\partial \chi}{\partial z}\right)}{\ln\left(\frac{z}{z_0}\right)} \cdot \Phi$$

and replaced Eq. 23 $\quad Ri = \dfrac{g}{T_a} \dfrac{\left(\frac{\partial T}{\partial z}\right)}{\left(\frac{\partial u}{\partial z}\right)^2} \qquad \text{by} \qquad Ri = \dfrac{z_{2m}/L}{1 + 5 \cdot z_{2m}/L}$

By making these changes, we could avoid making estimates for d$u$/d$z$, and d$T$/d$z$. See details at L372-380. This caused higher aNRW rate, but makes our two end-member mixing model more close to M57 approach (see results L526-535, and Figure 10).

Figure 3: Since COSMO is now used the reference to Hersbach et al., 2020 is not relevant anymore?

Thanks for pointing out the mistake. We changed the reference into "(Doms et al., 2018; Westerhuis et al., 2020)". See Figure 3 caption.

Figures: I think it is good for the readability to list in the captions also what P1a,b, and P2a,b are. Now the reader has to go back to the start of the paper to rediscover.

We added the captions for P1a, b, and P2a, b in the related figures 5, ,6 ,7 ,8 ,10,and 11.

**Reviewer #2**

The authors considered and clarified within their revised manuscript most of my concerns. Please find from below some last comments or open question. Thanks again for this nice study and analysis on non-rainfall water based on isotopic data, which was a very interesting read.

Specific comments:

L96: Not all surfaces are cover by vegetation so I recommend to generalized it and state [..] dew formation: 1) the downward pathway through the condensation of water vapor on the plant and/or soil surface.[….] or write Monteith (1957) identified two input pathways for dew formation on surface covered by vegetation.

We changed it as suggested. "1) the downward pathway through the condensation of ambient water vapor on the plants and/or on soil surface" L95-96

L215-217: To my understanding it appears that the main rain deficit occurred before event 1 (-44%). After this rain was rather similar to the long-term mean year-to-precipitation as the deficit in % did not further increase nor decrease substantially. However, this also means that the rain after event 1 was not sufficiently to refill the depleted soil water storage during the other periods, which explains why the observed soil water content remained low until event 3.

We added this explanation after the SWC description. "The rainfall after event 1 was not sufficiently to refill the deficient soil water storage, which explains why the observed SWC remained low until event 3." L231-232

L232-237: thanks for the clarification. However, for event 2 bi-hourly values were taken for NRW droplets and leaf samples. Are these leaf samples correspond to that of the NRW droplets? In addition, dew or fog accumulates over time on the plant surface, so it would be good to make clear if you actually measured the NRW droplets on same leaves of the corresponding plant or on different leaves of the same plant species in the area?

Why this might be important: NRW droplets evolve over time, so if you measure on the same plant in the night you actually can measure NRW for a specific time period, or if you measure it on different leaves than you basically sample the accumulated NRW signature since the start of the NRW formation. Please clarify this here.

We added "randomly" into the sentence as shown at L235-236:

"NRW droplets on foliage (fNRW) were absorbed in triplicates with cotton balls from the leaf surfaces of randomly selected plants". This means the NRW samples we took from the leaf surfaces were accumulated NRW.

And at L240-241:

"Simultaneously, leaf samples were taken in triplicates from the randomly selected plants for the three species after softly drying the leaf surfaces with tissue paper.".

We added at L241-243:

"To prevent the disturbance of destructive sampling on the effect of dew and fog formations, the NRW droplets and leaf samples were taken from different plants of the same species in the sampling area."

L239-241: please also describe more in detail how bi-hourly soil samples were taken

We added an explanation into the sentence as shown at L245-246:

"soil samples in event 2 was taken without replicate within 2 h before sunset, as well as every two hours (i.e., four times of sampling in event 2) during the night".

L382: please add abbreviation RH behind relative humidity

We added "(RH)" after "relative humidity". L395.

L354: please change to: which was computed from the simulated wet vegetation surface temperature $T_{0w}$, and the observed soil….

We changed the sentence into "which was computed from the simulated wet vegetation surface temperature $T_{0w}$, and soil temperature $T_{s1cm}$". L365-366.

L585: I guess different root water uptake might be also a factor here (different depths and amount of water).

We added "and root water uptake" at L597:

"In our study, we found significant among-species differences in $\delta^{18}O$ and $\delta^{2}H$ of leaf water (Table 2), most likely resulting from species-specific leaf water evaporation and root water uptake".

L621-623: This might in addition also relate to spatial variability of shallow soil moisture and its stable isotope values

We added at L634-635:

"Furthermore, the spatial variability of shallow soil water content and its isotopic composition might enlarge the variability of $\delta^{18}O$ and $\delta^{2}H$ for distillation."

L625: Indeed that would be helpful! But I think it might be also related to the way how to measure NRW on leaves. Taking the moisture from the leaves by cotton balls will first disturb the further formation or condensation of dew or fog on the leaf (in case sampling occurs at the same leaf during the night), or in case that NRW was sampled at nearby leaves that water taken from the leaf contains water and isotopic signature that is representative for the entire period since start of formation of NRW.

We added at L641-642:

"Another reason might be that the NRW droplets we took from the different leaves represent the accumulated NRW, while the temporal variability of NRW droplets on foliage might enlarge the uncertainty of NRW partitioning."
And also in methods L235-243.

L640-644: Please make two sentence out of it and omit as already mentioned in the first round to place too much information into brackets.

We changed it into two sentences:

"The ecological relevance of distillation can be expected if the transfer of moisture is from one hydrological pool that is inaccessible to plants to another that is actually accessible to plants. For example, distillation could transfer soil-diffusing vapor from layers deeper than the effective rooting zone of grassland to droplets forming or depositing on leaf surfaces or surface soil where it can be accessed by the fine roots." (L660-664)